# A revised global ozone dry deposition estimate based on a new two-layer parameterisation for air-sea exchange and the multi-year MACC composition reanalysis

Ashok K. Luhar[1], Matthew T. Woodhouse[1], Ian E. Galbally[1]

[1]CSIRO Oceans and Atmosphere, Aspendale, 3195, Australia

*Correspondence to*: Ashok K. Luhar (ashok.luhar@csiro.au)

**Abstract.** Dry deposition at the Earth's surface is an important sink of atmospheric ozone. Currently, dry deposition of ozone to the ocean surface in atmospheric chemistry models has the largest uncertainty compared to deposition to other surface types, with implications for global tropospheric ozone budget and associated radiative forcing. Most global models assume that the dominant term of surface resistance in the parameterisation of ozone dry deposition velocity at the oceanic surface is constant. There have been recent mechanistic parameterisations for air-sea exchange that account for the simultaneous waterside processes of ozone solubility, molecular diffusion, turbulent transfer, and first-order chemical reaction of ozone with dissolved iodide and other compounds, but there are questions about their performance and consistency. We present a new two-layer parameterisation scheme for the oceanic surface resistance by making the following realistic assumptions: (a) the thickness of the top water layer is of the order of a reaction-diffusion length scale (a few micrometres) within which ozone loss is dominated by chemical reaction and the influence of waterside turbulent transfer is negligible; (b) in the water layer below, both chemical reaction and waterside turbulent transfer act together and are accounted for; and (c) chemical reactivity is present through the depth of the oceanic mixing layer. The new parameterisation has been evaluated against dry deposition velocities from recent open-ocean measurements. It is found that the inclusion of only the aqueous iodide-ozone reaction satisfactorily describes the measurements. In order to better quantify the global dry deposition loss and its interannual variability, modelled 3-h ozone deposition velocities are combined with the 3-h MACC (Monitoring Atmospheric Composition and Climate) reanalysis ozone for the years 2003–2012. The resulting ozone dry deposition is found to be $98.4 \pm 30.0$ Tg $O_3$ yr$^{-1}$ for the ocean and $722.8 \pm 87.3$ Tg $O_3$ yr$^{-1}$ globally. The new estimate of the ocean component is approximately a third of the current model estimates. This reduction corresponds to an approximately 20% decrease in the total global ozone dry deposition, which (with all other components being unchanged) is equivalent to an increase of approximately 5% in the modelled tropospheric ozone burden and a similar increase in tropospheric ozone lifetime.

# 1 Introduction

In the troposphere, the budget of ozone ($O_3$) is determined by its transport from the stratosphere, dry deposition at the Earth's surface, and chemical production and loss. Dry deposition is a significant sink of ozone (Galbally and Roy, 1980), influencing ozone mixing ratio, its lifetime and long range transport. The average dry deposition velocity of $O_3$ to the ocean is less than that to terrestrial surfaces, but because of the larger coverage of the Earth's surface by the oceans there is substantial dry deposition to water. A current estimate of total global dry deposition of $O_3$ is $1094 \pm 264$ Tg yr$^{-1}$ (IPCC, 2013; Young et al., 2013), of which about 35% is to the ocean (Ganzeveld et al., 2009; Hardacre et al., 2015). Hardacre et al. (2015) observed that ozone dry deposition to the water surface in models has the largest uncertainty compared to other surface types. A proper treatment of dry deposition to the ocean in atmospheric chemistry models is thus necessary for more accurate ozone estimates and better representation of feedback cycles, e.g. that involving iodine chemistry (Carpenter et al., 2013). Although dry deposition of ozone to the ocean is the focus of the present paper, we also place ocean dry deposition in the context of total global dry deposition. In this paper the word deposition means dry deposition.

Dry deposition flux, $F_{O_3}$, of ozone to the surface is normally calculated as the product of its concentration, $[O_3]$, in the air near the surface and a (downward) dry deposition velocity, $v_d$ :

$$F_{O_3} = v_d . [O_3] . \tag{1}$$

A common approach to parameterising $v_d$ is to express it as a linear sum of three resistances (e.g., Wesely, 1989):

$$v_d = \frac{1}{r_a + r_b + r_c} , \tag{2}$$

where the aerodynamic resistance $r_a$ is the resistance to transfer by turbulent mixing in the atmospheric surface layer, the atmospheric viscous (or quasi laminar) sublayer resistance $r_b$ is the resistance to movement across a thin layer ($0.1 - 1$ mm) of air that is in direct contact with the surface, and the surface resistance $r_c$ is the resistance to uptake by the surface itself that can be controlled by physical, chemical, biological or other processes depending on the surface type and the species of interest.

At this point it is useful to define the waterside layers near the sea surface that are relevant (Figure 1). The top few millimetres of the sea surface is often termed the sea surface microlayer which may be composed of various sublayers or scales depending on the physical, chemical or biological properties being considered (Soloviev and Lukas, 2014; Carpenter

et al., 2015). Very close to the water surface is a viscous sublayer (~ 1 mm) within which viscous processes dissipate the turbulent kinetic energy associated with the smallest of the eddies (of the size of Kolmogorov microscale) into heat. Thus the viscous sublayer thickness is of the order of the level at which the turbulent eddy diffusivity falls below the kinematic viscosity. A level exists within the viscous sublayer at which the diminishing eddy diffusivity falls below the molecular

5 diffusivity, and this level is approximately the thickness of the diffusive sublayer (~ 50 μm for ozone). Embedded within the diffusive sublayer can be another sublayer (which we call reaction-diffusion sublayer) characterised by chemical reactivity and molecular diffusivity, whose thickness is scaled by a reaction-diffusion length scale (typically 3 μm for the ozone-iodide reaction in water). In the surface turbulent layer (or mixing layer) (~ 10–50 m) below the surface microlayer, turbulent processes dominate.

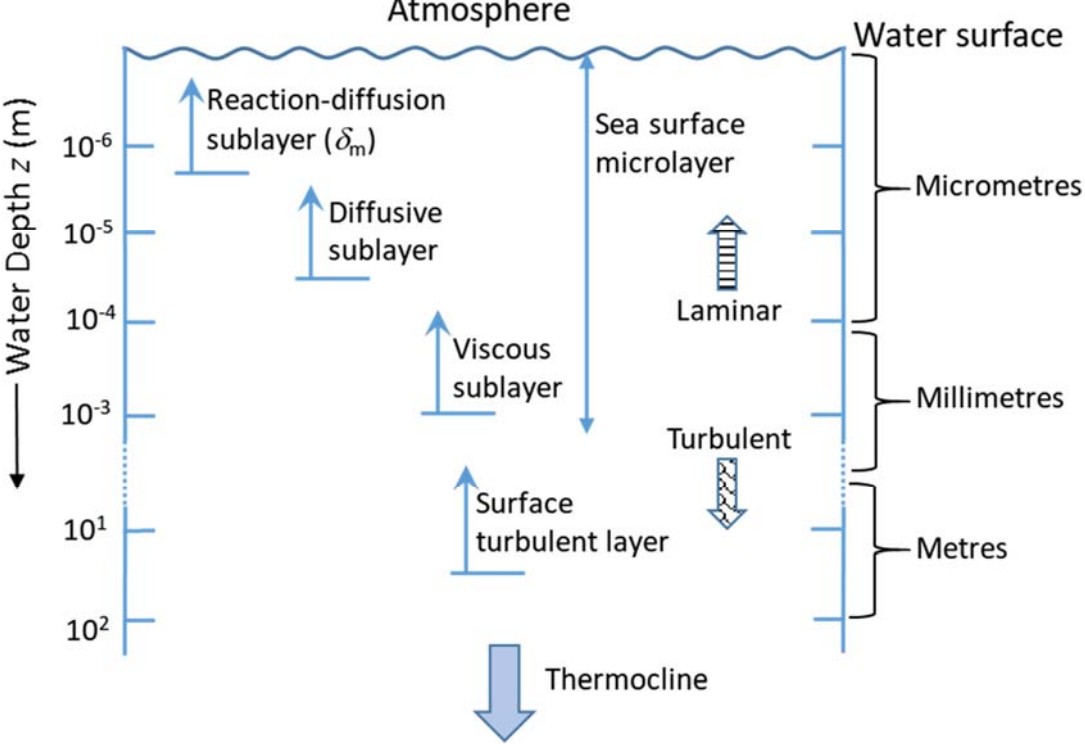

**Figure 1: Idealised representation of the vertical structure of the top few metres of sea water. The depth of the reaction-diffusion sublayer ($\delta_m$) will vary according to the chemical reactivity of the ocean water to ozone**

For ozone dry deposition to water surfaces, $r_c$ is the dominant term in Eq. (2). It is commonly assumed that $r_c$ for water is constant ($\approx 2000$ s m$^{-1}$) based on Wesely's (1989) widely used deposition parameterisation, and to our knowledge this approach is used by default in most global chemical transport models, e.g. MATCH-MPIC (von Kuhlmann et al., 2003), MESSy (Kerkweg et al., 2006), MOZART-4 (Emmons et al., 2010), CAM-chem (Lamarque et al., 2012), GEOS-Chem (Mao et al., 2013) and UKCA (Abraham et al., 2012).

Recently, Luhar et al. (2017) demonstrated that the use of a constant $r_c$ for water results in a near-constant behaviour of $v_d$ with sea surface temperature (SST) which overestimates the open-ocean deposition velocity measurements of Helmig et al. (2012) by as much as a factor of 2 to 4 for cooler SSTs. Luhar et al. (2017) also tested a mechanistic, one-layer reactivity scheme for $r_c$ proposed by Fairall et al. (2007) which includes the influence of waterside processes acting on ozone, namely solubility, molecular diffusion, turbulent transfer and a first-order chemical reaction of ozone with dissolved iodide, and they found that the one-layer scheme too overestimates the deposition velocity measurements (albeit to a slightly lesser degree than the constant $r_c$ approach) due to an overestimation of turbulent (or eddy) diffusivity within the waterside viscous sublayer. Ganzeveld et al. (2009) included the one-layer scheme in a global model and found that compared to the Wesely constant $r_c$ approach the one-layer scheme leads to only a slight reduction in the total oceanic deposition of ozone, which is consistent with the findings from the same one-layer scheme by Luhar et al (2017).

Following Fairall et al. (2007), Luhar et al. (2017) formulated a two-layer reactivity scheme for $r_c$ in which the chemical reactivity of ozone with dissolved iodide was assumed to be present only within the reaction-diffusion sublayer ($\delta_m \sim 3$ μm) with the water region below $\delta_m$ having a near-zero background chemical reactivity (through the assumption that the iodide concentration below $\delta_m$ was virtually zero). This two-layer reactivity scheme when used in a global chemistry-climate model, namely ACCESS-UKCA (Australian Community Climate and Earth System Simulator – United Kingdom Chemistry and Aerosol), was able to describe well the absolute magnitude and the sea surface temperature dependence of the deposition velocity measurements of Helmig et al. (2012) over the ocean.

Although the two-layer reactivity scheme of Luhar et al. (2017) was successful in describing the observations, its assumption that chemical reactivity is only present within a depth of water that is of the order of only a few micrometres is arbitrary given that in reality iodide is present through the depth of the oceanic surface turbulent layer ($\sim 10$–50 m) and even deeper (Chance et al., 2014). The primary reason the two-layer reactivity scheme worked well was that limiting chemical reactivity to the reaction-diffusion sublayer artificially compensated for the effects of the overestimation of turbulent diffusivity ($K_t$) (see below) in this layer, thereby effectively restricting the vertical extent of the ozone-iodide reaction and its interaction with turbulence to the scale $\delta_m$ and thereby circumventing an overestimation of $v_d$.

The overestimation of $K_t$ alluded to above in both one- and two-layer formulations results from the use of the linear parameterisation $K_t = \kappa u_{*_w} z$, where $u_{*_w}$ is waterside friction velocity, $\kappa$ is the von Karman constant (= 0.4) and $z$ is depth from the surface. This parameterisation is valid for a fully turbulent surface layer that lies beyond the viscous sublayer. For depths within the viscous sublayer, the viscous dissipation of turbulence causes the eddy diffusivity to diminish much more rapidly with decreasing $z$ than provided by the above linear relationship. A more appropriate parameterisation for $K_t$ which varies as $z^m$ in the viscous sublayer where $m = 2$–$3$ (Fairall et al., 2000) can be considered but a corresponding analytical solution for $r_c$ that includes chemical reaction, molecular diffusion and turbulent transfer has not so far been found.

The aims of the present paper are twofold. First, to formulate a new two-layer parameterisation for $r_c$ that eliminates the assumption inherent in the (old) two-layer reactivity scheme that chemical reactivity is only present within the top few microns of the water surface. Instead the new scheme makes the valid assumption that chemical reactivity is present through the depth of the oceanic mixing layer, as supported by observations. The new scheme employs a plausible assumption with regards to the extent of reaction-dominated deposition regime, and has an asymptotic behaviour that is consistent with the known limits when turbulent transfer dominates over chemical reaction and vice versa. This new scheme is incorporated into ACCESS-UKCA and the results on deposition velocity are compared with the data of Helmig et al. (2012) and other schemes.

Second, given that there are significant biases in global modelling for ozone in the lower atmosphere, one alternative to constrain ozone dry deposition budgets better is to use ozone reanalyses involving data assimilation, which are taken as a more reliable source of near-surface ozone data than that obtained by models alone. By adopting this approach, the oceanic and global dry deposition budgets of ozone are estimated by combining the gridded global reanalyses for near-surface ozone from the European MACC (Monitoring Atmospheric Composition and Climate) program and the ozone deposition velocities estimated using the new two-layer oceanic deposition scheme in ACCESS-UKCA for ten years (2003–2012). The interannual variability and uncertainty in these budgets are investigated and the latter are compared with those from other studies.

## 2 A new two-layer scheme for surface resistance $r_c$

Assuming horizontal homogeneity and stationarity, the mass conservation equation for a chemical species in water is (Geernaert et al., 1998; Fairall et al., 2007):

$$\frac{\partial}{\partial z}\left[\{D + K_t(z)\}\frac{\partial C(z)}{\partial z}\right] - a\,C(z) = 0 , \tag{3}$$

where $z$ is depth from water surface, $C(z)$ is the concentration of the species, $D$ is the molecular diffusivity of the species in water, $K_t(z)$ is the turbulent diffusivity and $a$ is a first-order reaction rate coefficient which for the ozone-iodide reaction ( $O_3 + I^- \rightarrow$ products) is determined as the pertinent second-order rate coefficient ($k$) multiplied by the iodide concentration ( $[I^-]$ ).

A flux variable $F_0$ (which we will just refer to as flux) that is invariant with water depth $z$ can be defined by integrating Eq. (3) (Fairall et al., 2007):

$$-[D + K_t(z)]\frac{\partial C(z)}{\partial z} + a\int_0^z C(z)dz = F_0 \; . \tag{4}$$

The first term on the left hand side of Eq. (4) is the mixing flux (molecular diffusion plus turbulent mixing) which decreases with depth as the reacting gas is absorbed. This component is balanced by the second term on the left hand side which is the integrated loss rate of ozone by chemical reaction between the ocean surface and depth $z$.

We now consider an alternative two-layer approach in which chemical reaction in the top water layer of depth $\delta_m$ (i.e. the reaction-diffusion sublayer that is embedded within the viscous sublayer) is fast enough such that it dominates over turbulent transfer, with the assumption $K_t = 0$, and transport is maintained by molecular diffusion (Figure 2). The thickness of this layer is thus of the order of the so-called reaction-diffusion length scale $l_m = (D/a)^{1/2}$ for the ozone-iodide reaction in seawater which is typically a few micrometres. This length scale for the said reaction is even smaller than the Kolmogorov microscale (the latter is indicative of the smallest of the turbulent eddies present in the flow) so it is fair to assume that $K_t = 0$ within the reaction-diffusion sublayer. The second layer which is deeper than the reaction-diffusion sublayer (i.e. $z > \delta_m$ ) has both chemical reaction and turbulent mixing included, and a linear parameterisation for turbulent diffusivity $K_t = \kappa u_{*w} z$ is used (Figure 2). The second layer can thus include part of the viscous sublayer and extend to the surface turbulent layer. The chemical reaction of ozone predominantly occurs in the first layer. In the second layer, turbulence-chemistry interaction is weak compared to transfer by turbulent mixing. It is therefore reasonable to use a linearly varying $K_t$ throughout the second layer. Both layers have the same reactivity $a$, i.e. the iodide concentration is uniform through the oceanic surface mixed layer. (In contrast to Figure 2, the two-layer scheme of Luhar et al. (2017) assumed $K_t = \kappa u_{*w} z$ in

both layers and $a \approx 0$ in the second layer. The one-layer scheme of Fairall et al. (2007) is equivalent to setting $K_t = \kappa u_{*_w} z$ in both layers or $\delta_m \to 0$ in Figure 2.)

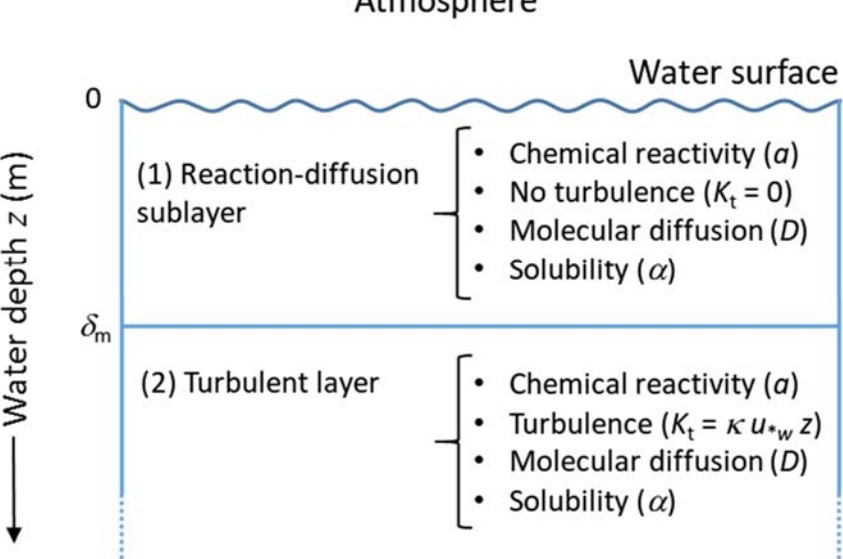

Figure 2: A simplified, two-layer structure used to represent the near-surface ocean in the model and the processes included in the calculation of ozone dry deposition to sea water.

With the above assumptions, Eq. (3) can be solved for concentration ($C_1$) in the first (i.e. top) layer and that ($C_2$) in the second (i.e. bottom) layer to yield:

$$C_1(z) = A_1 \exp\left( z\sqrt{\frac{a}{D}} \right) + B_1 \exp\left( -z\sqrt{\frac{a}{D}} \right),$$  (5)

$$C_2(z) = B_2 K_0(\xi),$$  (6)

where $\xi = [2ab(z + (bD/2))]^{1/2}$, $b = 2/(\kappa u_{*_w})$, $\kappa = 0.4$, and $K_0(\xi)$ is the modified Bessel function of the second kind of order 0.

The expressions for the mixing component (which includes both turbulent and molecular diffusion parts) of the flux $F_0$ in the first and second layers follow from the first part on the left hand side of Eq. (4) coupled with Eqs. (5) and (6):

$$F_{m1}(z) = -(aD)^{1/2}\left[A_1 \exp\left(z\sqrt{\frac{a}{D}}\right) - B_1 \exp\left(-z\sqrt{\frac{a}{D}}\right)\right], \tag{7}$$

$$F_{m2}(z) = \frac{B_2}{b}\xi K_1(\xi), \tag{8}$$

where and $K_1(\xi)$ is the modified Bessel function of the second kind of order 1.

The three unknown coefficients $A_1$, $B_1$ and $B_2$ are determined by imposing three boundary conditions. The first two, namely the flux at the water surface ($z = 0$) obtained using Eq. (4) should be equal to $F_0$ and the concentration at the interface of the two layers ($z = \delta_m$) should be continuous, lead to the following equations, respectively:

$$F_0 = F_{m1(z=0)} = (aD)^{1/2}(-A_1 + B_1), \tag{9}$$

$$A_1 \exp(\lambda) + B_1 \exp(-\lambda) - B_2 K_0(\xi_\delta) = 0, \tag{10}$$

where $\xi_\delta = [2ab(\delta_m + (bD/2))]^{1/2}$ and $\lambda = \delta_m (a/D)^{1/2}$.

The third boundary condition can be imposed in a couple of ways, both of which lead to the same answer. First, the total flux at the interface is continuous, i.e.

$$F_{m1}(\delta_m) + a\int_0^{\delta_m} C_1(z)dz = F_{m2}(\delta_m) + a\int_0^{\delta_m} C_1(z)dz, \tag{11}$$

which leads to $F_{m1}(\delta_m) = F_{m2}(\delta_m)$. This after substituting the flux Equations (7) and (8) yields

$$(aD)^{1/2}\{A_1 \exp(\lambda) - B_1 \exp(-\lambda)\} + \frac{B_2}{b}\xi_\delta K_1(\xi_\delta) = 0. \tag{12}$$

Another option as suggested by Fairall et al. (2007) is that as $z \to \infty$ the mixing term in Eq. (4) becomes 0 so $F_0$ equals the total absorption of concentration by chemical reaction, i.e.

$$F_0 = a\int_0^{\delta_m} C_1(z)dz + a\int_{\delta_m}^{\infty} C_2(z)dz. \tag{13}$$

This condition leads to exactly the same expression as Eq. (12) when $F_0$ is substituted from Eq. (9).

Solving Eqs. (9), (10) and (12) yields

$$B_1 = \frac{F_0 \exp(\lambda)}{2(a\,D)^{1/2}} \left[ \frac{\psi K_1(\xi_\delta) + K_0(\xi_\delta)}{\psi K_1(\xi_\delta)\cosh(\lambda) + K_0(\xi_\delta)\sinh(\lambda)} \right]. \tag{14}$$

Now $A_1$ and $B_2$ can be determined using Eq. (9) and (10), respectively, after substituting $B_1$ from Eq. (14). Using Eqs. (5) and (9) we can obtain an expression for the waterside deposition velocity $v_{dw}$ as the flux ($F_0$) divided by concentration ($C_0$) at

5  $z = 0$

$$v_{dw} = (a\,D)^{1/2} \left[ \frac{-A_1 + B_1}{A_1 + B_1} \right], \tag{15}$$

which after substituting for $A_1$ and $B_1$ results in

$$v_{dw} = (a\,D)^{1/2} \left[ \frac{\psi K_1(\xi_\delta)\cosh(\lambda) + K_0(\xi_\delta)\sinh(\lambda)}{\psi K_1(\xi_\delta)\sinh(\lambda) + K_0(\xi_\delta)\cosh(\lambda)} \right], \tag{16}$$

where $\psi = \xi_\delta / (a\,b^2 D)^{1/2} = [1 + (\kappa u_{*w}\delta_m / D)]^{1/2}$. Eq. (16) is the final expression for $v_{dw}$ and is used to determine $r_c$ as

$$r_c = \frac{1}{\alpha v_{dw}}, \tag{17}$$

10  where $\alpha$ is the dimensionless solubility of ozone in water (which is the ratio of the aqueous-phase ozone concentration to its gas-phase concentration and is related to Henry's law coefficient). The modified Bessel functions that appear Eq. (16) were calculated using the algorithms given in Press et al. (1997).

**2.1 Asymptotic limits**

In the limit $\delta_m \to 0$, Eq. (16) reduces to

$$v_{dw} = (a\,D)^{1/2} \left[ \frac{K_1(\xi_0)}{K_0(\xi_0)} \right], \tag{18}$$

where $\xi_0 = b(aD)^{1/2}$. This is equivalent to the one-layer model of Fairall et al. (2007) which employs a linearly varying $K_t$ with $z$ and which as mentioned in Introduction overestimates the oceanic deposition velocity measurements of Helmig et al. (2012) by as much as a factor of 2–3 for lower SSTs (Luhar et al., 2017).

In the limit $\delta_m \rightarrow \infty$, the waterside turbulent transfer is neglected and the formulation becomes equivalent to the diffusion-reaction formulation considered by Garland et al. (1980):

$$v_{dw} = (aD)^{1/2},$$
(19)

which underestimates the oceanic deposition velocity measurements for SSTs below 15°C (Luhar et al., 2017).

### 2.2 Behaviour of the new scheme and specification of $\delta_m$

The above scheme for determining the oceanic ozone deposition velocity requires specification of the dissolved iodide concentration $[I^-]$ and the second-order rate coefficient ($k$) for the ozone-iodide reaction used in the calculation of chemical reactivity via $a = k.[I^-]$, the dimensionless solubility of ozone in water ($\alpha$), the molecular diffusivity of ozone in water ($D$) and the waterside friction velocity ($u_{*w}$). We use

$$[I^-] = 1.46 \times 10^6 \exp\left(\frac{-9134}{T_s}\right)$$
(20)

from MacDonald et al. (2014) where $[I^-]$ is in mole per litre (or molar, M). This parameterisation is based on iodide data from cruises in the Atlantic and Pacific oceans covering the latitudes 50°S to 50°N, and is a function of SST ($T_s$ (K)) which varies with space and time. Eq. (20) yields highest iodide concentrations in warm tropical waters and lowest in cool waters at higher latitudes. Chance et al. (2014) examined statistical relationships between iodide and parameters such as SST, nitrate, salinity, chlorophyll-a and mixed layer depth, and found that SST was the strongest predictor of iodide in surface waters. Ganzeveld et al. (2009) used oceanic surface nitrate as a proxy for $[I^-]$.

The second-order rate coefficient $k$ (M$^{-1}$ s$^{-1}$) based on the data from Magi et al. (1997) is

$$k = \exp\left(\frac{-p}{T_s} + q\right),$$
(21)

where $p = 8772.2$ and $q = 51.5$.

The ozone solubility is (Morris, 1988)

$$\log_{10}(\alpha) = -0.25 - 0.013(T_s - 273.16).$$ (22)

The molecular diffusivity $D$ (m$^2$ s$^{-1}$) of ozone in water is given as (Johnson and Davis, 1996)

$$D = 1.1 \times 10^{-6} \exp\left(\frac{-1896}{T_s}\right).$$ (23)

The waterside friction velocity $u_{*_w}$ is calculated as $u_{*_w} = (\rho_a / \rho_w)^{1/2} u_*$ where $u_*$ is the airside friction velocity, $\rho_a$ is the air density and $\rho_w$ is the water density.

The depth of the reaction-diffusion sublayer ($\delta_m$) needs to be specified. As mentioned earlier, it is of the order of the reaction-diffusion length scale $l_m [= (D/a)^{1/2}]$ so one option is to take $\delta_m = c_0 l_m$ (in that case $\lambda = c_0$), where $c_0$ is a constant. (With the above parameterisations for $[I^-]$, $k$ and $D$, $l_m$ varies between 24.0–1.2 µm for the SST range 2–33°C, and it is 3 µm at 23°C.) Figure 3 presents the variation of the oceanic component of dry deposition velocity multiplied by the ozone solubility, i.e. $\alpha v_{dw}$ ($= 1/r_c$), calculated from Eq. (16) as a function of SST (Figure 3a) and reactivity ($a$) (Figure

3b) for three $c_0$ values for a typical value of the waterside friction velocity ($u_{*_w}$) of 0.01 m s$^{-1}$ (which corresponds to an airside $u_*$ of approximately 0.3 m s$^{-1}$). The plotted variations show that $\alpha v_{dw}$ increases with SST and with the logarithm of $a$, both in a very similar manner. As $c_0$ decreases (hence $\delta_m$ decreases) the two-layer model behaviour approaches the behaviour of the one-layer scheme given by Eq. (18) in which turbulent diffusivity is a linear function of depth and chemical reaction is included. On the other hand, as $c_0$ gets larger (hence $\delta_m$ gets larger) the extent of the reaction-diffusion regime in

the two-layer scheme gets larger and the model behaviour approaches the limiting behaviour $\alpha v_{dw} = \alpha (a D)^{1/2}$ (Eq. (19)) as originally discussed by Garland et al. (1980). In the old two-layer reactivity scheme of Luhar et al. (2017), in some cases $\alpha v_{dw}$ can go below the variation implied by the diffusion-reaction limit (19) which is not realistic and which does not occur with the new scheme.

In Figure 3b, as $a$ decreases $\delta_m$ increases (since $\delta_m = c_0 (D/a)^{1/2}$) the model approaches the diffusion-reaction limit

Eq. (19) of Garland et al. (1980), and as $a$ increases $\delta_m$ decreases and the model approaches the one-layer solution Eq. (18). Figure 3a shows the same behaviour but in terms of SST.

It is found that the use of $\delta_m = c_0\, l_m$ together with the parameterisations (20)–(23) does not fully describe the variation of the measured deposition velocities with SST (presented later) regardless of the value of $c_0$. For example, with $c_0 = 0.7$ there is an underestimation by the model of the measured deposition velocities for SSTs less than 18°C and an overestimation for higher SSTs. For $c_0 < 0.7$ the overestimation gets worse. For $c_0 > 0.7$ the underestimation gets worse

5    and the $\alpha\, v_{dw}$ variation approaches the diffusion-reaction behaviour.

Another method for specifying $\delta_m$ is to assume that it is constant. Figure 3 shows the variation of $\alpha\, v_{dw}$ calculated from Eq. (16) as a function of SST (Figure 3c) and $a$ (Figure 3d) for several fixed values of $\delta_m$ between 0.5 and 10 µm. These variations look different compared to those in Figure 3a and Figure 3b but like the latter they all fall within the two limits. As $\delta_m$ decreases the $\alpha\, v_{dw}$ variation approaches the one-layer solution Eq. (18) and as $\delta_m$ increases this variation

10   approaches the diffusion-reaction limit Eq. (19).

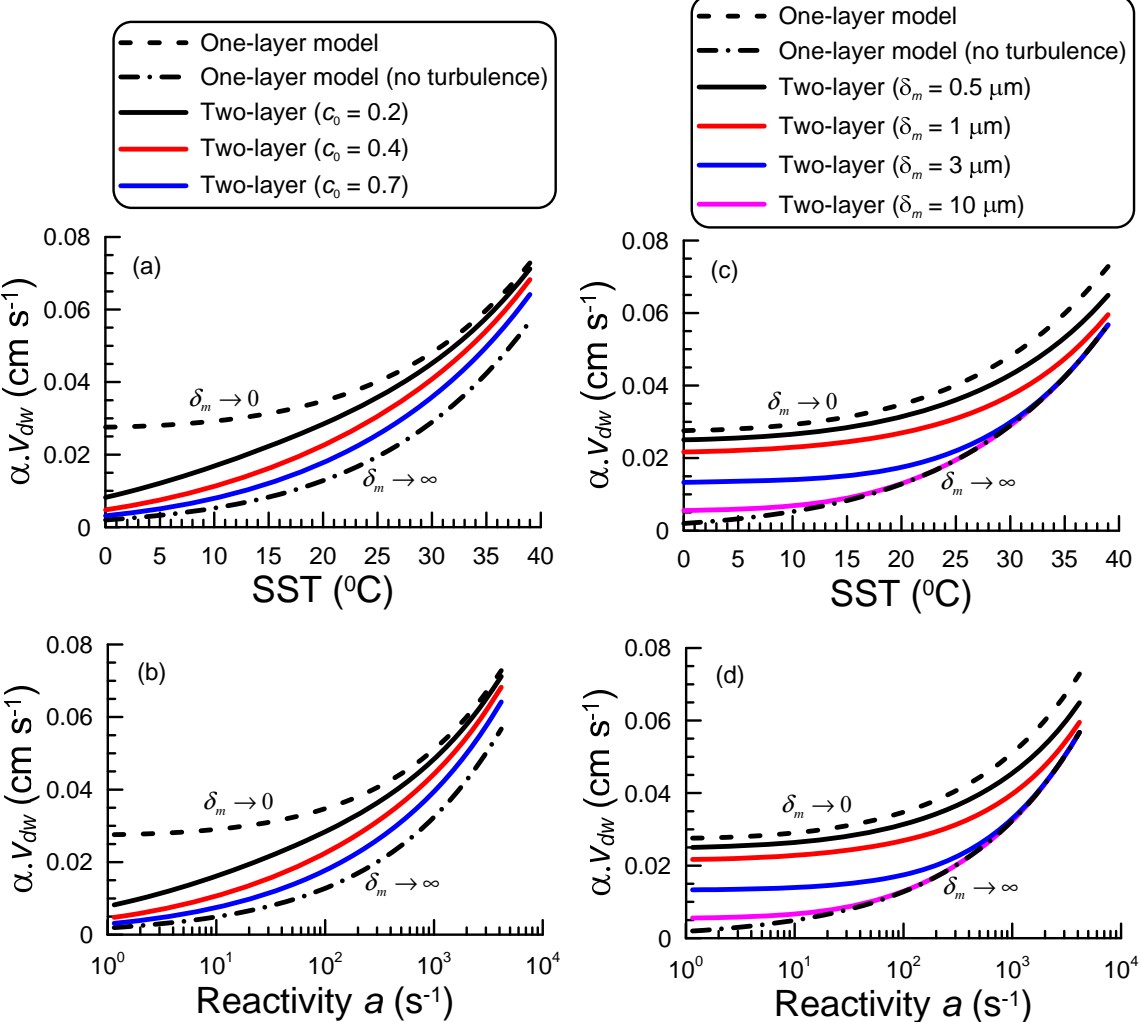

**Figure 3: Variation of the waterside component of ozone dry deposition velocity multiplied by ozone solubility, $\alpha v_{dw}$ (= $1/r_c$), as a function of sea surface temperature (SST, °C), (a, c); and reactivity $a$ (s$^{-1}$), (b, d). Curves determined using the two-layer deposition scheme (Eq. (16)) for several $c_0$ values used in $\delta_m = c_0 l_m$, (a, b); and several $\delta_m$ values, (c, d). The variations obtained using the one-layer deposition scheme with (Eq. (18)) and without (Eq. (19)) waterside turbulent transfer (i.e. reaction-diffusion only) are also shown. The waterside friction velocity ($u_{*w}$) used was 0.01 m s$^{-1}$.**

There are further considerations to the parameterisations. There is uncertainty in the parameterisations (20)–(23), particularly in the second-order rate coefficient $k$ for which there is a paucity of data. The expression (21) is based on the data from Magi et al. (1997) which are plotted in Figure 4 with the associated uncertainty. Also, plotted are the single data points from Garland et al. (1980), Hu et al. (1995) and Liu et al. (2001). Clearly there is a large scatter in the data. Five options are

considered (option 6 is discussed later) with regards to parameterising $k$ via an exponential fit of the form Eq. (21): 1) only consider the data of Magi et al. (1997) (so the fit is the same as Eq. (21) with stated $p$ and $q$ values); 2) consider all the data (which gives $p = 2349.2$ and $q = 29.2$ ); 3) consider all the data except the data point of Hu et al. (1995) which is treated as an outlier (which gives $p = 5632.9$ and $q = 40.3$ ); 4) assume a constant $\delta_m = 3$ μm with $k$ given by Eq. (21) using only the data of Magi et al. (1997); and 5) assume a constant $k = 2 \times 10^9$ M$^{-1}$ s$^{-1}$ as in MacDonald et al. (2014). In Figure 3a and Figure 3b, all five curves fall within the two asymptotic limits (equivalent to $c_0 \to 0$ and $c_0 \to \infty$ ). The $c_0 = 0.4$ curve roughly lies in the middle of the two asymptotic limits and this value of $c_0$ was used in calculating $\delta_m = c_0 \, l_m$ in all the above options, except option (4) which does not need a specification of $c_0$ (but there will obviously be an implied variation of $c_0$ through the relation $c_0 = \delta_m / l_m$). Figure 5 shows the variation of $\alpha \, v_{dw}$ calculated from Eq. (16) as a function of SST for the above five options. All five options provide qualitatively similar variations of $\alpha \, v_{dw}$ with SST, but when compared with the cruise measurements of oceanic deposition velocities ( $v_d$ ) discussed later (which themselves have substantial scatter) in Section 4.1, options (3) and (4) provide better agreement overall with the measurements compared to the other options (noting that $v_d$ is dominated by the term $\alpha \, v_{dw}$ for water). Option (3) tends to underestimate the observed deposition velocities by roughly 15% for SSTs less than around 12°C whereas option (4) tends to overestimate them by about the same degree. For higher SSTs, both options perform similarly, with option (4) being very slightly better for SSTs greater than 20°C, within the scatter of the measurements.

We also include in Figure 5 an additional curve as option 6 which is the same as option 4 but using the Chance et al. (2014) parameterisation for iodide concentration (in molar)

$$[I^-] = [0.225 \, (T_s - 273.16)^2 + 19] \times 10^{-9}. \tag{24}$$

Compared to option 4, option 6 results in larger $\alpha \, v_{dw}$ values and the relative difference between the two increases with SST; for example, for SSTs 5, 20 and 30°C, the option 6 value is larger by 13, 29 and 33%, respectively. Consequently, option 6 would overestimate the observed ozone deposition velocity data presented later in Figure 7, almost passing along the upper limits of the observed fluctuations in $v_d$ . However, if option 6 is used along with the second-order rate constant ($k$) without considering the data point of Hu et al. (1995) as in option 3 (which gives lower $k$ values), then the values of deposition velocity obtained are comparable to those obtained using option 4 for the same $\delta_m$ . This is because a larger

iodide concentration is compensated by a lower $k$ in the expression for reactivity $a(=k.[I^-])$ which is what goes directly into the deposition velocity calculation.

Clearly there is significant uncertainty in the representation of iodide concentration $[I^-]$ and $k$, and certain choices of the input parameterisations can describe deposition velocity observations better than others. Overall, the calculation of ozone deposition velocity using the model presented here is a rather poorly constrained problem where multiple choices of input parameters can give the same or very similar calculated deposition velocity. The best constrained at this stage is the comparison of the calculated deposition velocity with that observed.

In the calculations below, unless otherwise stated, we have used option 4 for $\delta_m$ and $k$. In Section 4.1, this option 4 is used for calculating $v_d$ in ACCESS-UKCA and comparing the modelled $v_d$ values with measurements and with other deposition schemes/configurations. In Section 5.3 we estimate a measure of uncertainty in our deposition flux estimates taking into account the scatter in the ocean deposition velocity data used.

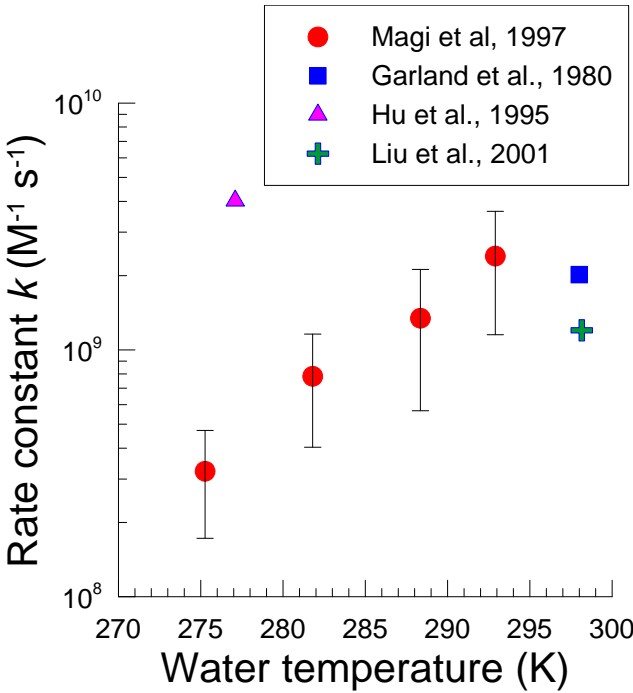

**Figure 4: The second-order rate coefficient ($k$) for the ozone-iodide reaction as a function of water temperature. Data from various studies are shown.**

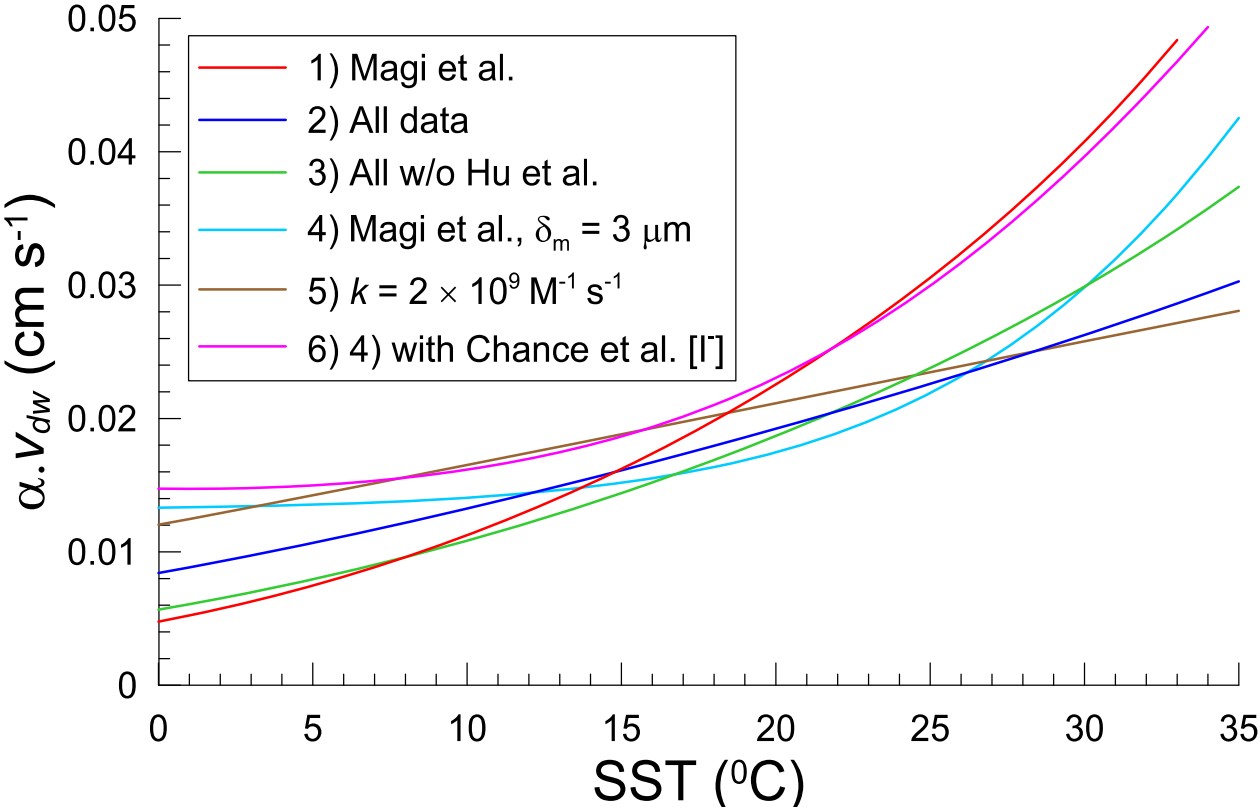

**Figure 5: Variation of the oceanic component of ozone dry deposition velocity multiplied by ozone solubility,** $\alpha\,v_{dw}$ **(= $1/r_c$ ), as a function of sea surface temperature (SST, °C). Curves determined using the two-layer deposition scheme (Eq. (16)) for various options for parameterising the second-order rate coefficient ($k$) (see text). The waterside friction velocity ( $u_{*_w}$ ) used was 0.01 m s⁻¹.**

## 2.3 Impact of ozone reaction with dissolved organic carbon (DOC)

Some studies have considered the impact on dry deposition of ozone reaction with dissolved compounds other than iodide. In general, the inclusion of additional reactions in the deposition mechanism enhances the ozone loss to the ocean and thus increases deposition velocities. Chang et al. (2004) included reactions of ozone with iodide, dimethyl sulfide (DMS), ethene and propene and showed that the reaction with iodide was by far the fastest (hence most important) in most cases. In their global modelling, Ganzeveld et al. (2009) included ozone reaction with chlorophyll-a as a first order approximation to examine the possible role of dissolved organic matter (DOM), and found that this reaction significantly increased dry deposition velocities at coastal sites (with mixed results compared to observations) and yielded only small changes to deposition velocity for open ocean sites. Sarwar et al. (2016) included ozone reactions with iodide, dissolved organic carbon (DOC) (a measure of DOM), DMS and bromide in their ozone modelling for summer months in the Northern Hemisphere,

and found that the impact of DOC on the simulated deposition velocity was comparable to that of iodide, with the other reactions contributing much less. Coleman et al. (2010) showed that in addition to iodide the inclusion of DOC in their empirical scheme described daytime deposition observations better in coastal waters of North Atlantic. We are not aware of any previous studies that have compared modelled deposition velocities involving the impact of the aqueous O$_3$-DOC reaction with the open-ocean observations of Helmig et al. (2012) and their dependence on SST. It is instructive to carry out a simple sensitivity analysis involving ozone reaction with DOC. (Our new two-layer parameterisation is applicable to any other chemical compounds that are taken up by the oceanic mixing layer as long as the reaction-diffusion length scale is smaller than the depth of the viscous sublayer.)

For open ocean surface waters, Hansell et al. (2009) report DOC concentration values of 70–80 µM in tropical and subtropical regions (40°N to 40°S), ~ 40–50 µM in subpolar seas and in the circumpolar Southern Ocean (> 50°S), and about 70 µM in the Arctic Ocean (> 70°N). Based on Hansell et al. (2009), Sarwar et al. (2016) used a mean DOC concentration of 67 µM over the Northern Hemisphere. A recent analysis by Massicotte et al. (2017) gives an average DOC value of 52 µM for oceans.

There are no definitive, directly measured values available for the second-order rate coefficient ($k$) for the DOC-O$_3$ reaction. Coleman et al. (2010) empirically derived $k = 3.44 \times 10^6$ M$^{-1}$ s$^{-1}$ based on data fitting, whereas Sarwar et al. (2016) used $k = 4.0 \times 10^6$ M$^{-1}$ s$^{-1}$ noting that this value together with their selected DOC concentration yields a first order rate constant of ~ 268 s$^{-1}$ that lies between the two values 100 s$^{-1}$ (open-ocean) and 500 s$^{-1}$ (coastal waters) used by Carpenter et al. (2013) based on the modelling by Ganzeveld et al. (2009). This reactivity value lies within the range of reactivity plotted in Figure 3.

Clearly there is considerable uncertainty in $k$ for the DOC-O$_3$ reaction, and in the DOC concentration and its variability. Dependencies of $k$, such as how it may vary with SST, are not known. For our purposes, we use a mean $k = 3.7 \times 10^6$ M$^{-1}$ s$^{-1}$ (which lies is in the middle of the two values noted above) and a DOC concentration of 52 µM (Massicotte et al., 2017) in our two-layer scheme, together with an integrated chemical loss rate $a = \sum k_i C_i$ , where the summation is over the iodide and DOC reactions with ozone ($i = 1, 2$).

Figure 6 shows the variation of $\alpha v_{dw}$ as a function of SST determined using our two-layer deposition scheme incorporating the ozone reaction with: (1) only iodide (this curve is the same as option 4 in Figure 5 with $\delta_m = 3$ µm), (2) only DOC, and (3) the two reactions together for three values of $\delta_m$. Compared to the iodide-only curve, the inclusion of DOC leads to a progressive increase in $\alpha v_{dw}$ as SST decreases for all values of $\delta_m$. As $\delta_m$ increases $\alpha v_{dw}$ decreases,

but increasing $\delta_m$ beyond 6 μm has virtually has no impact on $\alpha v_{dw}$ (not shown). When only DOC is considered, $\alpha v_{dw}$ decreases with SST. Given that the $\alpha v_{dw}$ term dominates in the determination of $v_d$, the behaviour of the former represents that of the latter. In Section 4.1 below, we show that the addition of DOC in the deposition scheme deteriorates the model-data agreement for deposition velocity. (We note that for coastal waters, not explicitly investigated here (see Section 4.2), the case may be different.) Lowering the DOC concentration in the model to the lowest levels (~ 40 μM) reported by Hansell et al. (2009) does not improve the agreement either. A $k$ value for the DOC-O$_3$ reaction that decreases with SST (like that for the iodide-O$_3$ reaction) could explain the deposition velocity data better but any such $k$ observations are lacking at present.

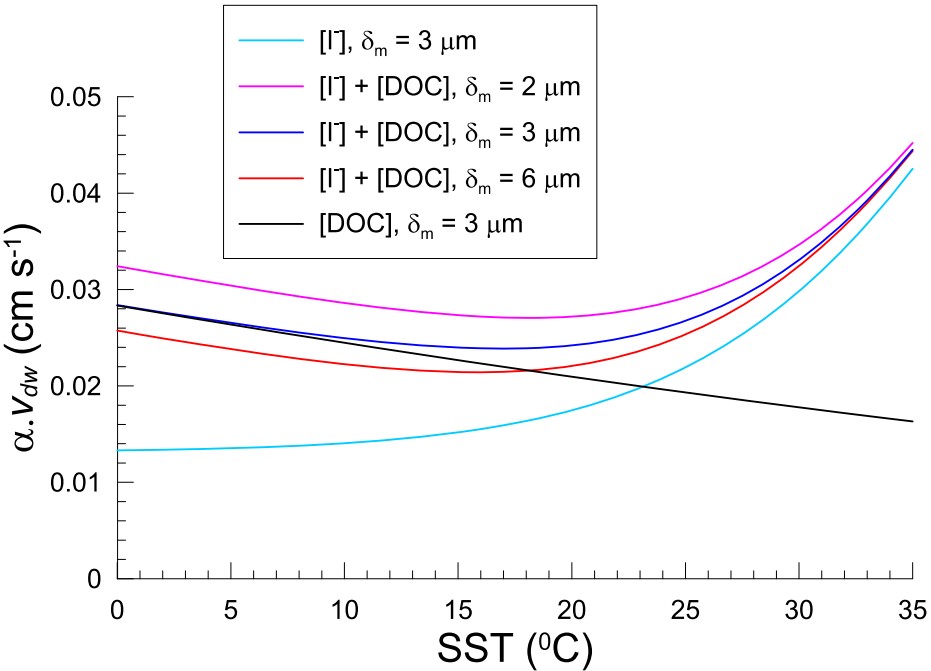

**Figure 6: Variation of the oceanic component of ozone dry deposition velocity multiplied by ozone solubility, $\alpha v_{dw}$ (= $1/r_c$), as a function of sea surface temperature. Curves determined using the present two-layer deposition scheme incorporating the ozone reaction with iodide [I-], dissolved organic carbon [DOC], and the two reactions together ([I-] + [DOC]). The waterside friction velocity ($u_{*w}$) used was 0.01 m s⁻¹.**

## 3 ACCESS-UKCA chemistry-climate modelling system

The two-layer dry deposition scheme developed above was implemented in the UKCA (http://www.ukca.ac.uk) global atmospheric composition model (Morgenstern et al., 2009; Abraham et al., 2012; O'Connor, 2014) which is a component in ACCESS (Bi et al., 2013; Woodhouse et al., 2015). The physical atmosphere component of ACCESS-UKCA is the same as

the UK Met Office's Unified Model (UM) (at UM vn8.4; Walters et al., 2014). In our simulations, ACCESS-UKCA is essentially the same as UM-UKCA since the ACCESS specific ocean and land-surface components are not invoked. This is because we run the model in atmosphere-only mode with prescribed SSTs, and the UM's original land-surface scheme (JULES) is used. The particular UKCA configuration used here (at UM vn8.4) is the so-called Chemistry of the Stratosphere and Troposphere (CheST). ACCESS-UKCA uses the monthly-mean sea surface temperature and sea ice fields prescribed from the Atmospheric Model Intercomparison project (AMIP). The atmospheric model has a horizontal resolution of 1.875° in longitude and 1.25° in latitude, and 85 levels extending from the surface to approximately 85 km (the N96L85 configuration). The model was nudged to the ERA-Interim meteorological reanalyses (Dee et al., 2011), given on pressure levels, for the horizontal wind and potential temperature in the free troposphere (Uhe and Thatcher, 2015). Other model setup details are as in Woodhouse et al. (2015).

There are nine surface types in the model, namely broad-leaf trees, needle-leaf trees, C3 grass, C4 grass, shrub, urban, water, bare soil, and land ice. For every surface grid box the three resistances $r_a$, $r_b$ and $r_c$ are calculated for each surface type and a corresponding $v_d$ is then computed. For the water surface, standard expressions for $r_a$ and $r_b$ are used by ACCESS-UKCA in Eq. (2) (see Abraham et al., 2012; Luhar et al., 2017) and $r_c$ is computed using Eq. (17) together with Eq. (16). Currently there is only one water surface type in the model, so the same deposition scheme is used for both seawater and freshwater. The SSTs prescribed in the model for every grid box vary with time and are used in the input parameterisations (20)–(24). A grid-box mean deposition velocity and the corresponding loss rate are calculated using the individual deposition velocities weighted by the fractions of the surface types present in the grid box and this loss rate is applied to the lowest model grid box in the species mass conservation equation. For a coastal grid box that also includes fractions of non-water surfaces, we use the two-layer deposition scheme when the fraction of the water surface in the grid box is greater than 60%. In all other cases the default Wesely (1989) scheme for $v_d$ is used, involving the use of $r_c$ = 2200 s m$^{-1}$ for the water surface.

The ozone dry deposition velocity in the model is solely a function of parameters of the physical component of the model (e.g., SST, flow properties and turbulent mixing, and surface characteristics) and prescribed input parameters (e.g., reactivity, ozone molecular diffusivity and solubility in water), and is unrelated to the tropospheric ozone chemistry within the model.

Ozone dry deposition budgets can be better constrained by using tropospheric ozone reanalyses which are taken as a more reliable source of ozone data than those obtained by models alone. In section 5, we follow this approach, in which the gridded 3-hourly MACC global reanalyses of near-surface ozone for the period 2003–2012 (Inness et al., 2013; http://apps.ecmwf.int/datasets/data/macc-reanalysis) are multiplied by the gridded 3-hourly dry deposition velocities

obtained from ACCESS-UKCA to calculate ozone deposition flux (and hence the annual deposition loss). Because we use the MACC ozone the derived deposition fluxes do not depend on ACCESS-UKCA's ozone chemistry.

## 4 Ozone dry deposition velocity to the ocean

### 4.1 Comparison with observations

We use the ozone dry deposition velocity measurements of Helmig et al. (2012) taken over the open ocean from a ship-based system during 2006–2008 which spanned 45°N to 50°S Additional details of these data are given by Bariteau et al. (2010). Surface based ozone flux stations employing the eddy-covariance technique enables a direct measurement of ozone dry deposition velocity. The data of Helmig et al. (2012) are the only such measurements available to date over the open ocean. These authors also summarise deposition velocity measurements reported in earlier studies, which are very sparse and none

of these studies involved a surface-based eddy-covariance technique over the open ocean (there were a few data points for coastal locations and from aircraft-based systems using such a technique). Given the substantially larger sample size for a range of SST, and the (perceived) use of improved instrumentation and analysis techniques in the cruise measurements of Helmig et al. (2012) compared to those reported by earlier studies, we only consider the cruise data.

The ship-based experiments were conducted on five cruises, namely: (1) TexAQS06 (July 7 to September 12, 2006), (2)
STRATUS06 (October 9–27, 2006), (3) GOMECC07 (July 11 to August 4, 2007), (4) GasEx08 (February 29 to April 11, 2008), and (5) AMMA08 cruises (April 27 to May 18, 2008). The respective areas covered were: (1) North-western Gulf of Mexico, (2) the persistent stratus cloud region off Chile in the eastern Pacific Ocean, (3) the Gulf of Mexico and the US east coast, (4) the Southern Ocean, and (5) the southern and northern Atlantic Ocean. Helmig et al. (2012) present bin-averaged deposition velocity data as a function of SST and wind speed for each of the five cruises. As in Luhar et al. (2017), the $v_d$
versus SST cruise data used for comparison with the model are those with the wind speed dependence retained (Ludovic Bariteau, personal communication, 2016) and not the data originally reported by Helmig et al. (2012) in which the wind-speed dependence was removed. While this approach is logically correct, there is not a large difference between the data with and without the wind-speed dependence.

ACCESS-UKCA output including dry deposition parameters is available at 3-h time intervals and also as monthly averages
over the period 2003–2012. Because the data are averaged with respect to SST or wind-speed bins for each cruise and as a result there is no explicit dependence present as to the exact timings and locations of the data along a cruise track, we used the same methodology as that in Luhar et al. (2017) for comparing the $v_d$ data with ACCESS-UKCA. In summary, as the months corresponding to the cruise experiments are known, the model monthly averages matching the experimental months were selected. For a given month, the monthly-averaged model output was extracted at a series of grid-box locations (fully
covered by water) with almost uniform spacing along the tracks of the experimental cruises, and the modelled values at these

locations were used for comparison with the measurements. Clearly this is an approximate matching of the deposition velocity data and the modelled values in terms of time and location.

Figure 7 shows the observed ozone dry deposition velocity ($v_d$) as a function of SST from the five field experiments and the corresponding values obtained from the ACCESS-UKCA model using the new two-layer scheme (Eq. (16)). The SST range for the measurements is 2–33 °C with the lowest values being for GasEx08 and the highest for TexAQS06 and GOMECC07. Despite the large fluctuations within the field data, an increasing trend of $v_d$ with SST is clearly identifiable. Helmig et al. (2012) compiled a historical record of ozone deposition velocities over water (their Figure 4) starting from 1969 which lie within the range 0.01–0.15 cm s$^{-1}$. The range of the cruise measurements in Figure 7, which are the only direct, open-ocean flux measurements, is 0.005 – 0.06 cm s$^{-1}$ which is on the lower end of the range of the historical data. As stated by Helmig et al. (2012), the earlier experiments, lacking ocean-deployable measurement techniques, are biased toward coastal waters which may carry higher concentrations of ozone reactants that lead to increased deposition velocities. Another reason for the difference could be the use of improved experimental techniques in the cruise measurements.

Figure 8 presents a scatter plot of the observed deposition velocities averaged over the data from each of the five cruise experiments versus the corresponding values obtained from the model (with the error bars representing one standard deviation variation). The correlation between the modelled $v_d$ and data is good ($r^2 = 0.86$). Although Figure 7 and Figure 8 show that the model is able to describe the sea surface temperature dependence and the absolute magnitude of the field measurements, it is clear that there are some significant fluctuations in the measurements, particularly for SSTs within the range 8–24 °C, that are not present as prominently in the modelled values. There could be a number of possible reasons for this: 1) the monthly-averaged modelled deposition values used and the approximate method followed for matching the data and for time and location; 2) the dissolved iodide concentrations are not directly available and the parameterisation used here only depends on SST; and 3) the observed SSTs used in our atmosphere-only model set up are monthly averaged—a model setup with a coupled ocean model that interacts with the atmosphere at sub-diurnal intervals would provide a better SST variability which would in turn influence the variability in the iodide concentration and thus impact the modelled deposition velocity. Needless to say, additional measurements of ozone dry deposition velocity and governing parameters (e.g. iodide concentrations, SST, DOC, nitrate etc.) with greater temporal and spatial coverage would help to further assess the scheme.

The model results in Figure 7 are very similar to those obtained by Luhar et al. (2017) using their two-layer reactivity scheme (their Figure 6), but unlike the old two-layer scheme the new two-layer scheme performs well for the right reasons—as discussed earlier the old scheme artificially limits chemical reactivity to the reaction-diffusion sublayer in order to compensate for the overestimation of the impact of waterside turbulence due to a turbulent diffusivity parameterisation that is not appropriate very close to the water surface.

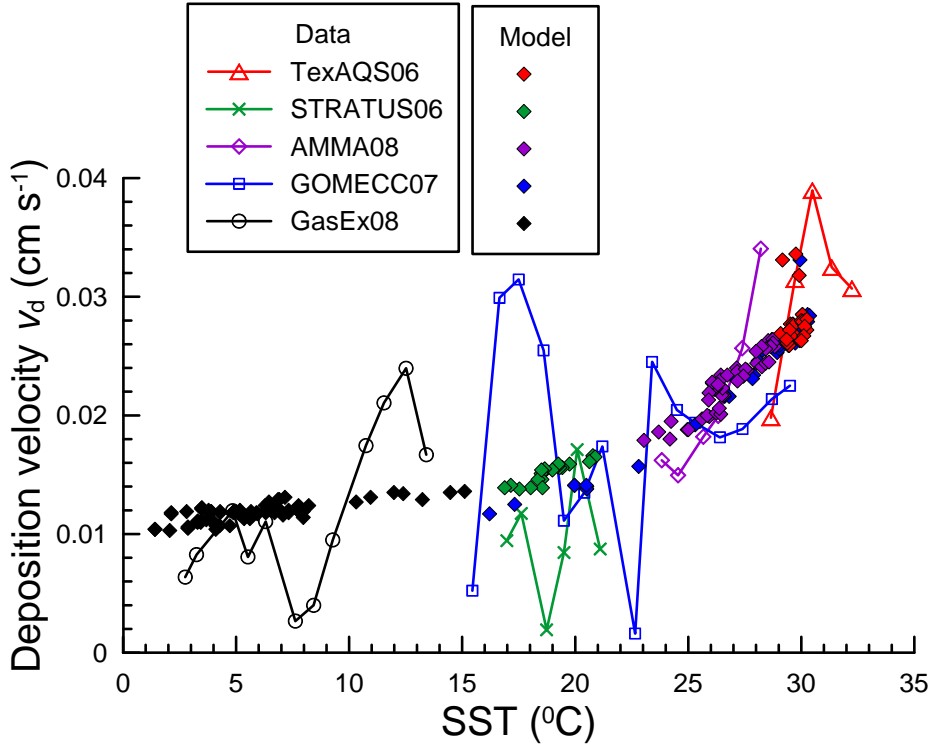

**Figure 7: Ozone dry deposition velocity ($v_d$) as a function of sea surface temperature (SST) from five field experiments (Helmig et al., 2012; Ludovic Bariteau, personal communication, 2016) and the corresponding values obtained from the ACCESS-UKCA model using the new two-layer scheme (Eq. (16)) for ozone deposition to the ocean.**

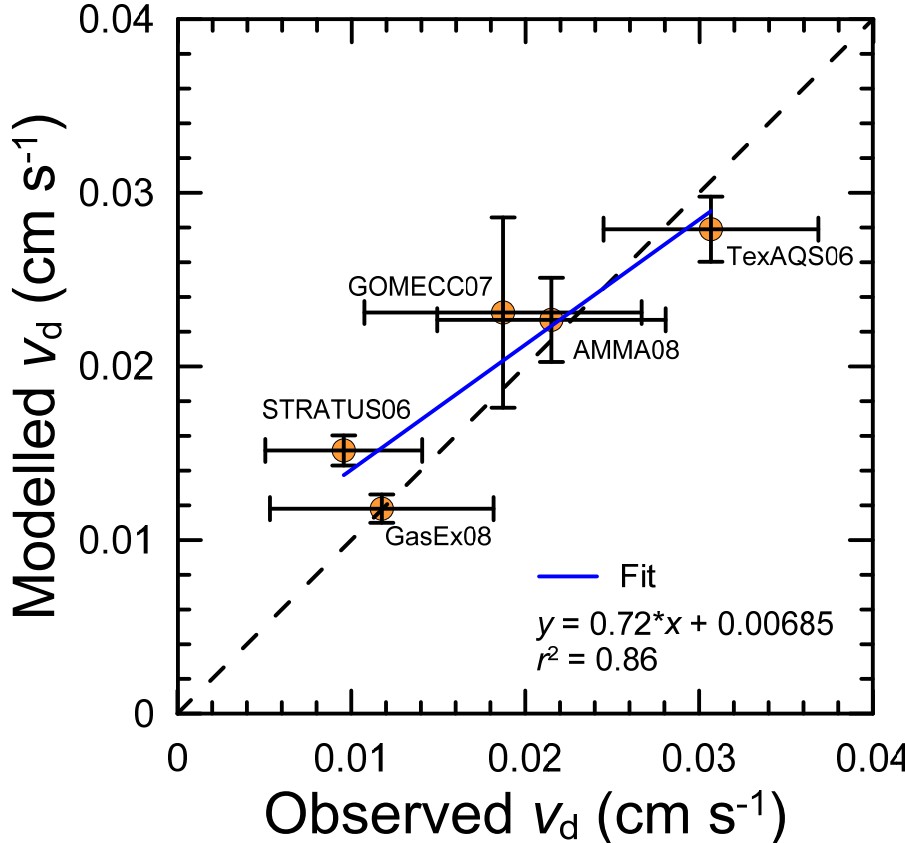

**Figure 8: Scatter plot of the ozone dry deposition velocities (*v$_d$*) obtained from the five cruise experiments versus the corresponding values obtained from the ACCESS-UKCA model using the new two-layer scheme (Eq. (16)). Each point corresponds to the average over all values from one experiment. The error bars represent one standard deviation variation. Horizontal error bars are for the observed values and the vertical ones are for the modelled values.**

The performance with ranking of the various ozone dry deposition velocity schemes/configurations (within ACCESS-UKCA) for seawater discussed above compared to the $v_d$ observations shown in Figure 7 is summarised in Table 1 in terms of some commonly used statistical measures, which were calculated using the bin-averaged modelled and observed $v_d$ values within SST bins of 5°C lying within the range 1–35°C (sample size = 7). The measures used are the ratio of modelled mean to observed mean ($\overline{M} / \overline{O}$), fractional bias (*FB,* varies between +2 (underestimation) and -2 (overestimation)), normalised mean square error (*NMSE*), and index of agreement (*IOA,* varies between 0 and 1). For a perfect agreement, the values of these parameters should be 1, 0, 0 and 1, respectively. Unlike the correlation coefficient (*r*), IOA is sensitive to differences between the observed and model means as well as to certain changes in proportionality, and is therefore preferred (Willmott, 1981).

In the schemes in Table 1, unless stated otherwise, the second-order rate coefficient is given by Eq. (21) using only the data of Magi et al. (1997), and the MacDonald et al. (2014) iodide parameterisation (20) is used. In Table 1, the new two-layer scheme with $\delta_m$ = 3 µm (as used in the model-data comparison plots above and in all the calculations below) performs the best, followed by the same scheme with the Chance et al. (2014) iodide parameterisation (24) which overestimates the $v_d$

5    data. The next two schemes in the ranking are the one-layer scheme of Fairall et al. (2007) without waterside turbulence, Eq. (19), which is equivalent to the diffusion-reaction formulation considered by Garland et al. (1980)) and which underestimates the deposition velocity data, and the two-layer scheme that also includes the DOC-$O_3$ reaction which overestimates the data. The largest overestimation of the $v_d$ data is through the use of constant $r_c$ = 2200 s m$^{-1}$ and to a slightly lesser extent by the one-layer scheme of Fairall et al. (2007), Eq.(18). The above model-data comparison suggests

10    that the two-layer scheme with the soundly based iodide mechanism is able to describe well the deposition velocity measurements for the open ocean and we use this setup hereafter.

**Table 1: Performance statistics of various ozone dry deposition velocity schemes for seawater compared to the observations shown in Figure 7[1].**

| Scheme | $\overline{M}$ / $\overline{O}$ | FB | NMSE | IOA | Ranking |
|---|---|---|---|---|---|
| Present two-layer scheme | 0.99 | 0.01 | 0.06 | 0.92 | 1 |
| Present two-layer scheme with Chance et al. (2014) $[I^-]$ | 1.20 | -0.19 | 0.09 | 0.88 | 2 |
| Present two-layer scheme with DOC | 1.30 | -0.26 | 0.14 | 0.73 | 4 |
| One-layer scheme of Fairall et al. (2007) | 1.63 | -0.48 | 0.29 | 0.65 | 5 |
| One-layer scheme of Fairall et al. (2007) without waterside turbulence | 0.73 | 0.32 | 0.18 | 0.87 | 3 |
| Wesely's (1989) scheme with $r_c$ = 2200 s m$^{-1}$ | 1.90 | -0.62 | 0.51 | 0.45 | 6 |

[1]Observed mean ($\overline{O}$), modelled mean ($\overline{M}$), fractional bias (FB) = $2(\overline{O} - \overline{M}) / (\overline{O} + \overline{M})$, normalised mean square error

(NMSE) = $\overline{(M - O)^2} / (\overline{M}.\overline{O})$, and index of agreement (IOA) = $1 - [\overline{(M-O)^2} / \overline{(|M-\overline{O}|+|O-\overline{O}|)^2}]$.

**4.2 Global distribution**

10    Figure 9 shows the distribution of ozone deposition velocity (cm s$^{-1}$) to the ocean (not including sea ice) obtained using the new two-layer scheme within ACCESS-UKCA for the year 2005. The year 2005 is chosen to illustrate the spatial variability because, as will be discussed later, the MACC ozone reanalysis has the least bias for this year (however, we note that the interannual variability of the modelled deposition velocity fields is small). The largest open-ocean deposition velocities occur in the tropics where both the observed (Chance et al., 2014; MacDonald et al., 2014) and parameterised iodide

15    concentrations, which are proportional to SST, are the largest, and the magnitude of deposition velocities decreases with increasing latitude.

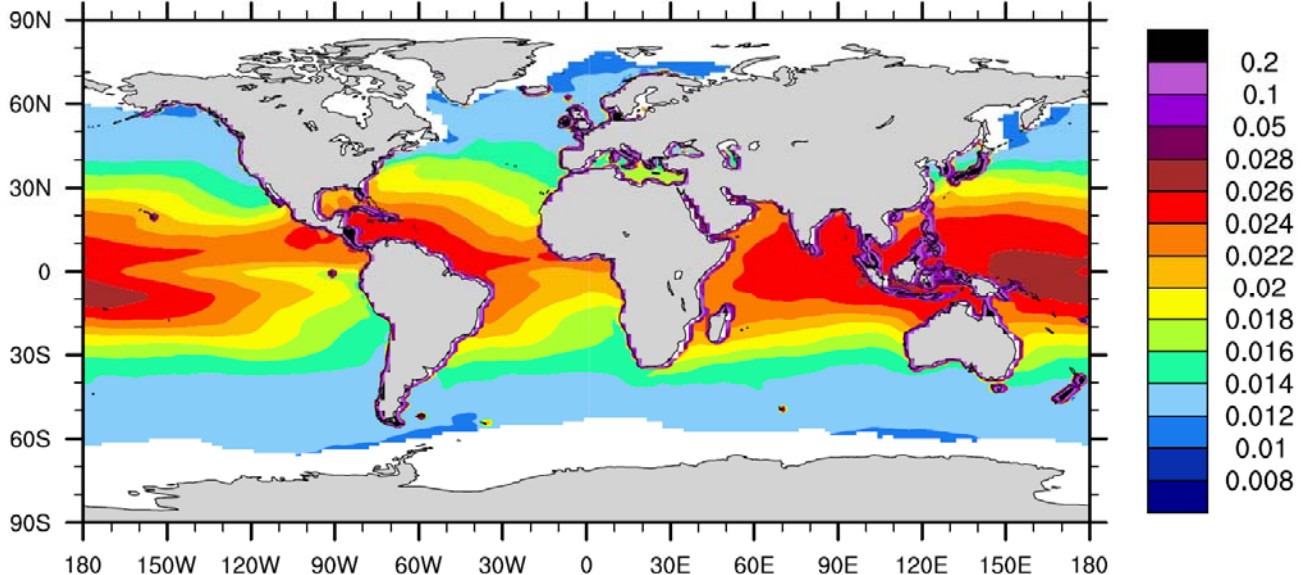

**Figure 9: Annual mean ozone dry deposition velocity ($v_d$, cm s$^{-1}$) to the ocean for the year 2005 obtained from the ACCESS-UKCA model incorporating with the oceanic dry deposition scheme proposed in this paper. Deposition velocities over the coastal grid boxes are also shown.**

Figure 9 also shows the modelled deposition velocities over the coastal grid boxes that contain some fractions of both water and land surfaces. For such grid boxes, the modelled deposition velocities are typically 0.1–0.2 cm s$^{-1}$ which are much greater than those for grid boxes fully covered by water. This is because terrestrial surfaces have higher deposition velocities than water and also partly because of the use of the larger deposition rate through the use of $r_c$ = 2200 s m$^{-1}$ for the water surface tile when its fraction is less than 60% within a grid cell. There is some evidence that the measured ozone deposition velocities over coastal waters are larger than those over open oceans (e.g. Coleman et al., 2010; Bariteau et al., 2010), which could be due to factors such as stronger chemical reactivity and turbulence, and advection from land if the distance between the monitor and coastline (i.e. fetch) is limited. Our approach for treating coastal water grid boxes is qualitatively consistent with ozone deposition velocities over coastal waters being larger than over the open sea, but here we have not examined or included any particular mechanistic processes that are relevant for coastal waters. High resolution, regional- or small scale-modelling could be useful in exploring such processes and their spatial and temporal scale and variability.

## 5 Dry deposition budgets using the MACC ozone reanalysis

Ozone deposition fluxes are calculated from ozone concentrations in near-surface air and associated ozone deposition velocities. In this study, the MACC near-surface ozone data are used.

The global model used for deriving the MACC reanalysis consists of the European Centre for Medium-Range Weather
Forecasts' (ECMWF) Integrated Forecast System (IFS) coupled to the MOZART (Model for OZone And Related chemical Tracers) chemistry transport model (Kinnison et al., 2007). The modelling system makes use of four-dimensional variational data assimilation to combine satellite retrievals of carbon monoxide, ozone, nitrogen oxides as well as the standard meteorological observations with the numerical model in order to produce a reanalysis of atmospheric composition. For ozone, profile, total column and partial column data are assimilated.

The MACC reanalysis has been evaluated against multiple observational networks of ground-based measurements, ozonesondes, and aircraft and satellite data (Inness et al., 2013; Gaudel et al., 2015; Giordano et al., 2015; Katragkou et al., 2015; [http://macc.copernicus-atmosphere.eu/documents/maccii/deliverables/val/MACCII_VAL_DEL_D_83.6_REAreport04_20140729.pdf](http://macc.copernicus-atmosphere.eu/documents/maccii/deliverables/val/MACCII_VAL_DEL_D_83.6_REAreport04_20140729.pdf)). These evaluation studies suggest that the assimilation of composition data generally improves the modelled tropospheric ozone
fields, noting that there are some exceptions which highlight the fact that assimilation does not always yield a close match with observations and that the results depend on several factors such as the quality and quantity of data being assimilated, and the type of modelling system and the data assimilation methodology used.

The MACC composition reanalysis is given at 60 hybrid sigma-pressure levels, from near the surface (1012 hPa, 10 m Geometric Altitude) to 0.1 hPa ($\sim$ 65.6 km) covering both the troposphere and the stratosphere. The MACC ozone data at the
10-m level (L60) were extracted at a horizontal resolution of $1.125° \times 1.125°$ at 3-h time intervals, and were re-gridded to the ACCESS-UKCA N96 horizontal grid using bilinear interpolation. These data were then multiplied by the time-matched 3-h deposition velocity fields obtained from ACCESS-UKCA (with the new two-layer ocean deposition scheme) to calculate the deposition flux and total deposition loss. The use of a 3-hourly temporal resolution, which is the finest available for the MACC reanalysis, ensures that any (e.g. diurnal) covariance of near-surface ozone and deposition velocity is accounted for
in calculating total dry deposition. We find that this covariance based on the 3-h fields for the ocean is small and leads to a small increase of 1.4% in the annual deposition flux to the ocean compared to when monthly averaged fields of deposition velocity and ozone concentration are used. On the other hand, this increase is about 28% over land surfaces, demonstrating a considerable degree of covariance. The likely reason for the small covariance over the ocean surface is that the near-surface ozone is influenced more by vertical turbulent exchange than by dry deposition due to the relatively small values of $v_d$ over
such surfaces. On the other hand, deposition velocities over land surfaces are large and they influence the near-surface ozone to a greater degree than turbulent vertical air exchange, particularly during stable conditions. The MACC data for all ten years were used, which is useful for examining interannual variability of deposition.

## 5.1 Global distribution of surface ozone and dry deposition flux

As an example, Figure 10a shows the mean surface ozone mixing ratio (ppbv) based on the MACC reanalysis for 2005. It is apparent that relatively high values occur in the Northern Hemisphere, particularly in the mid latitudes which can be attributed to the larger precursor emissions in these areas. The mixing ratios over the ocean are generally greater than those over the land, which can be partly attributed to the smaller dry deposition velocities to the ocean and hence lower deposition. There are ozone minima around the Equator, especially over the Pacific Ocean.

The annual oceanic ozone dry deposition flux obtained using the MACC ozone reanalysis coupled with the deposition velocities from ACCESS-UKCA averaged over 2005 presented in Figure 10b indicates that the largest flux values between 0.014–0.02 $\mu$g m$^{-2}$ s$^{-1}$ are observed within latitudes 10–40°N. The flux in the Southern Hemisphere is lower than that in the Northern Hemisphere and decreases with latitude.

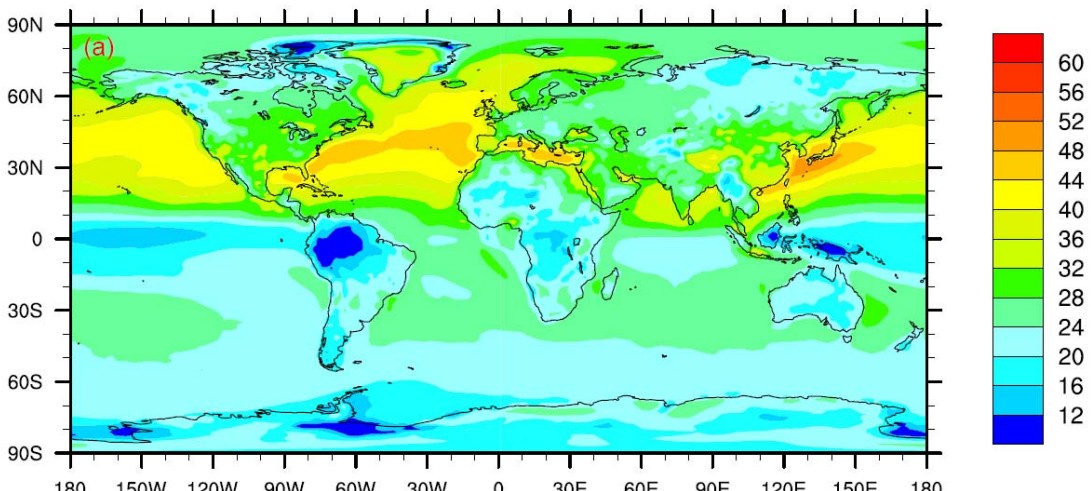

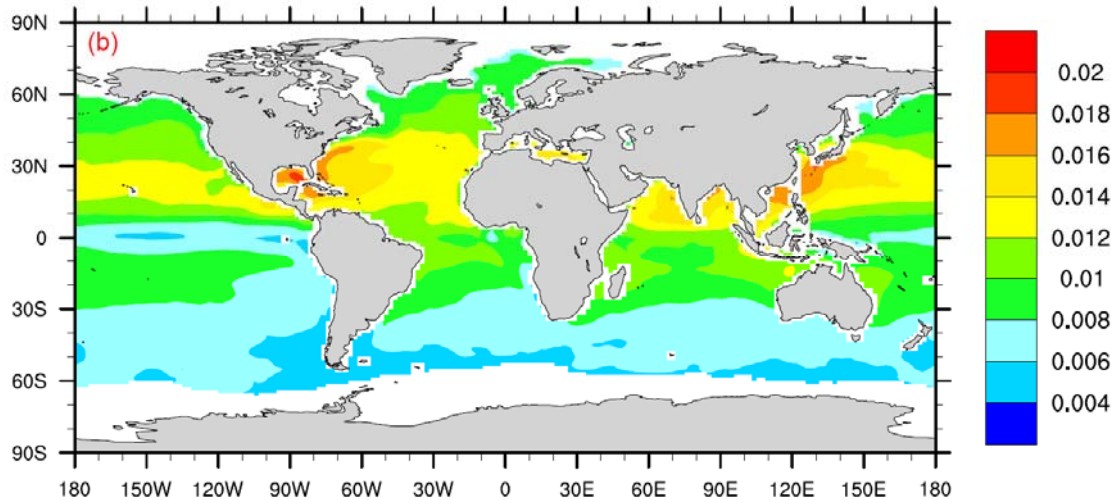

**Figure 10: (a) Mean surface ozone mixing ratio (ppbv) and (b) mean oceanic ozone dry deposition flux ($\mu$g m$^{-2}$ s$^{-1}$), for 2005 obtained from the MACC reanalysis.**

## 5.2 Dry deposition budgets

Figure 11 presents the annual ozone dry deposition obtained using the MACC reanalysis as a function of year. The oceanic deposition (Figure 11a) lies between 86.5–108.3 Tg O$_3$ yr$^{-1}$ with the average being 93.9 ± 7.5 Tg yr$^{-1}$ where the error bounds correspond to one standard deviation and are solely due to interannual variation. (In our calculations, the oceanic component excludes sea ice and coastal grid boxes and on average covers 62.4% of the Earth's surface.) The largest deposition occurs for 2005–2007. Oceanic deposition in the Northern Hemisphere (49.0 ± 3.4 Tg yr$^{-1}$) is somewhat larger than that in the Southern Hemisphere (44.9 ± 4.5 Tg yr$^{-1}$) due to the higher O$_3$ concentrations and slightly larger oceanic deposition velocities in the former, although the Earth's area covered by the ocean is larger by approximately 30% in the Southern Hemisphere. The main reason why the dry deposition velocities to the ocean in the Northern Hemisphere are larger (e.g. 0.020 vs. 0.017 m s$^{-1}$ on average for the year 2005) is that the SSTs for the Northern Hemisphere are warmer than those for the Southern Hemisphere (e.g. 295.3 K vs. 291.2 K on average for the same year). In our formulation, deposition velocity to the ocean is dominated by the surface-resistance term ( $r_c$ ) which in turn depends on SST. Overall the higher the SST the higher the oceanic deposition velocity. There is a hint in Figure 11a that the pattern of interannual variability of the global oceanic deposition follows that for the Southern Hemisphere more closely.

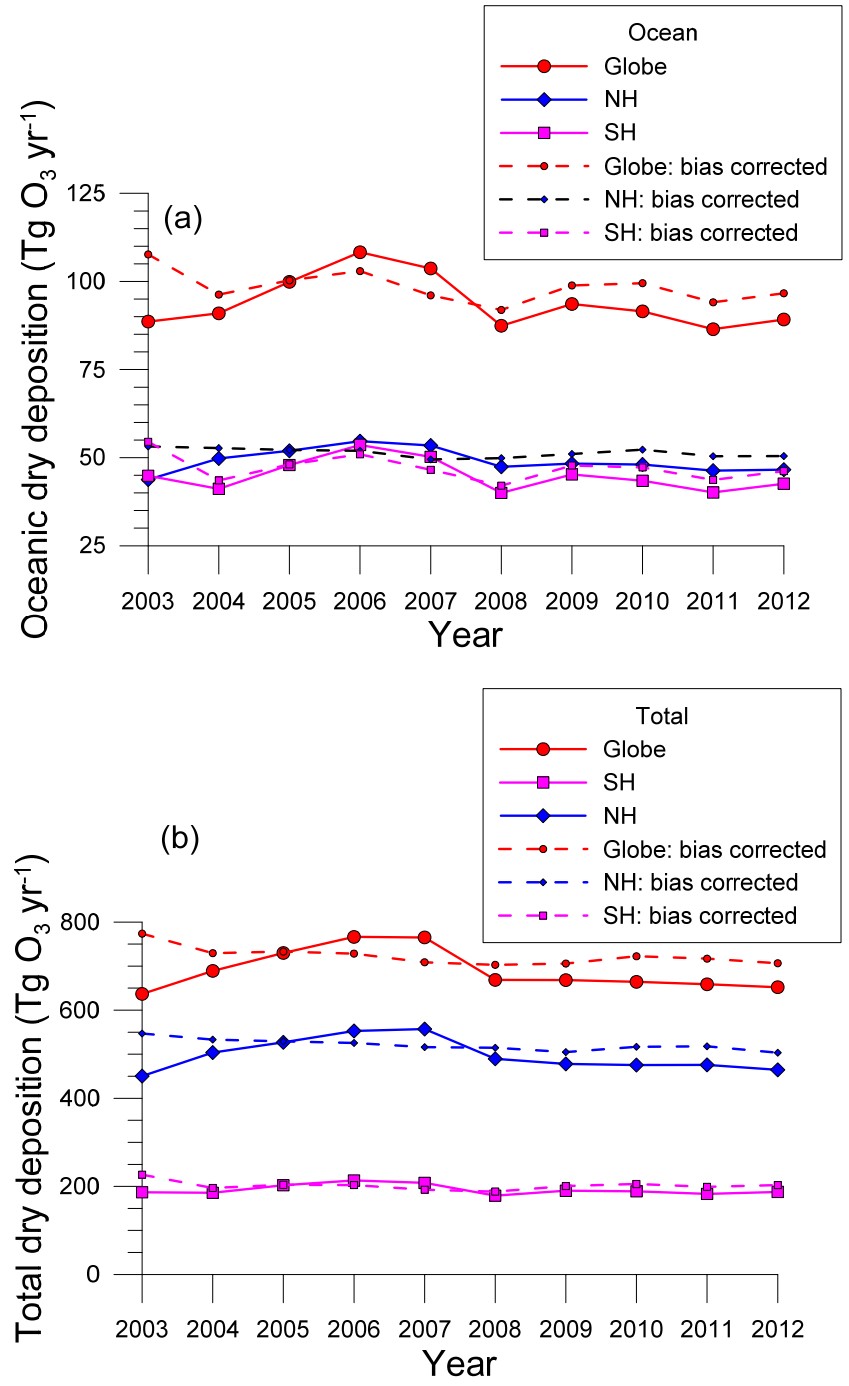

**Figure 11: Annual variation of the dry deposition of ozone (Tg O₃ yr⁻¹) obtained using the ACCESS-UKCA model (dotted lines) and the MACC reanalysis (solid lines): (a) the ocean component and (b) total. (NH = Northern Hemisphere, SH = Southern Hemisphere).**

The variation of the global total deposition obtained using the MACC reanalysis in Figure 11b is in the range 636.9–766.3 Tg yr$^{-1}$ and the mean value is 689.9 ± 47.0 Tg yr$^{-1}$ with the largest deposition amounts for 2005–2007. The total deposition to the Northern Hemisphere (497.5 ± 36.9 Tg yr$^{-1}$) is 72% of the total deposition and is two and a half times larger than that to the Southern Hemisphere (192.4 ± 11.4 Tg yr$^{-1}$) because in the former the O$_3$ concentrations are larger coupled with the larger coverage of the Earth's area by land for which deposition velocities are larger than for water. On average, deposition to the ocean is approximately 14% of the total deposition. The pattern of interannual variability of the global deposition is dominated by that for the Northern Hemisphere. This variability is driven by MACC ozone concentration changes rather than changes in deposition velocity.

The MACC reanalysis is not free from bias as demonstrated in a number of studies (e.g., Inness et al., 2013; Gaudel et al., 2015; Giordano et al., 2015; Katragkou et al., 2015). With regards to global bias in surface ozone, Figure 12 presents the annual averaged normalised median bias (%) of the MACC ozone mixing ratios relative to the Global Atmosphere Watch (GAW) surface observations (http://www.wmo.int/pages/prog/arep/gaw/gaw_home_en.html) for the years 2003–2012. We have derived this bias using the seasonal bias data taken from Benedictow et al. (2014) (http://macc.copernicus-atmosphere.eu/documents/maccii/deliverables/val/MACCII_VAL_DEL_D_83.6_REAreport04_20140729.pdf). Figure 12 shows that except for the first year the bias has remained within ±10%, and has been negative since 2008. The bias is the smallest for the year 2005. The total deposition for that year is 729.8 Tg yr$^{-1}$ of which 527.1 Tg yr$^{-1}$ is to the Northern Hemisphere and 202.7 Tg yr$^{-1}$ is to the Southern Hemisphere. The total oceanic deposition for that year is 99.9 Tg yr$^{-1}$ of which 52.0 Tg yr$^{-1}$ is to the Northern Hemisphere and 47.9 Tg yr$^{-1}$ is to the Southern Hemisphere. Thus the total deposition to non-water surfaces is 629.9 Tg yr$^{-1}$.

Interestingly, the shape of the interannual variation of total deposition in Figure 11b (and also the interannual variation of total oceanic deposition in Figure 11a) is similar to that of the bias in Figure 12, suggesting that the interannual variability of dry deposition may at least partly be due to the interannual variability of bias in the MACC ozone. Figure 13 is a scatter plot of the annual averaged bias (%) in the MACC ozone versus the total global deposition and total oceanic deposition determined based on the MACC data for the years 2003–2012. The annual bias and deposition appear well correlated, with a liner correlation $r^2 = 0.83$ for the total deposition and $r^2 = 0.65$ for the oceanic deposition. Based on the linear fits, the annual ozone deposition value corresponding to zero bias is 717.6 Tg yr$^{-1}$ for the globe and 97.8 Tg yr$^{-1}$ for the ocean.

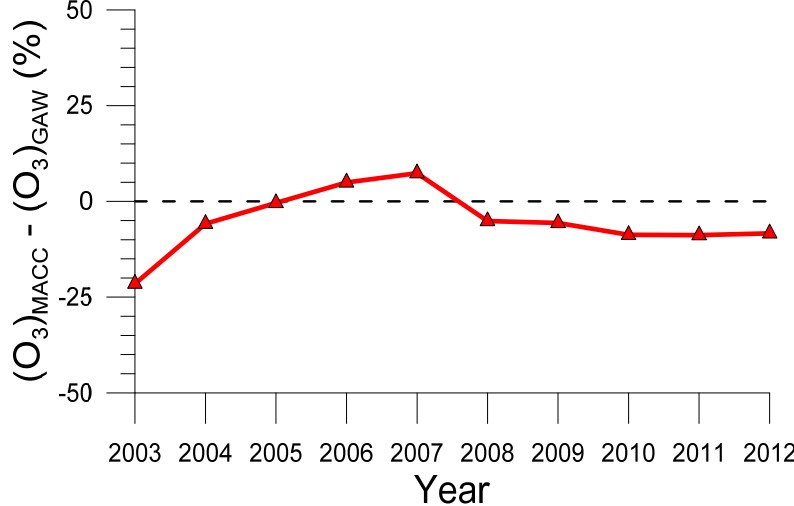

**Figure 12: Annual averaged normalised median bias (%) of the MACC ozone reanalysis mixing ratios relative to the Global Atmosphere Watch (GAW) surface observations for the years 2003–2012.**

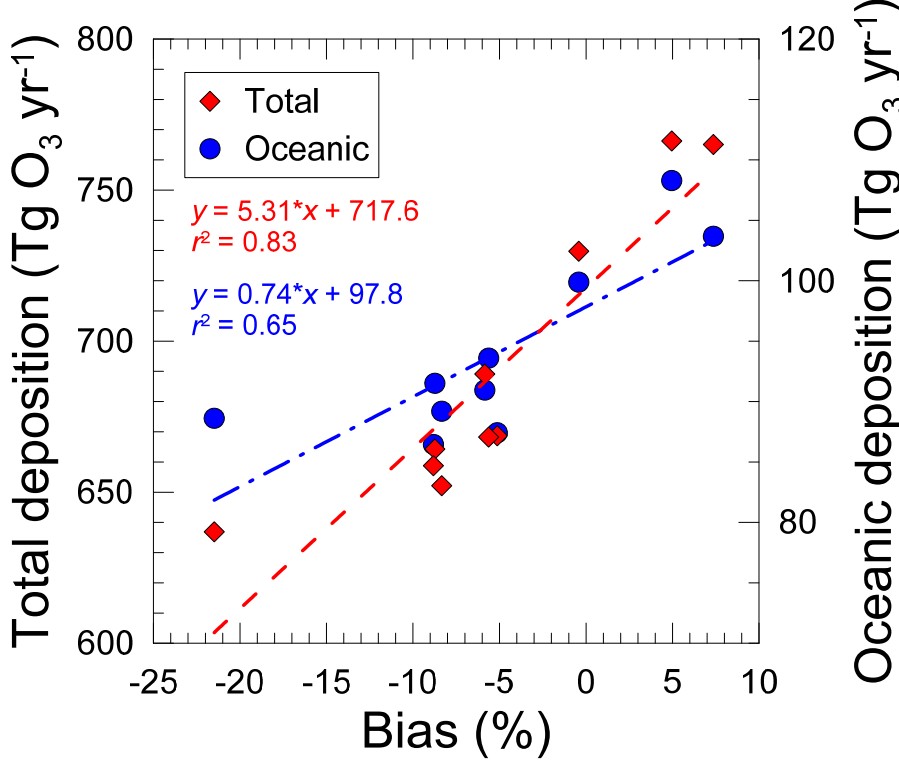

**Figure 13: Scatter plot of annual averaged normalised median bias (%) in the MACC ozone reanalysis mixing ratios relative to the Global Atmosphere Watch (GAW) surface observations for the years 2003–2012 versus the total global deposition and total deposition to the ocean determined based on the MACC reanalysis. The best fit lines are also shown.**

The above bias correction does not provide information on annual variability without bias. A simple (but rather crude) way to correct the MACC based deposition ($d_{p0}$) for each year for the bias ($b_s$) is to calculate a new annual deposition $d_p = d_{p0}(1 - b_s/100)$ and then calculate the average over the ten years and the corresponding standard deviation. By assuming that the observed global bias is uniform over the land, ocean and hemispheric components, averages and standard deviations for these components can also be derived. The bias corrected deposition values are plotted in Figure 11. Based on these, the average oceanic deposition is 98.4 ± 4.5 Tg yr$^{-1}$ and the average total global deposition is 722.8 ± 20.9 Tg yr$^{-1}$. The total deposition to non-water surfaces is 624.4 ± 17.4 Tg yr$^{-1}$. These averages are very similar to those for the year 2005 and those corresponding to the zero bias in Figure 13. The $1\sigma$ uncertainties on these figures are due to interannual variability alone.

**Table 2: Mean ozone dry deposition based on the MACC data for the years 2003–2012 (Tg O$_3$ yr$^{-1}$)[1], and from other studies that also report the oceanic component. The uncertainties are ±1$\sigma$.**

| Method | Ocean | | | Land | | | Total | | |
|---|---|---|---|---|---|---|---|---|---|
| | NH | SH | Global | NH | SH | Global | NH | SH | Global |
| Galbally and Roy (1980)[2] | 191 | 300 | 491 | 459 | 141 | 600 | 650 | 441 | 1091 |
| Ganzeveld et al. (2009)[3] | - | - | 291.5 | - | - | 543.5 | - | - | 835 |
| Hardacre et al. (2015)[4] | - | - | 340 | - | - | 638 | 646 | 332 | 978 ± 127 |
| Present study | 51.3 | 47.1 | 98.4 ± 30.0 | 469.6 | 154.8 | 624.4 ± 82.0 | 520.9 | 201.9 | 722.8 ± 87.3 |

[1]The ocean component excludes sea ice and coastal grid boxes (which are included in the land component) and on average covers 62.4% of the Earth's surface; [2]In Galbally and Roy (1980), the oceanic component includes ice and there is an uncertainty of ±50% in their estimates; [3]avearge values from two model runs; [4]the oceanic component is based on the average of values from two different land-cover schemes, and the NH and SH components based on Hardacre (2017, personal communication); NH = Northern Hemisphere, SH = Southern Hemisphere.

The above MACC based deposition amounts can be compared with other studies, going as far back as Galbally and Roy (1980) (see Table 2). The total land-based deposition in Galbally and Roy (1980) is similar to the present estimates but their

oceanic deposition is five times as large. This may be due to the fact that at that time there were only coastal measurements of ozone uptake by seawater with larger deposition velocities than for the open ocean.

More recently, Hardacre et al. (2015) analysed monthly ozone dry deposition fluxes from 15 global chemistry transport models (not including UKCA) driven by meteorological fields for the year 2001. These models use Wesely's scheme (1989)

for the deposition velocity calculation for both water and terrestrial surfaces. ACCESS-UKCA also uses Wesely's scheme for terrestrial surfaces. A comparison of observed dry deposition fluxes with those obtained from the above global chemistry transport models for terrestrial surfaces is presented by Hardacre et al. (2015). These authors noted that differences in ozone dry deposition flux to the ocean, driven by small absolute differences in dry deposition velocity but with large areal coverage by the ocean, are the largest contributor to differences in the total global $O_3$ deposition compared to any other surface type.

They determined that the mean total global deposition was $978 \pm 127$ Tg $O_3$ yr$^{-1}$ where the range corresponds to one standard deviation. By using two different land-cover schemes for partitioning fluxes, they determined that deposition to the ocean was in the range 250–591 (average 361) Tg yr$^{-1}$ across the model ensemble using one land-cover scheme that had 71.2% of the Earth's surface covered by water, and 209–538 (average 319) Tg yr$^{-1}$ using the other that had 68.6% of the global surface covered by water. The modelling study by Ganzeveld et al. (2009) points to an oceanic dry deposition estimate of 283–300

Tg yr$^{-1}$ and a global total of 833–837 Tg yr$^{-1}$ (Table 2). The oceanic deposition budgets in all these studies are more than three times larger than the 98.4 Tg yr$^{-1}$ value obtained in the present study. This much of difference cannot be explained by the slightly lower fraction of the global surface covered by water in the present calculations (i.e. 62.4%). The primary reason for this difference, as alluded earlier, is that the global chemistry transport models in these studies are all largely based on Wesely's (1989) deposition scheme which uses a constant surface resistance for water. As shown by Luhar et al. (2017) the

use of $r_c = 2200$ s m$^{-1}$ overestimates open ocean deposition velocity compared to the open-ocean measurements of Helmig et al. (2012) by a factor of 2 to 4 for cooler SSTs. The smaller oceanic deposition budget presented in this paper is consistent with these currently best available open-ocean measurements. The total deposition to non-water surfaces based on the MACC data is 624.4 Tg yr$^{-1}$, which is similar to 638 Tg yr$^{-1}$ obtained by Hardacre et al. (2015) (using an average oceanic deposition of 340 Tg yr$^{-1}$ in their calculations) and 600 Tg yr$^{-1}$ obtained by Galbally and Roy (1980).

There are other studies that report on the total global dry deposition. Stevenson et al. (2006) report an average global ozone dry deposition of $1003 \pm 200$ Tg yr$^{-1}$ for the year 2000 based on 21 models. The average deposition calculated by Wild (2007) using 17 post 2000 modelling studies is $949 \pm 222$ Tg yr$^{-1}$, whereas that reported by Young et al. (2013) for the year 2000 based on a subset of six models participating in the ACCMIP intercomparison study is $1094 \pm 264$ Tg yr$^{-1}$. However, these studies do not report values of the oceanic deposition separately.

It is clear from the above comparison that the land component of total deposition remains similar in all the studies (after subtracting an oceanic contribution of $\sim 300$ Tg yr$^{-1}$ from the total in the previous studies). The new estimate of dry deposition to the ocean of $\sim 100$ Tg $O_3$ yr$^{-1}$ is approximately a third of the current model estimates. This reduction

corresponds to an approximately 67% decrease in the modelled oceanic dry deposition and 20% decrease in the modelled total dry deposition.

Based on a simple calculation involving the tropospheric ozone budgets given in IPCC (2013) following Young et al. (2013), we estimate that the reduction in the modelled dry deposition rate by ~ 200 Tg $O_3$ yr$^{-1}$ over the ocean presented here (with all other factors being unchanged) results in roughly 5% increase in modelled tropospheric ozone burden and an equivalent increase in tropospheric ozone lifetime. In the marine boundary layer at mid to high latitudes, the effect of the ozone increase would be expected to be larger.

**5.3 Uncertainty in annual ozone dry deposition**

The ($1\sigma$) uncertainty in the global ozone deposition of 1003 ± 200 Tg yr$^{-1}$ reported by Stevenson et al. (2006), 949 ± 222 Tg yr$^{-1}$ by Wild (2007), 1094 ± 264 Tg yr$^{-1}$ by Young et al. (2013) and 978 ± 127 Tg yr$^{-1}$ by Hardacre et al. (2015) based on multi-model runs is by and large all due to model to model variations. Here we attempt a comparable uncertainty estimate.

Our modelling yields an ozone deposition loss to the ocean of 98.4 ± 4.5 Tg yr$^{-1}$ and a total global deposition of 722.8 ± 20.9 Tg yr$^{-1}$, with the $1\sigma$ error bounds in these estimates only representing the 10-year interannual variability in the modelled deposition velocity and MACC concentration fields. These error bounds do not include any uncertainties that may arise due to the approximations and assumptions used in the deposition velocity (e.g. iodide concentration, reaction rate constant etc.) or MACC ozone reanalysis methodologies.

In earlier discussion of the oceanic dry deposition velocity in Section 2.2 it was identified that calculations of the reaction-diffusion length scale ($l_m = \delta_m / c_0$) based on oceanic observations of iodide and SST give results varying between 24.0–1.2 μm for the SST range 2–33°C, and it is 3 μm at 23°C. In the subsequent work in this paper a value of 3 μm is used for the depth of the reaction-diffusion sublayer ($\delta_m$). Considering Figure 3c and Figure 3d, the waterside deposition velocity varies by at most a factor of 2 for the range of variations in $\delta_m$ that lie between the two extreme physical limits of a one-layer diffusive model and a one-layer turbulent model. These limits implicitly encompass the uncertainties in the rate constant and iodide concentrations. Assuming these limits can be described as three-sigma, we estimate that the one-sigma uncertainty in $\delta_m$ is approximately ±30%. This uncertainty can directly feed into the uncertainty of the global ozone deposition rates. Combining it with the $1\sigma$ error bounds in the MACC ozone reanalysis gives a combined relative uncertainty of ±31% or our total oceanic deposition of 98.4 Tg yr$^{-1}$ with an uncertainty ± 30.5 Tg yr$^{-1}$.

An alternate approach to estimate uncertainty for the oceanic component, which is the main focus here, is to consider the scatter in the deposition velocity observations of Helmig et al. (2012) in Figure 7, which show some large fluctuations in the $v_d$ data that are not present in the modelled values. We take the difference between the amount of scatter in the $v_d$ data and

that in the modelled values as a measure of uncertainty that is not captured by the model. We call this difference residual uncertainty ($\sigma_{vdr}$) which we aim to account for. In order to quantify $\sigma_{vdr}$, four SST ranges, namely < 15, 15–21, 21–28 and > 28 °C were considered, which approximately correspond to the SST ranges of the cruise experiments shown in Figure 7. For each SST range, the $v_d$ data were detrended by fitting a linear $v_d$ versus SST regression line and the variance ($\sigma_{vdo}^2$) of the detrended data was calculated. Similarly, the variance ($\sigma_{vdm}^2$) of the detrended modelled $v_d$ values was calculated for the same SST range. Thus $\sigma_{vdr} = (\sigma_{vdo}^2 - \sigma_{vdm}^2)^{1/2}$, and its values were 0.0046, 0.0093, 0.0049 and 0.0056 cm s$^{-1}$ for the above SST ranges, respectively.

To calculate the uncertainty in oceanic deposition flux due to $\sigma_{vdr}$, the 3-h modelled depositions velocity at each grid point was perturbed by $\pm \sigma_{vdr}$ (which is selected from one of the above four values depending on which SST range the SST at the grid point falls into) and then multiplied with the 3-h MACC ozone fields. This was done for all 10 years which yielded the uncertainty in the oceanic ozone deposition flux due to $\pm \sigma_{vdr}$ to be ± 30.0 Tg yr$^{-1}$ (which includes the small interannual variability of ± 4.5 Tg yr$^{-1}$ stated earlier).

Thus for our total oceanic deposition of 98.4 Tg yr$^{-1}$ we have two uncertainty estimates of ± 30.5 Tg yr$^{-1}$ based on the uncertainty in the deposition velocity model and the uncertainty ± 30.0 Tg yr$^{-1}$ based on the random differences between the model and observations for the available data. While these independent estimates agree very well, the wider issue is that the world's oceans are under-sampled with regard to ozone uptake measurements, it cannot be assumed that the available measurements are a representative sample of the ozone uptake over the world's oceans and the uncertainties, consequently, are probably underestimates.

The total oceanic deposition and uncertainty estimates calculated here can be contrasted with the value 340.0 ± 98.6 Tg yr$^{-1}$ obtained by Hardacre et al. (2015). It is interesting to note that our mean and standard deviation are both approximately a third of the respective values obtained by Hardacre et al. (2015). There would also be uncertainty in the MACC ozone data apart from their interannual variability which we have not considered.

With regards to the uncertainty in deposition to non-water surfaces, since our model uses the same Wesely (1989) deposition scheme as most other global models for such surfaces, we assume that the corresponding uncertainty would be similar to that in those models. Only Hardacre et al. (2015) report uncertainties in deposition fluxes to both water and non-water surfaces, with the latter calculated to be ± 80.0 Tg yr$^{-1}$. This value when combined with the interannual variability of ± 17.4 Tg yr$^{-1}$ for non-water surfaces obtained here leads to a total uncertainty of ± 82.0 Tg yr$^{-1}$ for such surfaces. Hence the total uncertainty combining this non-water component (± 82.0 Tg yr$^{-1}$) and the water component derived above (± 30.0 Tg yr$^{-1}$) is ± 87.3 Tg yr$^{-1}$.

The global oceanic and total deposition fluxes with the revised uncertainty are 98.4 ± 30.0 Tg yr$^{-1}$ and 722.8 ± 87.3 Tg yr$^{-1}$, respectively (Table 2). The reduction in the total uncertainty compared to Hardacre et al.'s (2015) value of ± 127 Tg yr$^{-1}$ is due to due to the reduction in the magnitude of the water component of deposition flux.

**6 Conclusions**

The ocean phase surface resistance term dominates over aerodynamic and atmospheric viscous sublayer resistances in commonly used parameterisations of ozone dry deposition velocity at the oceanic surface. Recent mechanistic schemes used to parameterise the oceanic surface resistance take into account the simultaneous effects of ozone solubility in water, waterside molecular diffusion and turbulent transfer, and first-order chemical reaction of ozone with dissolved iodide and other compounds. Luhar et al. (2017) formulated a semi-empirical scheme that described existing deposition velocity data

well, but in order to compensate for the impact of overestimation of turbulent transfer within the waterside viscous sublayer it put an artificial limit on the iodide concentration to a fixed depth of the order of a few micrometres from the water surface whereas in reality iodide is present through the depth of the oceanic mixing layer. Here we presented a new analytical two-layer formulation for the oceanic surface resistance that avoids making this limiting assumption. Instead, it makes the valid assumption that the influence of turbulent transfer can be neglected compared to the influence of chemical reaction within

the top layer of water that is of the order of the reaction-diffusion length scale (typically a few micrometres). In the water layer below, both chemical reaction and turbulent transfer act together and are accounted for. The new scheme has an asymptotic behaviour that is consistent with the current limits of ozone dry deposition when either chemical reaction or turbulent transfer dominate. When compared against the available observed deposition velocity dependencies on sea surface temperature, the performance of the new two-layer dry deposition scheme as realised within the global chemistry-climate

model ACCESS-UKCA (at UM vn8.4) was found to be satisfactory with the inclusion of only the aqueous iodide-O$_3$ reaction. However, additional ocean-based measurements are needed for further development and evaluation of ozone deposition parametrisations with an aim of reducing uncertainty in ozone modelling.

By using the 3-h MACC reanalysis for ozone concentration for the years 2003–2012 and the corresponding modelled 3-h deposition velocity values using the new dry deposition scheme for the ocean presented here and the default scheme for the

other surface types, the deposition budget has been calculated and quantified. The annual ozone deposition value is 722.8 ± 87.3 Tg O$_3$ yr$^{-1}$ for the globe and 98.4 ± 30.0 Tg O$_3$ yr$^{-1}$ for the ocean. This new estimate of oceanic dry deposition represents a reduction of approximately 67% over the current estimates of oceanic deposition. This reduction leads to a 20% decrease in the modelled total global dry deposition, an increase of approximately 5% in the modelled tropospheric ozone burden, and an equivalent increase in tropospheric ozone lifetime.

**Acknowledgements**

This research was undertaken with the assistance of resources and services from the National Computational Infrastructure (NCI), which is supported by the Australian government. Luke Abraham of University of Cambridge, and Fiona O'Connor and Mohit Dalvi of the U.K. Met Office are thanked for their assistance with the UM-UKCA model. C. Hardacre is acknowledged for providing additional details of their published deposition work. We are indebted to Ludovic Bariteau for kindly supplying data related to their published work. ERA-Interim data from the European Centre for Medium-Range Weather Forecasts and the ozone reanalysis data from the European MACC program were used in this research. We thank the three anonymous reviewers for their helpful comments.

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
