# Peer review of "A revised global ozone dry deposition estimate based on a new twolayer parameterisation for air-sea exchange and the multi-year MACC composition reanalysis"

_Atmospheric Chemistry and Physics, 2017_

## Referee Comment (RC1) · Anonymous Referee #2 · 14 Oct 2017

This paper proposes some updates to the paper the authors published earlier this year (10.5194/acp-17-3749-2017) describing the deposition of ozone to the ocean. Some changes are made to the parameterization and the resulting deposition velocities are used to explore the impacts on the global budget of ozone on the ACCESS-UK model. They have concerns about the veracity of their atmospheric chemistry model so explore the impact of the new deposition velocities with the MACC reanalysis. They conclude that their new parameterization has some skill in representing the rather sparse observational dataset and that with this new parameterization for deposition velocities, the

mass of ozone deposited to the ocean is significantly reduced with implications for both the budget and distribution of ozone.

I have concerns that this paper represents a small incremental advance over the previously published paper. For example, Figure 5 only shows small difference between the new and old schemes which was published only a few months ago. Ideally this paper would have been coupled into the paper published only a few months ago. However, this is a decision to be made by the editor.

Fundamentally this paper provides a description of an improved O3 deposition parameterization for the oceans, shows that there is some fit between the observations reasonably well and fundamentally changes the tropospheric budget for ozone especially over oceanic regions. These are important conclusions.

I have a few questions and queries to suggest for the improvement and shortening of the paper which I make below. Assuming that these can be made I would recommend publication.

Major comments.

Ocean O3 lifetime.

The premise of the paper is that reaction between O3 and I- is the only sink for O3 in the ocean. There is no discussion of the validity of this assumption. There is significant evidence that dissolved organic matter (DOM) may play a significant role in deposition of ozone to the surface (see for example 10.1029/2008GB003301). Yet this isn't discussed in the text. There should be some justification given for ignoring the role of DOM in their calculations.

The parameterization appears to do a reasonable job of simulating the deposition observation (Figure 4) without the need for an additional ocean side O3 sink. However there has been a tuning of the model (top half of page 12) so it isn't obvious that a missing O3 sink process (such as that offered by DOM) would be 'diagnosed' though

a model to measurement comparison. Figure 4 looks very similar to a figure shown in the author's previous paper. It would be useful to show this data in an x-y plot and give some indication of the error associated with the parameterization against the observations.

Our current understanding of DOM, its reactivity to O3 and distribution is poor. However, the authors should discuss the implications of them ignoring the potential DOM sink. Whilst they are doing that they should also discuss the implication of their choice of iodide distribution. They are using the distribution based on the parameterization of McDonald, but the literature also includes the Chance parameterization which gives higher I- concentrations and I think gives a slightly different spread. What are the implications of this?

There should be more of a discussion of the uncertainties of the O3 lifetime in the ocean, and how the parameterization tuning to the observations provides some solid ground to base the subsequent budget analysis. What impact do these uncertainties have on the budget?

Diagnosing the ozone deposition flux

The new parameterization is put into the ACCESS-UKCA model and this gives a global flux of O3 deposition to the ocean of ∼86 Tg yr-1. The model is known to have a low bias for O3 and so a significant body of work is done to calculate the flux from the MACC analysis fields of O3 and then a bias corrected MACC analysis fields. This lengthens the paper significantly for almost no gain. The canonical value for ocean deposition of O3 is around the 340 Tg yr-1 from the Hardacre study. The new parameterization gives the ACCESS model a deposition of 86 Tg yr-1, the MACC Analysis 93 Tg yr-1 and the bias corrected MACC Analysis 98.4 Tg yr-1. Compared to the Hardacre values these numbers are essentially the same (25%, 27% and 28%) respectively especially when the uncertainty in the parameterization are considered. There are pages of text describing the MACC data but I don't think it substantially changes the conclusions especially as the authors are forced to bias correct the MACC data. Would it not make more sense to bias correct the ACCESS data?

My suggestion is to remove this section or to perform the bias correction on the ACCESS data. It doesn't add anything to the story but it makes the document substantially longer.

Minor Comment.

There should be more details on the performance of the ACCESS physical model. There are no details of performance, parameterization choices etc. There should be more details given. What aspects of the model impact the parameterization used?

Typo on page 10, line 14 "fullydescribe" missing a space

---

## Referee Comment (RC2) · Anonymous Referee #4 · 17 Oct 2017

**Manuscript Review: Luhar et al., Revising global ozone dry deposition estimates based on a new mechanistic parameterisation for air-sea exchange and the multiyear, ACPD–2017-768.**

A modified version of a recently presented ozone ocean dry deposition scheme [*Luhar et al.*, 2017] is presented. The model performance is evaluated by comparing modelled ozone deposition velocities with previously published data from oceanic cruises. Further, the global ozone ocean flux is modeled based on this new model configuration, yielding a lower oceanic ozone sink than prior estimates. The ozone ocean flux is then compared with the ozone land sink, and a new total global ozone flux estimate is derived.

Major Comments:

While the authors repeatedly highlight their work as being a new 'scheme', as far as I understand this modeling in essence differs only in one aspect (the ocean layer is described in two, rather than a single layer) from their prior ACP publication [*Luhar et al.*, 2017] that was submitted only ten months prior to this current paper. The article claims this ozone flux parameterization and modeling to be a novel development. However, from reading the earlier publication [*Luhar et al.*, 2017] again, and the works by *Ganzeveld et al.* [2009] and *Fairall et al.* [2007], it appears that the the physical and ocean biochemical dependency description were mostly adaptations of principles presented in these earlier publications.

In this model the ozone ocean flux description builds exclusively on chemical removal of ozone by reaction with iodide ($I^-$). Consideration of this reaction is not that novel, having been proposed quite some time ago. Other previous work has suggested that, while the $I^-$ reaction has high significance, other secondary reactions, such as those with dissolved organic matter (DOM) in the ocean surface microlayer, may play a role in the ozone reaction as well [*Ganzeveld et al.*, 2009; *Coleman et al.*, 2010]. *Ganzeveld et al.* [2009] showed, for example, that evaluation of the simulated $O_3$ dry deposition velocities with a 1-layer version of the [*Fairall et al.*, 2007] model, including only $I^-$ in the calculation of total reactivity, underestimated the measured coastal deposition velocities. The role of dissolved organic matter (DOM)-$O_3$ chemistry was proposed to explain these discrepancies. [*Coleman et al.*, 2010] specifically addressed the role of DOM-$O_3$ chemistry in deposition to the Atlantic Ocean. These authors conclude: "… iodide reactions alone cannot account for observed deposition velocities. Consequently, we suggest a missing chemical sink due to reactions of ozone with organic matter at the air-sea interface." It does not appear that this Luhar et al. article takes this into consideration. The question if and how much uncertainty potentially results from this neglect is not addressed by their publication.

Further, building exclusively on $O_3 + I^-$ chemistry, the proper description and consideration of $I^-$ in the ocean must be of high importance. The article does not provide any detail on what data the $I^-$ oceanic description builds on. Are these new observations? Or is the $I^-$ modeled based on other relationships? In [*Ganzeveld et al.*, 2009], $I^-$ was estimated based on its correlation with nitrate. While this seemed to be a reasonable, and possibly the best possible approach at that time, does this paper take advantage of the much improved $I^-$ description presented by *Chance et al.* [2014]? Despite this progress, there certainly remains large uncertainty in the spatial and temporal representation of $I^-$, e.g. its concentrations in high-latitude waters, which is hampered by a lack of in-situ observations. This is actually the region where, according to this study by Luhar, the largest differences in the $O_3$ dry deposition velocities compared to the older/other deposition approaches are observed (Figure 9 in [*Luhar et al.*, 2017]).  As far as I understand, these uncertainties are likely many times larger than the rather narrow uncertainty windows in the ozone deposition budgets that are presented in this new Luhar et al. publication. Unfortunately, the authors do not elaborate on this question, which I consider a severe neglect.

Developed flux estimates are presented with error windows (see abstract line 23) that are on the order of 5%, but those windows are simply the standard deviation of the year to year variability in the modeled flux based on changing meteorology. They are not the uncertainty in the estimates of the ozone flux. Those, likely, would be much larger, making the way this is presented quite misleading.

Secondary analyses, such as comparison of modeled boundary layer ozone, global ocean flux budgets, and attribution of the oceanic flux to the total global flux that build on this modeling, are consequently highly uncertain as well. I therefore question the value of these secondary analyses. For instance, differences between the two schemes shown in Figures 9 and 10 are on the order of 0-25%. Of how much value are these results when the uncertainty in the reactivity is maybe on the order of 100-200%? To me, what I think needs to be addressed most urgently are these questions:

- How much of the total oceanic ozone flux can be attributed to I⁻, versus other reactants?
- What are the oceanic I⁻ fields? How does I⁻ change with time and location? And how can this variability be best incorporated into the model?

Unfortunately, these questions are not identified and addressed in this paper.

Other Comments:

The *Bariteau et al.* [2010] article makes a point that ozone fluxes are higher near the coasts compared to the open ocean. Was that considered in this modeling? And if not, how much uncertainty is potentially due to this neglect?

The performance of the deposition model leans heavily on data from the six open ocean cruises shown in Figures 4 and 5. Did the authors attempt other comparisons, for instance using any of the other data sets that were summarized in [*Helmig et al.*, 2012]? Given that, as currently done, it appears that the validation relies exclusively on the data from a single group, it should be shown that those cruises are representative for the entirety of available data. Furthermore, these data do not appear to be publicly available, or hosted by any data center? In our research center (and I think this is becoming more common within the community) it is customary to cite the doi of the data set, invite the providers of the data for co-authorship, or at least acknowledge the data providers, whenever those data make a significant contribution to a publication, including comparisons in modeling studies.

Page 1/Line 11:  I don't see what the term 'consistent' qualifies in this context (consistent with what?)? So, I recommend deleting this.

1/17:  As detailed above, I think the term 'new' is a bit of an overstatement. Yes, this paper does present some advancements in the ozone ocean uptake modeling, but most of the mechanisms, considered reactants, and dependencies were presented in prior publications.

1/25:  Atmospheric models appear to mostly overestimate surface ozone [*Parrish et al.*, 2014]. The results presented in this Luhar et al. manuscript show an increase of modeled ozone, thereby further increasing the discrepancy between models and observations. So, from that perspective, don't these changes go in the wrong direction?

3/1:  Is this ('commonly') indeed still the case, given that *Ganzeveld et al.* [2009] published a process-based parameterization and model implementation some 8 years ago?

3/9:  *Ganzeveld et al.* [2009] should also be cited here?

3/28:  How is the oceanic layer between the surface and 10 m depth represented?

3/24:  As mentioned earlier already, a section is needed here explaining how oceanic $I^-$ concentrations were derived and included in the modeling.

4/14:  …considered, but a ….

5/1:  Why 'consistent' ?

5/21: …. included, and a ….

16/1:  ….that the new ….

16/2:   …..(2017), but unlike the latter, the new ….

Figure 5:  As mentioned earlier, this figure nicely shows that improvements made through this work are merely nuances, while very large uncertainties and deficiencies in other areas are overlooked.

Figure 6:  *Ganzeveld et al.* [2009], in their Figure 3a and 3b provide similar analyses for January and July. Unfortunately, they do not show annual mean analyses. However, comparing their data with this Figure 6 reveals some very large differences. While *Ganzeveld et al.* [2009] report the high latitude oceans exhibiting the highest ozone deposition velocities, this Figure 6 shows that the ocean deposition velocity is highest over the tropical oceans. Isn't that a rather large disagreement that should trigger an in depth analysis and discussion?

18/9-10:  ACCESS-UKCA then seems to differ from other models that seem to overestimate surface ozone [*Parrish et al.*, 2014]?

20/6: As mentioned earlier, this seems to disagree with the results from [*Ganzeveld et al.*, 2009]?

20/10:  Replace 'concentration' with 'mixing ratio'.

21/5:  This really should not be called 'uncertainty' then.  Maybe use the term 'error bar'.

26/15-21:  In this discussion about the differences between this and the previous studies, changes are attributed to a better representation of the commonly applied constant $r_c$ of Wesely's scheme, as already demonstrated by *Ganzeveld et al.* [2009]. Their process-based approach arrived at a global $O_3$ oceanic deposition budget that was not that different from models using Wesely's constant $r_c$. This, in my opinion, calls for a discussion of how these large differences between these two process-based approaches, one being extended to two layers, only considering $I^-$, and the other one using a single layer but including more reactants including DOM, can be reconciled.

26/29: …., whereas that ….

27/26:  Given my reservations detailed above in my opinion this is a rather subjective and invalid evaluation.

28/8: ….deposition, an increase ….burden, and an ….

**Literature cited:**

Bariteau, L., D. Helmig, C. W. Fairall, J. E. Hare, J. Hueber, and E. K. Lang (2010), Determination of oceanic ozone deposition by ship-borne eddy covariance flux measurements, *Atmospheric Measurement Techniques*, *3*(2), 441-455.

Chance, R., A. R. Baker, L. Carpenter, and T. D. Jickells (2014), The distribution of iodide at the sea surface, *Environmental science. Processes & impacts*, *16*(8), 1841-1859, doi:10.1039/c4em00139g.

Coleman, L., S. Varghese, O. P. Tripathi, S. G. Jennings, and C. D. O'Dowd (2010), Regional-scale ozone deposition to North-East Atlantic waters, *Advances in Meteorology*, *Article ID 243701*, 1-16.

Fairall, C. W., D. Helmig, L. Ganzeveld, and J. Hare (2007), Water-side turbulence enhancement of ozone deposition to the ocean, *Atmospheric Chemistry and Physics*, *7*, 443-451.

Ganzeveld, L., D. Helmig, C. W. Fairall, J. Hare, and A. Pozzer (2009), Atmosphere-ocean ozone exchange: A global modeling study of biogeochemical, atmospheric, and waterside turbulence dependencies, *Global Biogeochemical Cycles*, *23*, doi:Gb4021 10.1029/2008gb003301.

Helmig, D., E. K. Lang, L. Bariteau, P. Boylan, C. W. Fairall, L. Ganzeveld, J. E. Hare, J. Hueber, and M. Pallandt (2012), Atmosphere-ocean ozone fluxes during the TexAQS 2006, STRATUS 2006, GOMECC 2007, GasEX 2008, and AMMA 2008 cruises, *J. Geophys. Res.*, *117*, D04305, doi:04310.01029/02011JD015955.

Luhar, A. K., I. E. Galbally, M. T. Woodhouse, and M. Thatcher (2017), An improved parameterisation of ozone dry deposition to the ocean and its impact in a global climate–chemistry model, *Atmos. Chem. Phys.*, *17*(5), 3749-3767, doi:10.5194/acp-17-3749-2017.

Parrish, D. D., et al. (2014), Long-term changes in lower tropospheric baseline ozone concentrations: Comparing chemistry-climate models and observations at northern midlatitudes, *Journal of Geophysical Research-Atmospheres*, *119*(9), 5719-5736, doi:10.1002/2013jd021435.

---

## Referee Comment (RC3) · Anonymous Referee #3 · 18 Oct 2017

**1   Overview**

The work described in this manuscript builds on the previous work of Luhar et al., 2017 in which the authors developed a more detailed, process based, two layer parametrization for dry deposition of ozone to oceans. In this study the two layer parametrization is refined and then implemented in the UKCA model. The model output is combined with MACC reanalysis data to calculate new estimates for global ozone deposition to wateroceans and total global ozone deposition. These new estimates are considerably

less than current estimates of global ozone deposition. The model output combined with MACC reanalysis data is also used to analyse inter-annual trends in ozone dry deposition.

**2 General comments**

Overall this manuscript is well written and describes an improved parametrization for ozone dry deposition to water. The improved parametrization addresses uncertainty in deposition of ozone to water, which is the main driver of uncertainty in global ozone dry deposition.

The manuscript is generally well laid out and the figures are clear. My main comments refer to Section 2. This section is quite important as it describes the new deposition parametrization, but it is a bit hard to follow.

(i) It would helpful if the authors could include a diagram of the different layers that form within the sea surface micro layer (e.g. reaction-diffusion sub-layer, bottom layer) that shows a summary of the processes (e.g. chemistry, chemistry/turbulence and reaction with iodide) that occur for in each layer and the main equations that are used to parameterize these processes.

(ii) I think it would also be helpful in Section 2 if the authors could more clearly describe how their improved scheme differs from that described in Luhar et al., 2017.
**3 Specific comments**

**3.1 Section 1**

P4, L13-14: Consider rephrasing to "A more appropriate parametrisation for Kt which varies **with** zm in the viscous sublayer..." to improve the readability and meaning of the sentence.

P4, L20-22: Could the authors provide a brief description of the "asymptotic behaviour" (also mentioned in the abstract). Or refer the reader to section 2.1.

**3.2 Section 2**

P5, L20-22: Consider rephrasing to "The second layer, **which is** deeper than the reaction-diffusion sublayer, ..." to improve the readability and meaning of the sentence.

P6, L12-13: Consider rephrasing to "The first two, namely the flux at the water surface (z = 0) obtained using Eq. (4) should be equal **to** F0 and the concentration at the interface..." to improve the readability and meaning of the sentence.

Figure 1 caption: Consider rephrasing to "Figure 1: Variation of the oceanic component of ozone dry deposition velocity multiplied by ozone solubility as a function of sea surface temperature (SST, $^\circ$C), **(a, c)**; and reactivity a (s-1), **(b, d)**. Curves determined using the two-layer deposition scheme (Eq. (16)) for several c0 values used in $\delta$ m = c0 l m, **(a, b)** and several $\delta$ m values, **(c, d)**. The variations obtained using the one-layer deposition scheme with (Eq. (18)) and without (Eq. (19)) waterside turbulent transfer (i.e. reaction-diffusion only) are also shown. The waterside friction velocity ( u*w ) used was 0.01 m s-1." to improve the readability.
**3.3 Section 5.2**

P21, L5-10: Can the authors suggest why there are larger ozone dry deposition velocities in the Northern Hemisphere?

---

## Short Comment (SC1) · 20 Oct 2017

I enjoyed reading this paper which carefully lays out improvements to the authors' previous oceanic O3 dry deposition formulation by including chemical reactivity below the reaction-diffusion sublayer. I have a question which I don't think any of the reviewers raise, on the reaction-diffusion sublayer thickness: how were the values of the constant c0 chosen?

[Figure]

2017.

---

## Author Comment (AC1) · 20 Dec 2017

**Reply by the authors to L. J. Carpenter's comment on**
"Revising global ozone dry deposition estimates based on a new mechanistic parameterisation for air-sea exchange and the multi-year MACC composition reanalysis" (#acp-2017-768)

**Comment:** *I enjoyed reading this paper which carefully lays out improvements to the authors' previous oceanic O3 dry deposition formulation by including chemical reactivity below the reaction-diffusion sublayer. I have a question which I don't think any of the reviewers raise, on the reaction-diffusion sublayer thickness: how were the values of the constant c0 chosen?*

**Response and changes in manuscript:** We thank Prof. Lucy Carpenter for her views on our work. In the reaction-diffusion sublayer, $l_m = (D/a)^{1/2}$ is an appropriate length scale. Thus, using scaling argument, it is reasonable to assume that the thickness of the reaction-diffusion sublayer ($\delta_m$) is proportional to $l_m$ with the coefficient of proportionality ($c_0$) being a constant of the order unity. In Figures 1a and 1b of our paper, we plot $1/r_c$ curves for three values of $c_0$, viz. 0.2, 0.4 and 0.7, which fall within the two asymptotic limits (equivalent to $c_0 \rightarrow 0$ and $c_0 \rightarrow \infty$). The value $c_0 = 0.4$ was selected for further sensitivity analysis reported in Figure 3 because it leads to a $1/r_c$ variation that roughly lies in the middle of the two asymptotic limits as shown in Figures 1a and 1b. As mentioned on Page 12 Line 13, in all our subsequent deposition calculations we used Option 4 with $\delta_m = 3$ microns (see the 1st para on Page 13) which obviously does not need a specification of $c_0$.

We include the above clarification in the revised version of the paper.

---

## Author Comment (AC2) · 20 Dec 2017

**Reply by the authors to the Referee #1's comments on**
"Revising global ozone dry deposition estimates based on a new mechanistic parameterisation for air-sea exchange and the multi-year MACC composition reanalysis" (#acp-2017-768)

**Anonymous Referee #1 (RC1)**

We are grateful to the Referee for his/her comments. In the following, we provide our responses to these comments.

**(1) Comment:** *This paper proposes some updates to the paper the authors published earlier this year (10.5194/acp-17-3749-2017) describing the deposition of ozone to the ocean. Some changes are made to the parameterization and the resulting deposition velocities are used to explore the impacts on the global budget of ozone on the ACCESS-UK model. They have concerns about the veracity of their atmospheric chemistry model so explore the impact of the new deposition velocities with the MACC reanalysis. They conclude that their new parameterization has some skill in representing the rather sparse observational dataset and that with this new parameterization for deposition velocities, the mass of ozone deposited to the ocean is significantly reduced with implications for both the budget and distribution of ozone.*

*I have concerns that this paper represents a small incremental advance over the previously published paper. For example, Figure 5 only shows small difference between the new and old schemes which was published only a few months ago. Ideally this paper would have been coupled into the paper published only a few months ago. However, this is a decision to be made by the editor.*

**Response:** The work conducted about a year ago by the authors that the Referee is referring to and that was published in November 2016 in ACPD and then in March 2017 in ACP (https://doi.org/10.5194/acp-17-3749-2017), provided the subsequent impetus and ideas to extend that work, resulting in the present work. The present work is novel in two important ways: First, it derives a new formula for the waterside ozone deposition velocity (Eq. 16) which corrects a basic flaw in the two-layer scheme reported in the previous paper by including chemical reactivity throughout the oceanic mixing layer (as is observed for dissolved iodide) rather than just within the reaction-diffusion sublayer a few microns thick. The new model will also apply to any other chemical compounds that are taken up by the oceanic mixing layer. Second, our work makes use of the European MACC ozone reanalyses for ten years to constraint the ozone dry deposition fluxes better and to provide a measure of interannual variability.

One could indeed say that the present paper represents an incremental advance (as is the case with much of the scientific research, if not most), and in our opinion this advance is significant.

We emphasise in the paper at a few places (e.g. see the 1st para on Page 16 and Figure 5) and the Referee also notes, that there is only a small difference between the new and old two-layer schemes in terms of performance in simulating oceanic deposition velocity data. However, the difference between the two, if you like, is that unlike the old scheme the new scheme performs well for the right reasons (the old scheme artificially limits chemical reactivity to the reaction-diffusion sublayer to compensate for the overestimation of the impact of waterside turbulence). The present work also demonstrates the importance of chemistry-turbulence interactions and how they differ in the new and old deposition schemes.

**Changes in manuscript:** We clarify the differences between the old and new two-layer schemes along the lines of the response given above.

**(2) Comment:** *Fundamentally this paper provides a description of an improved O3 deposition parameterization for the oceans, shows that there is some fit between the observations reasonably well and fundamentally changes the tropospheric budget for ozone especially over oceanic regions. These are important conclusions.*

*I have a few questions and queries to suggest for the improvement and shortening of the paper which I make below. Assuming that these can be made I would recommend publication.*

**Response:** Thanks for the comment.

**Changes in manuscript:** None.

**(3) Comment:** *Major comments.*

*Ocean O3 lifetime.*

*The premise of the paper is that reaction between O3 and I- is the only sink for O3 in the ocean. There is no discussion of the validity of this assumption. There is significant evidence that dissolved organic matter (DOM) may play a significant role in deposition of ozone to the surface (see for example 10.1029/2008GB003301). Yet this isn't discussed in the text. There should be some justification given for ignoring the role of DOM in their calculations.*

**Response:** The open-ocean ozone deposition velocity data of Helmig et al. (2012) we have used for model testing are limited in sample size and contain substantial fluctuations. However, they are the best available and only ones that have used a surface based eddy-covariance approach which provides a direct way of measuring deposition velocity. The present work demonstrates that the chemical reaction of $O_3$ with dissolved iodide is able to adequately describe these deposition velocity data within the observed scatter and uncertainty in the input parameterisations (e.g. the second-order reaction rate constant $k$). We have done additional work on dissolved organic matter (DOM) and it is reported below.

**Changes in manuscript:** With regards to DOM (or DOC, dissolved organic carbon), we include the following in the paper. This also addresses a comment made by Referee #2.

Reaction with dissolved organic carbon (DOC)

Some studies have considered the impact on ozone dry deposition of ozone reaction with dissolved compounds other than iodide. Chang et al. (2004) included reactions of ozone with iodide, dimethyl sulfide (DMS), ethene and propene and showed that the reaction with iodide was by far the fastest in most cases. In their global modelling, Ganzeveld et al. (2009) included ozone reaction with chlorophyll-a as a first order approximation to examine the possible role of dissolved organic matter (DOM), and found that this reaction significantly increased dry deposition velocities at coastal sites (with mixed results compared to observations) and yielded only small changes to deposition velocity for open ocean sites. Sarwar et al. (2016) included ozone reactions with iodide, dissolved organic carbon (DOC) (equivalent of DOM), DMS and bromide in their ozone modelling for summer months in the Northern Hemisphere, and found that the impact of DOC on the simulated deposition velocity was comparable to that of iodide, with the other reactions contributing much less. Coleman et al. (2010) showed that in addition to iodide the inclusion of DOC in their empirical scheme described daytime deposition observations better in coastal waters of North Atlantic. We are not aware of any previous study that has compared modelled deposition velocities involving the impact of the $O_3$-DOC reaction with the Helmig et al. (2012) data and their sea surface temperature (SST) dependence.

The work presented in our paper shows that the inclusion of only the iodide-$O_3$ reaction yields a satisfactory agreement of the modelled deposition velocities with the open-ocean observations of Helmig et al. (2012) within the uncertainty of the data and input parameterisations. Inclusion of additional reactions in the model would enhance the ozone loss to the ocean and thus increase deposition velocities. It is instructive, however, to carry out a simple sensitivity analysis involving ozone reaction with dissolved DOC.

For open ocean surface waters, Hansell et al. (2009) report DOC concentration values of 70–80 µM (M = molar or mole per litre) in tropical and subtropical regions (40°N to 40°S), ~ 40–50 µM in subpolar seas and in the circumpolar Southern Ocean (> 50°S), and about 70 µM in the Arctic Ocean (> 70°N). Based on Hansell et al. (2009), Sarwar et al. (2016) used a mean DOC concentration of 67 µM over the Northern Hemisphere. A recent analysis by Massicotte et al. (2017) gives an average DOC value of 52 µM for oceans.

There are no definitive, directly measured values available for the second-order rate coefficient ($k$) for the aqueous DOC-$O_3$ reaction. Coleman et al. (2010) empirically derived $k = 3.44 \times 10^6$ M$^{-1}$ s$^{-1}$ based on data fitting, whereas Sarwar et al. (2016) used $k = 4.0 \times 10^6$ M$^{-1}$ s$^{-1}$ noting that this value together with their selected DOC concentration yields a first order rate constant of ~ 268 s$^{-1}$ that lies between the two values 100 s$^{-1}$ (open-ocean) and 500 s$^{-1}$ (coastal waters) used by Carpenter et al. (2013) based on the modelling by Ganzeveld et al. (2009).

Clearly there is considerable uncertainty in the oceanic DOC concentration and its spatial and temporal variability, and in the corresponding second-order rate coefficient and its dependencies. For our purposes, we use a mean $k = 3.7 \times 10^6$ M$^{-1}$ s$^{-1}$ (which lies is in the middle of the above two values) and a DOC concentration of 52 µM (Massicotte et al., 2017) in our two-layer scheme, together with an integrated chemical loss rate $a = \sum_i k_i C_i$, where the summation is over the iodide and DOC reactions with ozone ($i = 1, 2$).

The attached Figure 1 shows the variation of $\alpha v_{dw}$ ($= 1/r_c$) (where $\alpha$ is ozone solubility, $v_{dw}$ is the waterside dry deposition velocity and $r_c$ is the surface resistance) as a function of SST determined using our two-layer deposition scheme incorporating the ozone reaction with: only iodide (this curve is the same as Option 4 in Figure 3 of our paper, with the reaction-diffusion layer thickness $\delta_m = 3$ µm), only DOC, and the two reactions together for three values of $\delta_m$. Compared to the iodide-only curve, the inclusion of DOC leads to a progressive increase in $\alpha v_{dw}$ as SST decreases for all values of $\delta_m$. As $\delta_m$ increases $\alpha v_{dw}$ decreases, but increasing $\delta_m$ beyond 6 µm has virtually has no impact on $\alpha v_{dw}$ (not shown). When only DOC is considered, $\alpha v_{dw}$ decreases with SST. Given that the $r_c$ term dominates in the determination of $v_d$, the behaviour of $\alpha v_{dw}$ represents that of $v_d$ (as is evident from Figures 3 and 4 in the paper). This suggests that the inclusion of DOC in the deposition scheme would worsen the model-data agreement shown in Figure 4, particularly for cooler water temperatures. Lowering the DOC concentration in the model to the lowest levels (~ 40 µM) reported by Hansell et al. (2009) does not improve the agreement either. A second-order rate coefficient for the DOC-$O_3$ reaction that decreases with SST (like that for the iodide-$O_3$ reaction) could explain the deposition velocity data better but any such observations are lacking at present. Clearly, more comprehensive observations of deposition velocity, its dependencies, and relevant input parameters for deposition calculations are needed to further constrain the waterside processes of ozone deposition.

In summary, there are considerable uncertainties concerning the oceanic DOC concentration and no directly measured values of the second-order rate coefficient ($k$) for the aqueous DOC-$O_3$ reaction.

Given the agreement of the existing soundly based iodide mechanism with the available uptake measurements for the open ocean, the inclusion of DOC in the open ocean ozone deposition is not undertaken. We note that for coastal waters, not investigated here, the case is probably different. The way coastal grid cells are treated in our global modelling is detailed in our Response #7 to Referee #2.

[Figure]

**Figure 1: Variation of the oceanic component of ozone dry deposition velocity multiplied by ozone solubility, $\alpha v_{dw}$ (= $1/r_c$), as a function of sea surface temperature. Curves determined using the present two-layer deposition scheme incorporating the ozone reaction with iodide [I⁻], dissolved organic carbon [DOC], and the two reactions together ([I] + [DOC]). The waterside friction velocity ($u_{*w}$) used was 0.01 m s⁻¹.**

References

Carpenter, L.J., MacDonald, S.M., Shaw, M.D., Kumar, R., Saunders, R.W., Parthipan, R., Wilson, J., Plane, J.M.C.: Atmospheric iodine levels influenced by sea surface emissions of inorganic iodine. Nat. Geosci. 6, 108–111. http://dx.doi.org/10.1038/ngeo1687, 2013.

Coleman, L., Varghese, S., Tripathi, O. P., Jennings, S. G., O'Dowd, C. D.: Regional-scale ozone deposition to North-East Atlantic waters. Advances in Meteorology, 2010, 16 pages, http://dx.doi.org/10.1155/2010/243701, 2010.

Ganzeveld et al. (2009). As referred to in the paper.

Hansell, D.A., Carlson, C.A., Repeta, D.J., Schlitzer, R.: Dissolved organic matter in the ocean. Oceanography 22, 202–211, 2009.

Massicotte, P., Asmala, E., Stedmon, C., Markager, S.: Global distribution of dissolved organic matter along the aquatic continuum: Across rivers, lakes and oceans, Science of the Total Environment 609, 180–191, http://dx.doi.org/10.1016/j.scitotenv.2017.07.076, 2017.

Sarwar, G., Kang, D., Foley, K., Schwede, D., Gantt, B., Mathur, R.: Technical note: Examining ozone deposition over seawater. Atmos. Environ. 141, 255−262, http://dx.doi.org/10.1016/j.atmosenv.2016.06.072, 2016.

**(4) Comment:** *The parameterization appears to do a reasonable job of simulating the deposition observation (Figure 4) without the need for an additional ocean side O3 sink. However there has been a tuning of the model (top half of page 12) so it isn't obvious that a missing O3 sink process (such as that offered by DOM) would be 'diagnosed' though a model to measurement comparison. Figure 4 looks very similar to a figure shown in the author's previous paper. It would be useful to show this data in an x-y plot and give some indication of the error associated with the parameterization against the observations.*

**Response and changes in manuscript:** As mentioned in our response on DOM above, the data available and used for model testing are not detailed enough to clearly discern or diagnose the potential impact of other reactions, let alone provide guidance on parameter values (e.g. reaction rate constant). Obviously, given such limitation there is some parameter value fitting, but this is informed by parameter bounds, for example the reaction-diffusion length scale, the asymptotes and the scatter in the iodide-O3 reaction rate constant, and the deposition velocity data.

As suggested by the Referee, we will include the attached Figure 2 showing an x-y plot of the modelled vs. observed deposition velocities. It presents the observed deposition velocities averaged over the data from each of the five cruise experiments versus the corresponding average values obtained from the model (with the error bars representing one standard deviation variation).

[Figure]

**Figure 2: Scatter plot of the ozone dry deposition velocities ($v_d$) obtained from the five cruise experiments versus the corresponding values obtained from the ACCESS-UKCA model using the new two-layer scheme (Eq. 16). Each point corresponds to the average over all values from one experiment. The error bars represent one standard deviation variation. Horizontal error bars are for the observed values and the vertical ones are for the modelled values.**

**(5) Comment:** *Our current understanding of DOM, its reactivity to O3 and distribution is poor. However, the authors should discuss the implications of them ignoring the potential DOM sink. Whilst they are doing that they should also discuss the implication of their choice of iodide distribution. They are using the distribution based on the parameterization of McDonald, but the literature also includes the Chance parameterization which gives higher I- concentrations and I think gives a slightly different spread. What are the implications of this?*

*There should be more of a discussion of the uncertainties of the O3 lifetime in the ocean, and how the parameterization tuning to the observations provides some solid ground to base the subsequent budget analysis. What impact do these uncertainties have on the budget?*

**Response:** As demonstrated in our response above on DOM, we agree with the Referee that our current understanding of DOM is poor.

A sensitivity analysis involving the iodide parameterisation by Chance et al. (2014) was reported by Luhar et al. (2017, https://doi.org/10.5194/acp-17-3749-2017) (see their Figure 5) and it was compared with the behaviour obtained by the MacDonald et al. (2014) iodide parameterisation. The Chance et al. (2014) iodide parameterisation gives larger iodide concentrations than the latter.

The attached Figure 3 is the same as Figure 3 in our present paper except that we have included an additional curve as Option 6 which is the same as Option 4 (the latter option is used in our deposition flux calculations) but using the iodide concentration parameterisation of Chance et al. (2014) $[I^-](nM) = 0.225(SST - 273.16)^2 + 19$. Compared to Option 4, Option 6 results in larger $\alpha v_{dw}$ ($= 1/r_c$) values and the relative difference between the two increases with SST; for example, for SSTs 5, 20 and 30°C, the Option 6 value is larger by 13, 29, 33%, respectively. Consequently, Option 6 overestimates the observed ozone deposition velocity data of Helmig et al. (2012) presented in Figure 4 of the paper, almost passing along the upper limits of the observed fluctuations in $v_d$. However, if Option 6 is used along with the second-order rate constant ($k$) without considering the data point of Hu et al. (1995) as in Option 3 (which gives lower $k$ values), then the values of deposition velocity obtained by Option 6 with the Chance et al. (2014) parameterisation are comparable to those by Option 4 that we have used.

What the above analysis suggests is that the deposition velocity data can also be described by the Chance et al. (2014) iodide parameterisation coupled with smaller values of $k$ (because it is the reactivity a, which is the product of the iodide concentration and second-order rate constant, that goes into the deposition velocity calculation) within the uncertainty of the deposition velocity data and second-order rate constant data.

Here we are guided and limited by the scant amount of deposition velocity observations that we have. Our deposition velocity scheme is developed based on sound arguments and with the selected parameters provides a good comparison with the data. However, as we have seen above, there is uncertainty in parameter values which would eventually be reflected in the deposition flux estimates. We have done additional work to estimate uncertainty in deposition flux taking into account the scatter in the ocean deposition velocity data, and this is described in our response to Comment #5 made by Referee #2.

[Figure]

**Figure 3: Variation of the oceanic component of ozone dry deposition velocity multiplied by ozone solubility, $\alpha v_{dw}$ (= $1/r_c$), as a function of sea surface temperature (SST, °C). Curves determined using the two-layer deposition scheme (Eq. 16) for various options for parameterising the second-order rate coefficient ($k$) (see text). The waterside friction velocity ($u_{*_w}$) used was 0.01 m s⁻¹.**

**Changes in manuscript:** We will include the bulk of the above discussion in the paper. The additional work on uncertainty in deposition flux which is described in our response to Comment #5 made by Referee #2 will also be included.

**(6) Comment:** *Diagnosing the ozone deposition flux*

*The new parameterization is put into the ACCESS-UKCA model and this gives a global flux of O3 deposition to the ocean of ~ 86 Tg yr-1. The model is known to have a low bias for O3 and so a significant body of work is done to calculate the flux from the MACC analysis fields of O3 and then a bias corrected MACC analysis fields. This lengthens the paper significantly for almost no gain. The canonical value for ocean deposition of O3 is around the 340 Tg yr-1 from the Hardacre study. The new parameterization gives the ACCESS model a deposition of 86 Tg yr-1, the MACC Analysis 93 Tg yr-1 and the bias corrected MACC Analysis 98.4 Tg yr-1. Compared to the Hardacre values these numbers are essentially the same (25%, 27% and 28%) respectively especially when the uncertainty in the parameterization are considered. There are pages of text describing the MACC data but I don't think it substantially changes the conclusions especially as the authors are forced to bias correct the MACC data. Would it not make more sense to bias correct the ACCESS data?*

*My suggestion is to remove this section or to perform the bias correction on the ACCESS data. It doesn't add anything to the story but it makes the document substantially longer.*

**Response:** The multi-year global MACC reanalyses are high-resolution, quality controlled data on atmospheric composition that are a valuable tool in developing and evaluating modelling schemes. They have not previously been used for deposition purposes. Their application in the second half of the paper together with the modelled deposition velocity distribution is an important component of

our work and is aimed at further reducing the uncertainty by constraining the ozone dry deposition budgets better.

From the point of view of ozone deposition to the ocean, the Referee is correct in saying that the deposition figures obtained using ACCESS, and the MACC analysis with and without the bias correction, are very similar (i.e. 86.1, 93.9 and 98.4 Tg yr$^{-1}$, respectively). However, when the total global deposition loss is calculated (including ocean, land and ice), the respective figures are 566.7, 689.9 and 722.8 Tg yr$^{-1}$ (see the top para on Page 18 and Table 1). Thus the underestimation of ozone by ACCESS is reflected more prominently in the deposition to non-ocean surfaces. We have used the MACC data to derive both oceanic and global deposition estimates, and we do think that these data have been usefully employed in the paper to constrain the oceanic and non-oceanic deposition losses of ozone.

Note that our modelled deposition velocity distribution that is multiplied with the MACC ozone data to calculate deposition flux does not depend on the chemistry component of the model. Deposition velocity is solely a function of parameters of the physical component of the model (e.g., SST (for iodide concentration), flow properties and turbulent mixing, and surface characteristics) and prescribed parameters (e.g., ozone molecular diffusivity and solubility).

ACCESS is based on the global UM-UKCA modelling system developed by a UK consortium (including the Met Office and Cambridge University) and we are dependent on them with regards to major model changes and upgrades. The UM-UKCA (v8.4) that we have implemented on our computer system generally underestimates tropospheric ozone. We hope have a better version in the future that would have a less bias. However, as mentioned above the ozone deposition velocity as determined in the model does not depend on the chemistry component of the model so its performance for ozone is not relevant here.

We are not sure what the Referee means by bias correcting the ACCESS data. If it means having a model that is almost bias free, then we do not think such a model is available as yet. If it means using ozone observations to bias fix ACCESS, then wouldn't this, in principle, be akin to using the MACC reanalyses which are basically bias corrected (or data assimilated) model estimates?

**Changes in manuscript:** In light of the above response, we do not agree with the Referee to remove the MACC data analysis. However, we will include some of the above points as clarification. In addition, since we do not use the ACCESS derived ozone flux anyway, and instead use the ACCESS derived deposition velocity coupled with the MACC ozone for the ozone flux calculation, we will delete relevant parts detailing the chemical component of ACCESS and also delete the ACCESS derived deposition flux estimates.

**(7) Comment:** *Minor Comment.*

*There should be more details on the performance of the ACCESS physical model. There are no details of performance, parameterization choices etc. There should be more details given. What aspects of the model impact the parameterization used?*

**Response:** ACCESS-UKCA uses the same physical atmosphere component as the UK Met Office's Unified Model (UM) and includes the UK Chemistry and Aerosol (UKCA) model for atmospheric composition (at UM vn8.4). In our simulations, ACCESS-UKCA is basically the same as UM-UKCA since the ACCESS specific ocean and land-surface components are not invoked. This is because we run the model in atmosphere-only mode with prescribed SSTs, and the UM's original land-surface scheme (JULES) is used.

For UKCA, we cite http://www.ukca.ac.uk, Morgenstern et al. (2009), Abraham et al. (2012), O'Connor (2014) and Woodhouse et al. (2015). The reference Abraham et al. (2012) is available at

http://www.ukca.ac.uk/images/b/b1/Umdp_084-umdp84.pdf which includes some detail of the dry deposition scheme (which is based on Wesely (1989)).

For ACCESS, a reference by Bi et al. (2013; http://www.bom.gov.au/amm/docs/2013/bi1.pdf) can be used. The assimilation of the ERA-Interim meteorological data into ACCESS is described by Uhe and Thatcher (2015; cited in the paper).

We use the MACC ozone reanalyses for the deposition flux calculations combined with the modelled deposition velocities. The latter do not depend on the chemistry component of the model so, as mentioned earlier, its performance is not relevant. Deposition velocity in the model is solely a function of parameters of the physical component of the model and prescribed inputs. Therefore, in effect we only use the physical atmosphere component of the model, and this component relevant to our model version is described by Walters et al. (2014, https://www.geosci-model-dev.net/7/361/2014/gmd-7-361-2014.pdf). A list of technical reports on UM given at http://cms.ncas.ac.uk/wiki/Docs/MetOfficeDocs (but accessing them requires username and password).

We show in the paper that our total deposition flux to non-water surfaces is similar to that calculated by other researchers (Lines 24-26, Page 25) and that the main difference lies in the oceanic deposition flux component.

**Changes in manuscript:** Based on the above response, we will revise the information and references provided in the paper.

**(8) Comment:** *Typo on page 10, line 14 "fullydescribe" missing a space*

**Response:** Done.

---

## Author Comment (AC3) · 20 Dec 2017

**Reply by the authors to the Referee #2's comments on**
"Revising global ozone dry deposition estimates based on a new mechanistic parameterisation for air-sea exchange and the multi-year MACC composition reanalysis" (#acp-2017-768)

**Anonymous Referee #2 (RC2)**

We are grateful to the Referee for a long set of comments. In the following, we provide a response to these comments.

(1) **Comment:** *A modified version of a recently presented ozone ocean dry deposition scheme [Luhar et al., 2017] is presented. The model performance is evaluated by comparing modelled ozone deposition velocities with previously published data from oceanic cruises. Further, the global ozone ocean flux is modeled based on this new model configuration, yielding a lower oceanic ozone sink than prior estimates. The ozone ocean flux is then compared with the ozone land sink, and a new total global ozone flux estimate is derived.*

**Response:** Thanks for the comment.

**Changes in manuscript:** None.

(2) **Comment:** *Major Comments:*

*While the authors repeatedly highlight their work as being a new 'scheme', as far as I understand this modeling in essence differs only in one aspect (the ocean layer is described in two, rather than a single layer) from their prior ACP publication [Luhar et al., 2017] that was submitted only ten months prior to this current paper. The article claims this ozone flux parameterization and modeling to be a novel development. However, from reading the earlier publication [Luhar et al., 2017] again, and the works by Ganzeveld et al. [2009] and Fairall et al. [2007], it appears that the the physical and ocean biochemical dependency description were mostly adaptations of principles presented in these earlier publications.*

**Response:** All the references mentioned above by the referee are included in our paper. We clearly elucidate in our paper what the shortcomings are of the previous model formulations, including the two-layer formulation used by Luhar et al. (2017, https://doi.org/10.5194/acp-17-3749-2017) which was based on the approach of Fairall et al. (2007). We agree that our new two-layer scheme (Eq. 16) includes the same overall physical and ocean biochemical processes as in the studies by these authors, but it improves the mathematical formulation by the use of a more realistic assumption. The new scheme corrects a serious flaw (as stated in the Luhar et al. (2017) paper) of the previous two-layer scheme by including chemical reactivity throughout the oceanic mixing layer (as is observed for dissolved iodide) rather than just within the top few microns of the water surface. The new model will also apply to any other chemical compounds that are taken up by the oceanic mixing layer. We accept that this represents an incremental advance in model development, but we believe it's a significant advance. Additionally, our work also makes use of the global ozone reanalyses developed under the European MACC program to constraint the ozone dry deposition budgets better. These reanalyses have not previously been used for ozone deposition purposes and thus provide scope for a novel application.

**Changes in manuscript:** To consider the Referee's point, perhaps we could qualify the title a little better to read "Revising global ozone dry deposition estimates based on a new two-layer parameterisation for air-sea exchange and the multi-year MACC composition reanalysis."

(3) **Comment:** *In this model the ozone ocean flux description builds exclusively on chemical removal of ozone by reaction with iodide (I-). Consideration of this reaction is not that novel, having been proposed quite some time ago. Other previous work has suggested that, while the I-reaction has high significance, other secondary reactions, such as those with dissolved organic matter (DOM) in the ocean surface microlayer, may play a role in the ozone reaction as well [Ganzeveld et al., 2009; Coleman et al., 2010]. Ganzeveld et al. [2009] showed, for example, that evaluation of the simulated O3 dry deposition velocities with a 1-layer version of the [Fairall et al., 2007] model, including only I- in the calculation of total reactivity, underestimated the measured coastal deposition velocities. The role of dissolved organic matter (DOM)-O3 chemistry was proposed to explain these discrepancies. [Coleman et al., 2010] specifically addressed the role of DOM-O3 chemistry in deposition to the Atlantic Ocean. These authors conclude: "… iodide reactions alone cannot account for observed deposition velocities. Consequently, we suggest a missing chemical sink due to reactions of ozone with organic matter at the air-sea interface." It does not appear that this Luhar et al. article takes this into consideration. The question if and how much uncertainty potentially results from this neglect is not addressed by their publication.*

**Response:** Part of this comment, particularly about DOM, is similar to a comment made by Referee #1. We thus refer to our Response #3 to Referee #1 where we present additional deposition calculations with DOC (equivalent of DOM) included.

One novelty of our work is to provide a better mathematical formulation for the inclusion of waterside chemical reactivity. The work of Luhar et al. (2017) showed clearly the limitation of the one-layer scheme of Fairall et al. (2007) in describing the deposition velocity data of Helmig et al. (2012) (as a result of an overestimation of waterside turbulence-chemistry interaction in this scheme), and replaced it with a two-layer model with an arbitrary constraint on chemical reactivity. The model presented here removes this arbitrary constraint on chemical reactivity and thereby is a more realistic model of ozone interaction with ocean water.

The focus of modelling in our paper is on ozone deposition to open ocean regions. Using the one-layer Fairall et al. (2007) scheme, Ganzeveld et al. (2009) show that the inclusion of the O3-DOM reaction (with DOM represented by chlorophyll-a) significantly increased deposition velocity at coastal sites but gave mixed results compared to observations, and that for open ocean sites, there were only small changes to deposition velocity. The work of Coleman et al. (2010) also relates to coastal waters. None of the papers by Fairall et al. (2007), Ganzeveld et al. (2009) and Coleman et al. (2010) has used the more recent open-ocean deposition velocity data of Helmig et al. (2012) that Luhar et al. (2017) and the present work use (because these data had not been available at the time).

Our work suggests that the ozone-iodide reaction is able to describe the available open-ocean deposition velocity data within the uncertainty of model parameters and scatter in the data. Clearly more observations are needed to establish the relative role of additional ozone reactions.

There is some evidence (e.g. Coleman et al., 2010) that ozone deposition velocities over coastal waters are larger than those over open oceans. Our deposition approach for coastal grid cells in the model is qualitatively consistent with that behaviour and is described in our Response #7 below.

We have calculated a measure of uncertainty in the global oceanic and total ozone deposition fluxes and the details are given in our Response #5 below.

**Changes in manuscript:** Our Response #3 to Referee #1 on DOM will be included along with a summary of the points discussed above.

(4) **Comment:** *Further, building exclusively on O3 + I- chemistry, the proper description and consideration of I- in the ocean must be of high importance. The article does not provide any detail on what data the I- oceanic description builds on. Are these new observations? Or is the I- modeled based on other relationships? In [Ganzeveld et al., 2009], I- was estimated based on its correlation with nitrate. While this seemed to be a reasonable, and possibly the best possible approach at that time, does this paper take advantage of the much improved I- description presented by Chance et al. [2014]? Despite this progress, there certainly remains large uncertainty in the spatial and temporal representation of I-, e.g. its concentrations in high-latitude waters, which is hampered by a lack of in-situ observations. This is actually the region where, according to this study by Luhar, the largest differences in the O3 dry deposition velocities compared to the older/other deposition approaches are observed (Figure 9 in [Luhar et al., 2017]). As far as I understand, these uncertainties are likely many times larger than the rather narrow uncertainty windows in the ozone deposition budgets that are presented in this new Luhar et al. publication. Unfortunately, the authors do not elaborate on this question, which I consider a severe neglect.*

**Response:** It is clearly stated in our paper that the ocean iodide concentration is based on Eq. (20), which is from MacDonald et al. (2014). Chance et al. (2014) examined statistical relationships between iodide and parameters such as SST, nitrate, salinity, chlorophyll-a, and mixed layer depth and found that SST was the strongest predictor of iodide. MacDonald et al. (2014, with Chance as a co-author) used data from several cruises in the Atlantic and Pacific oceans covering the latitudes 50°S–50°N to derive their parameterisation for iodide concentration, which we have used. A sensitivity analysis involving the iodide parameterisation by Chance et al. (2014) was reported by Luhar et al. (2017) (see their Figure 5) and it was compared with the behaviour obtained using the MacDonald et al. (2014) iodide parameterisation. We have included some discussion on the use of the Chance et al. parameterisation and included a deposition velocity curve as Option 6 based on this parameterisation in our Response #5 to Referee #1.

We agree that there is considerable uncertainty in the representation of iodide concentrations, particularly in high-latitude waters, due to the lack of in-situ observations. The rather narrow uncertainty windows in our annual ozone deposition fluxes are solely due to the interannual variability inherent in the modelled meteorology and the MACC ozone concentration fields. We have now done additional calculations to determine the uncertainty range better, and this is given as a response to Comment #5 below.

**Changes in manuscript:** The sensitivity to the Chance et al. (2014) iodide parameterisation conducted in our Response #5 to Referee #1 will be included. Additional work on uncertainty reported below under Comment #5 will be included.

(5) **Comment:** *Developed flux estimates are presented with error windows (see abstract line 23) that are on the order of 5%, but those windows are simply the standard deviation of the year to year variability in the modeled flux based on changing meteorology. They are not the uncertainty in the estimates of the ozone flux. Those, likely, would be much larger, making the way this is presented quite misleading.*

**Response:** The Referee is correct – the reported uncertainty in our annual ozone deposition fluxes is solely due to the interannual variability in the modelled meteorology (with nudging) and the MACC ozone concentration fields. We have done additional calculations to estimate uncertainty in deposition flux to the ocean better and this is described below. The global ocean and total ozone deposition fluxes with the new uncertainty estimates are **98.4 ± 30.0** Tg yr$^{-1}$ and **722.8 ± 87.0** Tg yr$^{-1}$, respectively.

**Changes in manuscript:** We plan to include a concise version of the following calculations.

Previous global deposition modelling studies include uncertainty estimates in the modelled annual deposition loss. Stevenson et al. (2006) report a global ozone deposition of 1003 ± 200 Tg yr[-1] for the year 2000 based on 21 models. The global deposition calculated by Wild (2007) using 17 post 2000 modelling studies is 949 ± 222 Tg yr[-1] whereas that reported by Young et al. (2013) for the year 2000 based on a subset of six models participating in the ACCMIP intercomparison study is 1094 ± 264 Tg yr[-1]. Hardacre et al. (2015) obtained a deposition loss of 978 ± 127 Tg yr[-1] based on 15 global chemistry transport models driven by meteorological fields for the year 2001. The ($1\sigma$) uncertainty in all these studies is by and large all due to model to model variations.

Our calculations in the paper yield a deposition loss to the ocean of **98.4 ± 4.5** Tg yr[-1] and a total global deposition of **722.8 ± 20.9** Tg yr[-1], with the $1\sigma$ error bounds in these estimates only representing the 10-year interannual variability in the modelled deposition velocity and MACC concentration fields. These error bounds do not include any other uncertainties in the deposition velocity modelling or those in the MACC concentration fields.

In our present single model study, there is no simple and reliable way of quantifying modelling uncertainty in deposition flux arising from the approximations and assumptions used. For the oceanic component, which is the main focus here, one way to estimate uncertainty is to consider the scatter in the deposition velocity observations of Helmig et al. (2012) shown in Figure 4 of our paper. This Figure shows some large fluctuations in the $v_d$ data that are not present in the modelled values. We take the difference between the scatter in the $v_d$ data and that in the modelled values as a measure of uncertainty that is not captured by the model. We call this difference the residual uncertainty which we aim to account for.

In order to quantify the residual uncertainty, four SST ranges, namely < 15, 15–21, 21–28 and > 28 °C were considered, which approximately correspond to the SST ranges of the cruise experiments shown in Figure 4. For each SST range, the $v_d$ data were detrended by fitting a linear regression line and the variance ($\sigma^2_{vdo}$) of the detrended data was calculated. Similarly, the variance ($\sigma^2_{vdm}$) of the detrended modelled $v_d$ values was calculated for each SST range. The quantity $(\sigma^2_{vdo} - \sigma^2_{vdm})^{1/2}$ is taken as the residual uncertainty, and it was 0.0046, 0.0093, 0.0049 and 0.0056 cm s[-1] for the above SST ranges, respectively.

To calculate the uncertainty range in oceanic deposition flux as a result of the residual uncertainty in $v_d$, the 3-h modelled depositions velocity fields were perturbed by the above residual uncertainty values (which depend on SST) and then multiplied with the 3-h MACC ozone fields. These calculations were done for all 10 years. The uncertainty range in the oceanic deposition flux thus obtained was ± 30.0 Tg yr[-1], which includes the small interannual variability of ± 4.5 Tg yr[-1] stated earlier. Our total oceanic deposition of 98.4 Tg yr[-1] with the new uncertainty ± 30.0 Tg yr[-1] can be contrasted with the value 340.0 ± 98.6 Tg yr[-1] obtained by Hardacre et al. (2015). It is interesting to note that our oceanic deposition and its uncertainty are both approximately a third of the respective values obtained by Hardacre et al. (2015). There would be some uncertainty in the MACC ozone data apart from their interannual variability which we have not considered.

With regards to uncertainty in deposition to non-water surfaces, since our model uses the same Wesely (1989) deposition scheme as most other models for such surfaces, we assume that the corresponding uncertainty would be similar to that in those models. Only Hardacre et al. (2015) report uncertainties in deposition fluxes to both water and non-water surfaces, with the latter calculated to be ± 80.0 Tg yr[-1]. This value when combined with the interannual variability of ± 17.4 Tg yr[-1] for non-water surfaces obtained in our paper leads to a total uncertainty of ± 82.0 Tg yr[-1] for such surfaces. Hence the total uncertainty combining this non-water component (± 82.0 Tg yr[-1]) and the revised water component (± 30.0 Tg yr[-1]) derived above is ± 87.0 Tg yr[-1].

In summary, with the revised uncertainty estimation, the global oceanic and total deposition fluxes are **98.4 ± 30.0** Tg yr$^{-1}$ and **722.8 ± 87.0** Tg yr$^{-1}$, respectively. The reduction in the total uncertainty compared to Hardacre et al.'s (2015) value of ± 127 Tg yr$^{-1}$ is due to the reduction in uncertainty in the water component of the deposition flux.

(6) **Comment:** *Secondary analyses, such as comparison of modeled boundary layer ozone, global ocean flux budgets, and attribution of the oceanic flux to the total global flux that build on this modeling, are consequently highly uncertain as well. I therefore question the value of these secondary analyses. For instance, differences between the two schemes shown in Figures 9 and 10 are on the order of 0-25%. Of how much value are these results when the uncertainty in the reactivity is maybe on the order of 100-200%? To me, what I think needs to be addressed most urgently are these questions:*

*- How much of the total oceanic ozone flux can be attributed to I-, versus other reactants?*

*- What are the oceanic I- fields? How does I- change with time and location? And how can this variability be best incorporated into the model?*

*Unfortunately, these questions are not identified and addressed in this paper.*

**Response:** Please see our Response #3 to Referee #1 on DOM which demonstrates that the ozone-iodide chemistry is sufficient to describe the available open-ocean $v_d$ measurements and their dependency on SST, and that the inclusion of other reactions (i.e. DOM) would deteriorate model performance.

As mentioned above, we used the iodide parameterisation of MacDonald et al. (2014) which is based on data from several cruises in the Atlantic and Pacific oceans covering the latitudes 50°S–50°N. This parameterisation is a function of SST, so depends on location and time. We have done additional uncertainty calculations described above. See Response #5 to Referee #1 on the Chance et al. (2014) iodide parameterisation.

**Changes in manuscript:** Additional calculations on DOM to be included. Details of the iodide parameterisation used will be provided. Appropriate text from our Response #5 to Referee #1 on the Chance et al. (2014) parameterisation to be included.

(7) **Comment:** *Other Comments:*

*The Bariteau et al. [2010] article makes a point that ozone fluxes are higher near the coasts compared to the open ocean. Was that considered in this modeling? And if not, how much uncertainty is potentially due to this neglect?*

**Response:** In our global modelling, the coastal grid cells that include terrestrial surface fractions are handled as follows.

A grid-box mean deposition loss rate is calculated using the individual modelled deposition velocities weighted by the fractions of the surface types present in the grid box. Our two-layer deposition scheme for the ocean is only used when the fraction of water surface in a grid box is greater than 60%. In all other cases Wesely's (1989) scheme for $v_d$ is used, including the use of $r_c$ = 2200 s m$^{-1}$ for water surface. Thus for coastal water grid cells that include fractions of other surface types, the modelled deposition velocities are greater than those for the grid cells fully covered by water because terrestrial surfaces have higher deposition velocities than water and also

because of the use of $r_c = 2200$ s m$^{-1}$ for the water title when its fraction is less than 60%. We will state this in the paper.

There is some evidence that the ozone deposition velocities over coastal waters are larger than those over open oceans (e.g. Coleman et al., 2010; Bariteau et al., 2010), which could be due to factors such as advection from land if the distance between the monitor and coastline (i.e. fetch) is limited, stronger chemical reactivity and turbulence with strong diurnal dependence. Our approach for treating coastal grid cells is qualitatively consistent with ozone deposition velocities over coastal waters being larger than over the open sea. But we do not include any additional/special processes for coastal waters.

**Changes in manuscript:** The above details to be summarised in the paper.

(8) **Comment:** *The performance of the deposition model leans heavily on data from the six open ocean cruises shown in Figures 4 and 5. Did the authors attempt other comparisons, for instance using any of the other data sets that were summarized in [Helmig et al., 2012]? Given that, as currently done, it appears that the validation relies exclusively on the data from a single group, it should be shown that those cruises are representative for the entirety of available data. Furthermore, these data do not appear to be publicly available, or hosted by any data center? In our research center (and I think this is becoming more common within the community) it is customary to cite the doi of the data set, invite the providers of the data for co-authorship, or at least acknowledge the data providers, whenever those data make a significant contribution to a publication, including comparisons in modeling studies.*

**Response:** It is correct that our model performance testing is based on published data from Helmig et al. (2012) which cover the latitudinal range 45°N–50°S. Surface based ozone flux stations employing the eddy-covariance technique enables a direct measurement of ozone dry deposition velocity. The data of Helmig et al. (2012) are the only such measurements available over the ocean.

We looked up the very sparse datasets by other researchers summarised by Helmig et al. (2012). None of these studies involved a surface-based eddy-covariance technique over the ocean. The ones that used such a technique were coastal measurements (i.e. Gallagher et al., 2001; Whitehead et al., 2010). The measurements by Lenschow et al. (1982) and Kawa and Peason (1989) used aircraft-based eddy-covariance over the ocean.

The considerably larger sample size and the possibility of the use of improved experimental techniques in the cruise measurements of Helmig et al. (2012) compared to those reported by earlier studies provide an incentive to use these data.

On line 23–26, on Page 14, we say "As in Luhar et al. (2017), the $v_d$ versus SST cruise data used for comparison with the model are those with the wind speed dependence retained (Ludovic Bariteau, personal communication, 2016) and not the data originally reported by Helmig et al. (2012) in which the wind-speed dependence was removed. While this approach is logically correct, there is not a large difference between the data with and without the wind-speed dependence." We will acknowledge L. Bariteau, as was done in Luhar et al. (2017).

**Changes in manuscript:** The other studies mentioned above can be cited. We will acknowledge L. Bariteau.

(9) **Comment:** *Page 1/Line 11: I don't see what the term 'consistent' qualifies in this context (consistent with what?)? So, I recommend deleting this.*

**Response:** Done.

(10) **Comment:** *1/17: As detailed above, I think the term 'new' is a bit of an overstatement. Yes, this paper does present some advancements in the ozone ocean uptake modeling, but most of the mechanisms, considered reactants, and dependencies were presented in prior publications.*

**Changes in manuscript:** We now use 'two-layer' in place of 'new' to qualify the work somewhat. Also, as we mentioned earlier, we will change the paper title slightly to read "Revising global ozone dry deposition estimates based on a new two-layer parameterisation for air-sea exchange and the multi-year MACC composition reanalysis."

(11) **Comment:** *1/25: Atmospheric models appear to mostly overestimate surface ozone [Parrish et al., 2014]. The results presented in this Luhar et al. manuscript show an increase of modeled ozone, thereby further increasing the discrepancy between models and observations. So, from that perspective, don't these changes go in the wrong direction?*

**Response:** The ACCMIP multi-model study by Young et al. (2013) shows an overall overestimation of ozone in the lower troposphere in northern mid-latitudes and underestimation in southern tropics and mid-latitudes. Parrish et al. (2014) also show that models overestimate ozone in northern mid-latitudes. Our version of UKCA in ACCESS generally underestimates observed tropospheric ozone, particularly in mid to high latitudes.

Our aim is to improve the process modelling of ozone deposition to the ocean. If a model performs worse when a particular process in improved then this points to issues with some other component(s) of the model.

To constraint the deposition fluxes better, we have used the MACC ozone reanalyses and in that case the deposition fluxes do not depend on ACCESS-UKCA's chemistry component because the modelled deposition velocity field used in the deposition flux calculation is solely a function of the input parameterisations Eqs. (20)–(23) and the physical component of the model (e.g., SST and turbulence).

**Changes in manuscript:** A summary of the above response will be included, noting that because we have used the MACC ozone reanalyses the deposition fluxes do not depend on ACCESS-UKCA's chemistry component.

(12) **Comment:** *3/1: Is this ('commonly') indeed still the case, given that Ganzeveld et al. [2009] published a process-based parameterization and model implementation some 8 years ago?*

**Response:** To our knowledge, all common global chemistry models described in the literature continue to use the Wesely (1989) approach (involving a constant value $r_c \approx 2000$ s m-1 for the surface resistance) for deposition to the ocean. Luhar et al. (2017) showed that the Wesely approach overestimates the deposition velocity data of Helmig et al. (2012). Ganzeveld et al. (2009) included the Fairall et al. (2007) one-layer scheme for ozone deposition velocity to the ocean in a global model, and found that compared to the Wesely approach it leads to 6% reduction in the total oceanic deposition of ozone and 0.5% reduction in the total global deposition. Luhar et al. (2017) showed that Fairall et al.'s one-layer scheme also overestimates the deposition velocity data due to flaws with its turbulence diffusivity assumption. Using better assumptions, the two-layer approach as presented in our paper describes these data much better.

**Changes in manuscript:** A summary of the above response can go into Introduction.

(13) **Comment:** *3/9: Ganzeveld et al. [2009] should also be cited here?*

**Response:** We do not think the Ganzeveld et al. (2009) reference is appropriate here because they used a one-layer scheme (of Fairall et al., 2007) and not a two-layer scheme.

**Changes in manuscript:** None.

(14) **Comment:** *3/28: How is the oceanic layer between the surface and 10 m depth represented?*

**Response:** The assumption regarding how the top oceanic layer is represented goes in the derivation of an expression for the waterside component of deposition velocity (or conversely surface resistance). This expression is implemented in the dry deposition module of our atmosphere-only global model. The global model itself does not include any explicit oceanic layer since there is no ocean model coupled. The prescribed SSTs are used in some of the input parameterisations for the waterside deposition velocity (Eq. 16) obtained using the two-layer scheme. Our scheme assumes that: chemical reactivity (or reactant) is present throughout the oceanic layer; in the top few micros of the oceanic layer ozone loss is dominated by chemical reaction (with no turbulent transfer); and in the oceanic layer below, both chemical reaction and turbulent transfer act together. We now provide a diagram (as suggested by Referee #3) to make this clearer (see our Response #2 to Referee #3).

**Changes in manuscript:** A new diagram as suggested by Referee #3 to be provided to make this clearer (see our Response #2 to Referee #3).

(15) **Comment:** *3/24: As mentioned earlier already, a section is needed here explaining how oceanic I- concentrations were derived and included in the modeling.*

**Response:** The following paragraph will be added:

**Changes in manuscript:** To add "This parameterisation is based on iodide data from cruises in the Atlantic and Pacific oceans covering the latitudes 50°S to 50°N, and is a function of SST (which is prescribed), so depends on location and time. It yields highest concentrations in warm tropical waters and lowest in cool waters at higher latitudes. Chance et al. (2014) examined statistical relationships between iodide and parameters such as SST, nitrate, salinity, chlorophyll-a and mixed layer depth, and found that SST was the strongest predictor of iodide in surface waters. In the study by Ganzeveld et al. (2009), oceanic surface nitrate was used as a proxy for iodide concentration. Luhar et al. (2017) used Eq. (20) in their modelling, but also presented a sensitivity of deposition velocity to the Chance at al. (2014) iodide parameterisation involving a dependence on SST." Appropriate text from our Response #5 to Referee #1 on the Chance et al. (2014) parameterisation also to be included.

(16) **Comment:** *4/14: …considered, but a ….*

**Response:** Done.

(17) **Comment:** *5/1: Why 'consistent' ?*

**Response:** The word has been deleted.

(18) **Comment:** *5/21: .... included, and a ....*

**Response:** Done.

(19) **Comment:** *16/1: ....that the new ....*

**Response:** Done.

(20) **Comment:** *16/2: .....(2017), but unlike the latter, the new ....*

**Response:** Done.

(21) **Comment:** *Figure 5: As mentioned earlier, this figure nicely shows that improvements made through this work are merely nuances, while very large uncertainties and deficiencies in other areas are overlooked.*

**Response:** The preference for the new scheme is summed up in Lines 1–3 on Page 10: "The model performance presented in Figure 4 leads us to conclude that new scheme performs as well as the two-layer reactivity scheme in Luhar et al. (2017) but unlike the latter the new scheme does not unrealistically/artificially limit the chemical reactivity to within a fixed depth of the order of a few micrometres and has consistent asymptotic limits."

As we said in our Response #1 to a comment by Referee #1 and in the paper, there is only a small difference between the new and old two-layer schemes in terms of their performance compared to data. However, it can be said that unlike the old two-layer scheme, the new scheme leads to right results for right reasons. The present work also provides a good understanding of the impact of chemistry-turbulence interactions and how they differ in the new and old deposition schemes.

**Changes in manuscript:** Some changes in the relevant text to better clarify the differences between the schemes.

(22) **Comment:** *Figure 6: Ganzeveld et al. [2009], in their Figure 3a and 3b provide similar analyses for January and July. Unfortunately, they do not show annual mean analyses. However, comparing their data with this Figure 6 reveals some very large differences. While Ganzeveld et al. [2009] report the high latitude oceans exhibiting the highest ozone deposition velocities, this Figure 6 shows that the ocean deposition velocity is highest over the tropical oceans. Isn't that a rather large disagreement that should trigger an in depth analysis and discussion?*

**Response:** The work by Ganzeveld et al. (2009) has been adequately commented upon by Luhar et al. (2017), including their Figure 3a and 3b (see in the latter paper the last para on Page 3761, and also in the first para of Section 7.3 on Page 3762). We do not think it is necessary to repeat that exercise here.

**Changes in manuscript:** None.

(23) **Comment:** *18/9-10: ACCESS-UKCA then seems to differ from other models that seem to overestimate surface ozone [Parrish et al., 2014]?*

**Response:** As shown by Woodhouse et al. (2015) and Luhar et al. (2017) our version of UKCA in ACCESS generally underestimates observed tropospheric ozone, particularly in mid to high latitudes. The potential reasons for the model underestimation of tropospheric ozone include

inaccuracies in the emission fields of precursor species, and shortcomings in chemical or physical processes simulated in the model. However, as mentioned earlier, we determine the ozone deposition flux using the MACC ozone reanalyses and the modelled deposition velocities which do not depend on model's chemistry component. Thus model's performance for ozone is not relevant here.

**Changes in manuscript:** The above response to be included.

(24) **Comment:** *20/6: As mentioned earlier, this seems to disagree with the results from [Ganzeveld et al., 2009]?*

**Response:** That is true, and as mentioned earlier the differences with the Ganzeveld et al. (2009) have been adequately commented upon by Luhar et al. (2017).

**Changes in manuscript:** None.

(25) **Comment:** *20/10: Replace 'concentration' with 'mixing ratio'.*

**Response:** Done.

(26) **Comment:** *21/5: This really should not be called 'uncertainty' then. Maybe use the term 'error bar'.*

**Response:** We now say error bounds.

(27) **Comment:** *26/15-21: In this discussion about the differences between this and the previous studies, changes are attributed to a better representation of the commonly applied constant rc of Wesely's scheme, as already demonstrated by Ganzeveld et al. [2009]. Their process-based approach arrived at a global O3 oceanic deposition budget that was not that different from models using Wesely's constant rc. This, in my opinion, calls for a discussion of how these large differences between these two process-based approaches, one being extended to two layers, only considering I-, and the other one using a single layer but including more reactants including DOM, can be reconciled.*

**Response:** The Referee's is correct in saying that the one-layer, process-based scheme (of Fairall et al. (2007)) implemented by Ganzeveld et al. (2009) gives a global O3 oceanic deposition budget that is not too different from models using Wesely's constant surface resistance ( $r_c$ ) approach. We have clarified the reason for this earlier, and it is discussed at great length by Luhar et al. (2017). The main reason is that the one-layer scheme overestimates the turbulence-chemistry interaction in the waterside viscous sublayer by assuming a turbulent diffusivity that increases linearly with depth. This assumption is not valid for the viscous sublayer. The two-layer approach fixes this problem. What this also implies is that getting the waterside turbulence-chemistry interaction correct in the model formulation is more important than including additional reactants (e.g. DOM).

The topic of DOM is discussed in detail in our reply to Referee #3.

**Changes in manuscript:** As mentioned earlier, a discussion on DOM based on our Response #3 to Referee #1 will be included.

(28) **Comment:** *26/29: ...., whereas that ....*

**Response:** Done.

(29) **Comment:** *27/26: Given my reservations detailed above in my opinion this is a rather subjective and invalid evaluation.*

**Response:** In our work, we developed a new two-layer parameterisation for deposition velocity that builds upon, and corrects a flaw of, the previous process-based schemes, tested it within the limitations of the available data and input information required, and used it with the 10-year MACC global ozone reanalyses for calculating deposition budgets, with a comparison of these budgets with those from other studies.

While we do not agree with Referee's comment, we have clarified various points (e.g. uncertainty, DOM, iodide, and coastal grids) raised by the referee in our replies above, which we think address this particular comment.

**Changes in manuscript:** None.

(30) **Comment:** *28/8: ....deposition, an increase ....burden, and an ....*

**Response:** Done.

---

## Author Comment (AC4) · 20 Dec 2017

**Reply by the authors to the Referee #3's comments on**
"Revising global ozone dry deposition estimates based on a new mechanistic parameterisation for air-sea exchange and the multi-year MACC composition reanalysis" (#acp-2017-768)

**Anonymous Referee #3 (RC3)**

We are grateful to the Referee for his/her comments. In the following, we provide our responses to these comments.

(1) **Comment:** *1 Overview*

*The work described in this manuscript builds on the previous work of Luhar et al., 2017 in which the authors developed a more detailed, process based, two layer parametrization for dry deposition of ozone to oceans. In this study the two layer parametrization is refined and then implemented in the UKCA model. The model output is combined with MACC reanalysis data to calculate new estimates for global ozone deposition to water/oceans and total global ozone deposition. These new estimates are considerably less than current estimates of global ozone deposition. The model output combined with MACC reanalysis data is also used to analyse inter-annual trends in ozone dry deposition.*

**Response:** Thank you for your comment.

**Changes in manuscript:** None.

(2) **Comment:** *2 General comments*

*Overall this manuscript is well written and describes an improved parametrization for ozone dry deposition to water. The improved parametrization addresses uncertainty in deposition of ozone to water, which is the main driver of uncertainty in global ozone dry deposition. The manuscript is generally well laid out and the figures are clear. My main comments refer to Section 2. This section is quite important as it describes the new deposition parametrization, but it is a bit hard to follow.*

*(i) It would helpful if the authors could include a diagram of the different layers that form within the sea surface micro layer (e.g. reaction-diffusion sub-layer, bottom layer) that shows a summary of the processes (e.g. chemistry, chemistry/turbulence and reaction with iodide) that occur for in each layer and the main equations that are used to parameterize these processes.*

*(ii) I think it would also be helpful in Section 2 if the authors could more clearly describe how their improved scheme differs from that described in Luhar et al., 2017.*

**Response and changes in manuscript:** (i) We agree with the referee. We intend to include two diagrams of the different ocean layers (see Figures 1 and 2 attached).

(ii) The old two-layer scheme assumes that chemical reactivity is present only in the reaction-diffusion sublayer (which is a few microns thick) even though chemical reactivity is observed to be present throughout the ocean mixing layer. The new two-layer scheme eliminates this arbitrary assumption. However, as mentioned in the paper in the 1st paragraph on Page 4 and elsewhere, the results obtained by the two schemes are similar. This is because in the old scheme assuming chemical reactivity only in the reaction-diffusion sublayer was necessary to compensate for the overestimation of the impact of turbulence that results from the use of a waterside turbulent diffusivity parameterisation that is not valid very close to the water surface.

We clarify the differences between the two schemes better in the revised paper, and the addition of Figure 2 (attached) helps in understanding these differences.

**(3) Comment:** *3 Specific comments*

*3.1 Section 1*

*P4, L13-14: Consider rephrasing to "A more appropriate parametrisation for Kt which varies with zm in the viscous sublayer..." to improve the readability and meaning of the sentence.*

**Response:** Done.

**(4) Comment:** *P4, L20-22: Could the authors provide a brief description of the "asymptotic behaviour" (also mentioned in the abstract). Or refer the reader to section 2.1.*

**Response:** Done.

**(5) Comment:** *3.2 Section 2*

*P5, L20-22: Consider rephrasing to "The second layer, which is deeper than the reaction-diffusion sublayer,..." to improve the readability and meaning of the sentence.*

**Response:** Done.

**(6) Comment:** *P6, L12-13: Consider rephrasing to "The first two, namely the flux at the water surface (z = 0) obtained using Eq. (4) should be equal to F0 and the concentration at the interface..." to improve the readability and meaning of the sentence.*

**Response:** Done.

**(7) Comment:** *Figure 1 caption: Consider rephrasing to "Figure 1: Variation of the oceanic component of ozone dry deposition velocity multiplied by ozone solubility as a function of sea surface temperature (SST, °C), (a, c); and reactivity a (s-1), (b, d). Curves determined using the two-layer deposition scheme (Eq. (16)) for several c0 values used in $\delta m = c0 \, l \, m$, (a, b) and several $\delta m$ values, (c, d). The variations obtained using the one-layer deposition scheme with (Eq. (18)) and without (Eq. (19)) waterside turbulent transfer (i.e. reaction-diffusion only) are also shown. The waterside friction velocity ( u\*w ) used was 0.01 m s-1." to improve the readability.*

**Response:** Done

**(8) Comment:** *3.3 Section 5.2*

*P21, L5-10: Can the authors suggest why there are larger ozone dry deposition velocities in the Northern Hemisphere?*

**Response and changes in manuscript:** We state in the paper that "Oceanic deposition in the Northern Hemisphere ($49.0 \pm 3.4$ Tg yr$^{-1}$) is somewhat larger than that in the Southern Hemisphere ($44.9 \pm 4.5$ Tg yr$^{-1}$) due to the higher $O_3$ concentrations and slightly larger oceanic deposition velocities in the former, although the Earth's area covered by the ocean is larger by approximately 30% in the Southern Hemisphere."

The average oceanic deposition velocity (weighted by the grid-cell area) for the Northern Hemisphere is slightly larger than that for the Southern Hemisphere (i.e. 0.020 vs. 0.017 cm s$^{-1}$ for the year 2005). The main reason for this difference is that the average sea surface temperature (SST) (weighted by the grid-cell area) for the Northern Hemisphere is larger than that for the Southern Hemisphere (i.e. 295.3 K vs. 291.2 K for the same year). As mentioned in the paper, in our formulation of deposition velocity to the ocean is dominated by the surface-resistance term ($r_c$) which in turn depends on SST. Overall the higher the SST the higher the oceanic deposition velocity.

We now include the above clarification in the paper.

[Figure]

**Figure 1: Idealised representation of the vertical structure of the top few metres of sea water. The depth of the reaction-diffusion sublayer will vary according to the chemical reactivity of the ocean water to ozone**

[Figure]

**Figure 2: A simplified model version of Figure 1 used and the processes included in the calculation ozone dry deposition to sea water.**

---

## Author Response (AR1)

**Reply by the authors to the Referee #1's comments on**

"A revised global ozone dry deposition estimate based on a new two-layer parameterisation for air-sea exchange and the multi-year MACC composition reanalysis" (#acp-2017-768)

**Anonymous Referee #1 (RC1)**

We are grateful to the Referee for his/her comments. In the following, we provide our responses to these comments. The locations of the changes made refer to those in the non-tracked version of the revised manuscript submitted.

**(1) Comment:** *This paper proposes some updates to the paper the authors published earlier this year (10.5194/acp-17-3749-2017) describing the deposition of ozone to the ocean. Some changes are made to the parameterization and the resulting deposition velocities are used to explore the impacts on the global budget of ozone on the ACCESS-UK model. They have concerns about the veracity of their atmospheric chemistry model so explore the impact of the new deposition velocities with the MACC reanalysis. They conclude that their new parameterization has some skill in representing the rather sparse observational dataset and that with this new parameterization for deposition velocities, the mass of ozone deposited to the ocean is significantly reduced with implications for both the budget and distribution of ozone.*

*I have concerns that this paper represents a small incremental advance over the previously published paper. For example, Figure 5 only shows small difference between the new and old schemes which was published only a few months ago. Ideally this paper would have been coupled into the paper published only a few months ago. However, this is a decision to be made by the editor.*

**Response:** The work conducted more than a year ago by the authors that the Referee is mentioning was published in November 2016 in ACPD and then in March 2017 in ACP (https://doi.org/10.5194/acp-17-3749-2017). This 2017 ACP provided the subsequent impetus and ideas to extend that work, resulting in the present work. The present work is novel in two important ways: First, it derives a new two-layer formula for the waterside ozone deposition velocity (Eq. 16) which corrects a basic flaw in the two-layer scheme reported in the previous paper by including chemical reactivity throughout the oceanic mixing layer (as is observed) rather than just within the reaction-diffusion sublayer a few microns thick. The new model will also apply to any other chemical compounds that are taken up by the oceanic mixing layer. Second, our work makes use of the gridded global MACC ozone reanalyses given at 3 hourly frequency for ten years to better constrain the oceanic ozone dry deposition fluxes.

One could indeed say that the present paper represents an incremental advance (as is the case with much of the scientific research, if not most), and in our opinion this advance is significant.

We emphasise in the paper (see last para on Page 21), and the Referee also notes, that there is only a small difference between the new and old two-layer schemes in terms of performance in simulating oceanic deposition velocity data. However, the difference between the two, if you like, is that unlike the old scheme the new scheme performs well for the right reasons (the old scheme artificially limits chemical reactivity to the reaction-diffusion sublayer to compensate for the overestimation of the impact of waterside turbulence resulting from the particular form of eddy diffusivity used). The present work also demonstrates the importance of chemistry-turbulence interaction. To explain the difference further, if the deposition of another gas that is taken up by reaction in the ocean surface layer was modelled with the old scheme, the results in nearly every case would be inconsistent whereas with the new scheme they would, prima facie, be correct.

**Changes in manuscript:** We clarify the differences between the old and new two-layer schemes better (see Page 4, Lines 16–30; Page 5, Lines 9–15; last para on Page 22; addition of new Figure 2 and its description).

**(2) Comment:** *Fundamentally this paper provides a description of an improved O3 deposition parameterization for the oceans, shows that there is some fit between the observations reasonably well and fundamentally changes the tropospheric budget for ozone especially over oceanic regions. These are important conclusions.*

*I have a few questions and queries to suggest for the improvement and shortening of the paper which I make below. Assuming that these can be made I would recommend publication.*

**Response:** Thanks for the comment.

**Changes in manuscript:** None.

**(3) Comment:** *Major comments.*

*Ocean O3 lifetime.*

*The premise of the paper is that reaction between O3 and I- is the only sink for O3 in the ocean. There is no discussion of the validity of this assumption. There is significant evidence that dissolved organic matter (DOM) may play a significant role in deposition of ozone to the surface (see for example 10.1029/2008GB003301). Yet this isn't discussed in the text. There should be some justification given for ignoring the role of DOM in their calculations.*

**Response:** The open-ocean ozone deposition velocity data of Helmig et al. (2012) we have used for model testing are limited in sample size and contain substantial fluctuations. However, they are the most comprehensive and only ones that have used a surface based eddy-covariance approach which provides a direct way of measuring deposition velocity. The present work demonstrates that the chemical reaction of $O_3$ with dissolved iodide is able to adequately describe these deposition velocity data within the observed scatter and uncertainty in the input parameterisations (e.g. the second-order reaction rate

constant $k$). However, in response to the Referee's comment we have done additional work on the impact of dissolved organic matter (DOM) and this is included in the revised version.

**Changes in manuscript:** The additional work on DOM (or DOC, dissolved organic carbon) is included as a new Section 2.3 "Impact of ozone reaction with dissolved organic carbon (DOC)." The new Table 1 gives model performance statistics when DOC is included.

**(4) Comment:** *The parameterization appears to do a reasonable job of simulating the deposition observation (Figure 4) without the need for an additional ocean side O3 sink. However there has been a tuning of the model (top half of page 12) so it isn't obvious that a missing O3 sink process (such as that offered by DOM) would be 'diagnosed' though a model to measurement comparison. Figure 4 looks very similar to a figure shown in the author's previous paper. It would be useful to show this data in an x-y plot and give some indication of the error associated with the parameterization against the observations.*

**Response and changes in manuscript:** As mentioned in our response on DOM above, the data available and used for model testing are not detailed enough to clearly discern or diagnose the potential impact of other reactions, let alone provide guidance on parameter values (e.g. reaction rate constant). Obviously, given such limitation there is some parameter value fitting, but this is informed by parameter bounds, for example the reaction-diffusion length scale, the asymptotes and the scatter in the iodide-$O_3$ reaction rate constant, and the deposition velocity data.

As suggested by the Referee, we have included an x-y plot of the modelled vs. observed deposition velocities as Figure 8. Additional work conducted on DOM is included as Section 2.3. The differences between the old and new two-layer schemes are now elucidated better (see Changes in manuscript under Response 1 above).

**(5) Comment:** *Our current understanding of DOM, its reactivity to O3 and distribution is poor. However, the authors should discuss the implications of them ignoring the potential DOM sink. Whilst they are doing that they should also discuss the implication of their choice of iodide distribution. They are using the distribution based on the parameterization of McDonald, but the literature also includes the Chance parameterization which gives higher I- concentrations and I think gives a slightly different spread. What are the implications of this?*

*There should be more of a discussion of the uncertainties of the O3 lifetime in the ocean, and how the parameterization tuning to the observations provides some solid ground to base the subsequent budget analysis. What impact do these uncertainties have on the budget?*

**Response:** As demonstrated in our work on DOM, we agree with the Referee that our current understanding of DOM is poor.

In Figure 5, we have added an additional curve as Option 6 which is the same as Option 4 (the latter option is used in our deposition flux calculations and involves a constant $\delta_m$ = 3 microns with $k$ given

by Eq. (21) using only the data of Magi et al. (1997)) but using the iodide concentration parameterisation of Chance et al. (2014). This parameterisation gives larger iodide concentrations than that by MacDonald et al. (2014).

Our results suggest that the deposition velocity data can also be described by the Chance et al. (2014) iodide parameterisation coupled but with smaller values of $k$ (because it is the reactivity $a$, which is the product of the iodide concentration and second-order rate constant $k$, that goes into the deposition velocity calculation) within the uncertainty of the deposition velocity and second-order rate constant data.

Here we are guided and limited by the scant amount of deposition velocity observations that we have. Our deposition velocity scheme is developed based on sound reasoning and with the selected parameters provides a satisfactory comparison with the data. However, as the discussion of Figure 5 suggests, there is uncertainty in parameter values which would eventually be reflected in the deposition flux estimates. We have done additional work to estimate uncertainty in ozone deposition flux taking into account the scatter in the ocean deposition velocity data, and this is described in our response to Comment #5 made by Referee #2 (see new Section 5.3).

**Changes in manuscript:** Sensitivity to Chance et al.'s (2014) iodide parameterisation is included in Figure 5 as option 6 and the behaviour discussed in the text (Page 14, last paragraph). The new Table 1 includes performance measures for various deposition schemes and configurations including the Chance et al. iodide parameterisation. Additional work on uncertainty in ozone deposition flux, which also addresses Comment #5 made by Referee #2, is included as a new Section 5.3 "Uncertainty in annual ozone dry deposition."

**(6) Comment:** *Diagnosing the ozone deposition flux*

*The new parameterization is put into the ACCESS-UKCA model and this gives a global flux of O3 deposition to the ocean of ~ 86 Tg yr-1. The model is known to have a low bias for O3 and so a significant body of work is done to calculate the flux from the MACC analysis fields of O3 and then a bias corrected MACC analysis fields. This lengthens the paper significantly for almost no gain. The canonical value for ocean deposition of O3 is around the 340 Tg yr-1 from the Hardacre study. The new parameterization gives the ACCESS model a deposition of 86 Tg yr-1, the MACC Analysis 93 Tg yr-1 and the bias corrected MACC Analysis 98.4 Tg yr-1. Compared to the Hardacre values these numbers are essentially the same (25%, 27% and 28%) respectively especially when the uncertainty in the parameterization are considered. There are pages of text describing the MACC data but I don't think it substantially changes the conclusions especially as the authors are forced to bias correct the MACC data. Would it not make more sense to bias correct the ACCESS data?*

*My suggestion is to remove this section or to perform the bias correction on the ACCESS data. It doesn't add anything to the story but it makes the document substantially longer.*

**Response:** The multi-year global MACC reanalyses are high-resolution, gridded, quality controlled data on atmospheric composition that are a valuable tool in developing and evaluating modelling schemes.

They have not previously been used for deposition purposes. Their application in the second half of the paper together with the modelled deposition velocity distribution is an important component of our work and is aimed at further reducing the uncertainty by constraining the ozone dry deposition budgets better.

Considering only the lowest layer of the atmosphere, the ozone deposition flux equals the ozone deposition velocity times the ozone concentration. We argue that the MACC analyses provide the best available gridded estimate of the ozone concentration and our oceanic deposition velocity parameterisation the best estimate of the ozone deposition velocity, and, therefore, combined the best estimate of the ozone deposition rate.

From the point of view of ozone deposition to the ocean, the Referee is correct in saying that the deposition figures obtained using ACCESS, and the MACC analysis with and without the bias correction are very similar (i.e. 86.1, 93.9 and 98.4 Tg yr$^{-1}$, respectively). However, when the total global deposition loss is calculated (including ocean, land and sea ice), the respective figures are 566.7, 689.9 and 722.8 Tg yr$^{-1}$ (see the top paragraph on Page 18 and Table 1 of the original manuscript). Thus the underestimation of ozone by ACCESS is reflected more prominently in the deposition to the non-ocean surfaces. We have used the MACC data to derive both oceanic and global deposition estimates, and we do think that these data have been usefully employed in the paper to constrain the deposition losses of ozone.

Note that our modelled deposition velocity distribution that is multiplied with the MACC ozone data to calculate deposition flux does not depend on the chemistry component of the model. Deposition velocity is solely a function of parameters of the physical component of the model (e.g., SST (for reactivity), flow properties and turbulent mixing, and surface characteristics) and prescribed parameters (e.g., ozone molecular diffusivity and solubility).

**Changes in manuscript:** In light of the above response, we do not agree with the Referee to remove the MACC data analysis and it is retained. However, we have included some of the above points as clarification (see Page 19, Lines 22–31). We do not use the ACCESS derived ozone flux. We use the ACCESS derived deposition velocity (which is unaffected by any shortcomings in the atmospheric chemistry scheme) coupled with the MACC ozone for the ozone flux calculation; therefore, we have shortened the description of the chemical component of ACCESS, and deleted the ACCESS-only (i.e. without MACC) derived deposition flux estimates.

**(7) Comment:** *Minor Comment.*

*There should be more details on the performance of the ACCESS physical model. There are no details of performance, parameterization choices etc. There should be more details given. What aspects of the model impact the parameterization used?*

**Response:** ACCESS-UKCA uses the same physical atmosphere component as the UK Met Office's Unified Model (UM) and includes the UK Chemistry and Aerosol (UKCA) model for atmospheric composition (at UM vn8.4). In our simulations, ACCESS-UKCA is basically the same as UM-UKCA

since the ACCESS specific ocean and land-surface components are not invoked. This is because we run the model in atmosphere-only mode with prescribed SSTs, and the UM's original land-surface scheme (JULES) is used.

For UKCA, we cite http://www.ukca.ac.uk, Morgenstern et al. (2009), Abraham et al. (2012), O'Connor (2014) and Woodhouse et al. (2015). The reference Abraham et al. (2012) is available at http://www.ukca.ac.uk/images/b/b1/Umdp_084-umdp84.pdf which includes some detail of the dry deposition scheme (which is based on Wesely (1989, cited)).

For ACCESS, a reference by Bi et al. (2013; http://www.bom.gov.au/amm/docs/2013/bi1.pdf) can be cited. The assimilation of the ERA-Interim meteorological data into ACCESS is described by Uhe and Thatcher (2015; cited).

We use the MACC ozone reanalyses for the deposition flux calculations combined with the modelled deposition velocities. As mentioned earlier, the latter do not depend on the ozone chemistry in the model so ACCESS-UKCA's performance for ozone is not relevant. Deposition velocity in the model is solely a function of parameters of the physical component of the model and prescribed inputs. Therefore, in effect we only use the physical atmosphere component of the model, and this component relevant to our model version is described by Walters et al. (2014, https://www.geosci-model-dev.net/7/361/2014/gmd-7-361-2014.pdf). A list of technical reports on UM given at http://cms.ncas.ac.uk/wiki/Docs/MetOfficeDocs (but accessing them requires username and password).

We show in the paper that our total deposition flux to non-water surfaces is similar to that calculated by other researchers (Page 33, Lines 6–9) and that the main difference lies in the oceanic deposition flux component.

**Changes in manuscript:** Following the above response, we have revised and expanded Section 3 on ACCESS-UKCA. The references of Bi et al. (2013) and Walters et al. (2014) are also included.

**(8) Comment:** *Typo on page 10, line 14 "fullydescribe" missing a space*

**Response:** Done.

**Reply by the authors to the Referee #2's comments on**

"A revised global ozone dry deposition estimate based on a new two-layer parameterisation for air-sea exchange and the multi-year MACC composition reanalysis" (#acp-2017-768)

**Anonymous Referee #2 (RC2)**

We are grateful to the Referee for a long set of comments. In the following, we provide a response to these comments. The locations of the changes made refer to those in the non-tracked version of the revised manuscript submitted.

(1) **Comment:** *A modified version of a recently presented ozone ocean dry deposition scheme [Luhar et al., 2017] is presented. The model performance is evaluated by comparing modelled ozone deposition velocities with previously published data from oceanic cruises. Further, the global ozone ocean flux is modeled based on this new model configuration, yielding a lower oceanic ozone sink than prior estimates. The ozone ocean flux is then compared with the ozone land sink, and a new total global ozone flux estimate is derived.*

**Response:** Thanks for the comment.

**Changes in manuscript:** None.

(2) **Comment:** *Major Comments:*

*While the authors repeatedly highlight their work as being a new 'scheme', as far as I understand this modeling in essence differs only in one aspect (the ocean layer is described in two, rather than a single layer) from their prior ACP publication [Luhar et al., 2017] that was submitted only ten months prior to this current paper. The article claims this ozone flux parameterization and modeling to be a novel development. However, from reading the earlier publication [Luhar et al., 2017] again, and the works by Ganzeveld et al. [2009] and Fairall et al. [2007], it appears that the the physical and ocean biochemical dependency description were mostly adaptations of principles presented in these earlier publications.*

**Response:** All the references mentioned above by the referee are included in our paper. We clearly elucidate what the shortcomings are of the previous model formulations, including the old two-layer formulation used by Luhar et al. (2017, https://doi.org/10.5194/acp-17-3749-2017). We agree that our new two-layer scheme (Eq. 16) includes the same overall physical and ocean biochemical processes as in the studies by these authors, but it improves the mathematical formulation by correcting a major flaw of the old two-layer scheme in that it includes chemical reactivity throughout the oceanic mixing layer (as is observed) rather than just within the top few microns of the water surface. The new model will also apply to any other chemical compounds that are taken up by the oceanic mixing layer. We accept

that this represents an incremental advance in model development, but we believe it is a significant advance. Additionally, our work also makes use of the global, high resolution ozone reanalyses developed under the European MACC program to better constrain the ozone dry deposition budgets. These reanalyses have not previously been used for the ozone deposition problem and thus provide scope for novel application.

**Changes in manuscript:** To consider the Referee's point, we have qualified the title a little better to read "A revised global ozone dry deposition estimate based on a new two-layer parameterisation for air-sea exchange and the multi-year MACC composition reanalysis." We also clarify the differences between the various schemes better in Introduction (Page 4, Lines 6–30).

(3) **Comment:** *In this model the ozone ocean flux description builds exclusively on chemical removal of ozone by reaction with iodide (I-). Consideration of this reaction is not that novel, having been proposed quite some time ago. Other previous work has suggested that, while the I- reaction has high significance, other secondary reactions, such as those with dissolved organic matter (DOM) in the ocean surface microlayer, may play a role in the ozone reaction as well [Ganzeveld et al., 2009; Coleman et al., 2010]. Ganzeveld et al. [2009] showed, for example, that evaluation of the simulated O3 dry deposition velocities with a 1-layer version of the [Fairall et al., 2007] model, including only I- in the calculation of total reactivity, underestimated the measured coastal deposition velocities. The role of dissolved organic matter (DOM)-O3 chemistry was proposed to explain these discrepancies. [Coleman et al., 2010] specifically addressed the role of DOM-O3 chemistry in deposition to the Atlantic Ocean. These authors conclude: "… iodide reactions alone cannot account for observed deposition velocities. Consequently, we suggest a missing chemical sink due to reactions of ozone with organic matter at the air-sea interface." It does not appear that this Luhar et al. article takes this into consideration. The question if and how much uncertainty potentially results from this neglect is not addressed by their publication.*

**Response:** Part of this comment, particularly about DOM, is similar to Comment #3 made by Referee #1. We thus refer to our Response #3 to Referee #1 where we describe additional deposition calculations done with dissolved organic carbon (DOC, equivalent of DOM) included.

One novelty of our work is to provide a better mathematical formulation for the inclusion of waterside chemical reactivity. The work of Luhar et al. (2017) demonstrated clearly the limitation of the one-layer scheme of Fairall et al. (2007) in describing the deposition velocity data of Helmig et al. (2012) (as a result of overestimation of waterside turbulence-chemistry interaction in this scheme). The two-layer scheme used by Luhar et al. (2017) was successful in simulating the deposition velocity data but it employed an arbitrary constraint on chemical reactivity (as stated in Response #1 above). The two-layer model presented here removes this arbitrary constraint and thereby is a more realistic model of ozone interaction with ocean water.

The focus of modelling in our paper is on ozone deposition to open-ocean regions. Using the one-layer Fairall et al. (2007) scheme, Ganzeveld et al. (2009) showed that the inclusion of the $O_3$-DOM reaction (with DOM represented by chlorophyll-a) significantly increased deposition velocity at coastal sites but

gave mixed results compared to observations. For open ocean sites there were only small changes to deposition velocity due to the inclusion of DOM. The work of Coleman et al. (2010) also relates to coastal waters. None of the papers by Fairall et al. (2007), Ganzeveld et al. (2009) and Coleman et al. (2010) has used the more recent open-ocean deposition velocity data of Helmig et al. (2012) (because these data had not been available at the time) that Luhar et al. (2017) and the present work use.

Our work suggests that the ozone-iodide reaction is able to describe the available open-ocean deposition velocity data within the uncertainty of model parameters and the scatter in the data. Clearly more observations are needed to establish the relative role of additional ozone reactions for different ocean regions.

There is some evidence (e.g. Coleman et al., 2010) that ozone deposition velocities over coastal waters are larger than those over open oceans. The deposition approach used for coastal grid cells in our model is qualitatively consistent with that behaviour and is described in our Response #7 below.

We have calculated a measure of uncertainty in the global oceanic and total ozone deposition fluxes and those details are given in our Response #5 below.

**Changes in manuscript:** Additional calculations on DOC are included as a new Section 2.3 "Impact of ozone reaction with dissolved organic carbon (DOC)." The new Table 1 includes performance measures for various deposition schemes and configurations, including when DOC is added. Introduction is revised and new Figure 2 is included to clarify the differences between the various deposition schemes.

(4) **Comment:** *Further, building exclusively on O3 + I- chemistry, the proper description and consideration of I- in the ocean must be of high importance. The article does not provide any detail on what data the I- oceanic description builds on. Are these new observations? Or is the I- modeled based on other relationships? In [Ganzeveld et al., 2009], I- was estimated based on its correlation with nitrate. While this seemed to be a reasonable, and possibly the best possible approach at that time, does this paper take advantage of the much improved I- description presented by Chance et al. [2014]? Despite this progress, there certainly remains large uncertainty in the spatial and temporal representation of I-, e.g. its concentrations in high-latitude waters, which is hampered by a lack of in-situ observations. This is actually the region where, according to this study by Luhar, the largest differences in the O3 dry deposition velocities compared to the older/other deposition approaches are observed (Figure 9 in [Luhar et al., 2017]). As far as I understand, these uncertainties are likely many times larger than the rather narrow uncertainty windows in the ozone deposition budgets that are presented in this new Luhar et al. publication. Unfortunately, the authors do not elaborate on this question, which I consider a severe neglect.*

**Response:** It is clearly stated in our paper that the ocean iodide concentration used is based on Eq. (20), which is from MacDonald et al. (2014). Chance et al. (2014) examined statistical relationships between iodide and parameters such as sea surface temperature (SST), nitrate, salinity, chlorophyll-a, and mixed layer depth and found that SST was the strongest predictor of iodide. MacDonald et al. (2014, with Chance as a co-author) used data from several cruises in the Atlantic and Pacific oceans covering the latitudes 50°S–50°N to derive their parameterisation for iodide concentration, which we have used. A

sensitivity analysis involving the iodide parameterisation of Chance et al. (2014) was reported by Luhar et al. (2017) (see their Figure 5) and it was compared with the behaviour obtained using the MacDonald et al. (2014) iodide parameterisation. We have included some discussion on the use of the Chance et al. parameterisation and included a deposition velocity curve as Option 6 in Figure 5 based on this parameterisation in our Response #5 to Referee #1.

We agree that there is considerable uncertainty in the representation of ocean iodide concentration, particularly in high-latitude waters, due to the lack of in-situ observations. The rather narrow uncertainty windows in our annual ozone deposition fluxes are solely due to the interannual variability inherent in the modelled meteorology and the MACC ozone concentration fields. We have now done additional calculations to determine the uncertainty range better, and this is given as a response to Comment #5 below.

**Changes in manuscript:** The sensitivity to the Chance et al. (2014) iodide parameterisation conducted in our Response #5 to Referee #1 is included in Section 2.2 (see option 6 in Figure, and last paragraph on Page 14). The new Table 1 includes performance measures for various deposition schemes and configurations including the Chance et al. iodide parameterisation. Additional work on uncertainty reported below under Response #5 is also included.

(5) **Comment:** *Developed flux estimates are presented with error windows (see abstract line 23) that are on the order of 5%, but those windows are simply the standard deviation of the year to year variability in the modeled flux based on changing meteorology. They are not the uncertainty in the estimates of the ozone flux. Those, likely, would be much larger, making the way this is presented quite misleading.*

**Response:** The Referee is correct—the reported uncertainty in our annual ozone deposition fluxes is solely due to the interannual variability in the modelled meteorology (with nudging) and the MACC ozone concentration fields. We have done additional calculations to estimate uncertainty in deposition flux to the ocean better and this is described in Section 5.3.

**Changes in manuscript:** A new Section 5.3 "Uncertainty in annual ozone dry deposition." is included. With the revised uncertainty estimation, the global oceanic and total deposition fluxes of ozone are $98.4 \pm 30.0$ Tg yr$^{-1}$ and $722.8 \pm 87.0$ Tg yr$^{-1}$, respectively.

(6) **Comment:** *Secondary analyses, such as comparison of modeled boundary layer ozone, global ocean flux budgets, and attribution of the oceanic flux to the total global flux that build on this modeling, are consequently highly uncertain as well. I therefore question the value of these secondary analyses. For instance, differences between the two schemes shown in Figures 9 and 10 are on the order of 0-25%. Of how much value are these results when the uncertainty in the reactivity is maybe on the order of 100-200%? To me, what I think needs to be addressed most urgently are these questions:*

*- How much of the total oceanic ozone flux can be attributed to I-, versus other reactants?*

*- What are the oceanic I- fields? How does I- change with time and location? And how can this variability be best incorporated into the model?*

*Unfortunately, these questions are not identified and addressed in this paper.*

**Response:** Please see our Response #3 to Referee #1 on DOM which demonstrates that the ozone-iodide chemistry is sufficient to describe the available open-ocean $v_d$ measurements and their dependency on SST, and that the inclusion of DOM would deteriorate model performance.

As mentioned above, we used the iodide parameterisation of MacDonald et al. (2014) which is based on data from several cruises in the Atlantic and Pacific oceans covering the latitudes 50°S–50°N. This parameterisation is a function of SST, so depends on location and time. We have done additional uncertainty calculations as mentioned above. See Response #5 to Referee #1 on the Chance et al. (2014) iodide parameterisation.

**Changes in manuscript:** Additional calculations on DOM included in Section 2.3. Details of the iodide parameterisation used are provided on Page 10, Lines 12–15. Sensitivity to Chance et al.'s (2014) iodide parameterisation is included as option 6 in Figure 5, the behaviour discussed, and the performance of this parameterisation reported in the new Table 1.

(7) **Comment:** *Other Comments:*

*The Bariteau et al. [2010] article makes a point that ozone fluxes are higher near the coasts compared to the open ocean. Was that considered in this modeling? And if not, how much uncertainty is potentially due to this neglect?*

**Response:** In our global modelling, the coastal grid cells that include terrestrial surface fractions are handled as follows.

A grid-box mean deposition velocity is calculated using the individual modelled deposition velocities weighted by the fractions of the surface types present in the grid box. Our two-layer deposition scheme for the ocean is only used when the fraction of water surface in a grid box is greater than 60%. In all other cases Wesely's (1989) scheme for $v_d$ is used, including the use of $r_c = 2200$ s m$^{-1}$ for water surface. Thus for coastal water grid cells that include fractions of other surface types, the modelled deposition velocities are greater than those for the grid cells fully covered by water (as evident in revised Figure 9) because terrestrial surfaces have higher deposition velocities than water and also because of the use of $r_c = 2200$ s m$^{-1}$ for the water surface tile when its fraction is less than 60%.

There is some evidence that the ozone deposition velocities over coastal waters are larger than those over open oceans (e.g. Coleman et al., 2010; Bariteau et al., 2010), which could be due to factors such as advection from land if the distance between the monitor and coastline (i.e. fetch) is limited, stronger chemical reactivity and turbulence etc. Our approach for treating coastal grid cells is qualitatively consistent with ozone deposition velocities over coastal waters being larger than over the open sea. But we do not include any additional/special processes for coastal waters.

**Changes in manuscript:** The above details are included in the paper (see Page 19, Lines 14–21; revised Figure 9 and its description on Page 26, Lines 8–18). The revised Figure 9 now includes coastal grid boxes with deposition velocities.

(8) **Comment:** *The performance of the deposition model leans heavily on data from the six open ocean cruises shown in Figures 4 and 5. Did the authors attempt other comparisons, for instance using any of the other data sets that were summarized in [Helmig et al., 2012]? Given that, as currently done, it appears that the validation relies exclusively on the data from a single group, it should be shown that those cruises are representative for the entirety of available data. Furthermore, these data do not appear to be publicly available, or hosted by any data center? In our research center (and I think this is becoming more common within the community) it is customary to cite the doi of the data set, invite the providers of the data for co-authorship, or at least acknowledge the data providers, whenever those data make a significant contribution to a publication, including comparisons in modeling studies.*

**Response:** It is correct that our model performance testing is based on published data from Helmig et al. (2012) which cover the latitudinal range 45°N–50°S. Surface based ozone flux stations employing the eddy-covariance technique enables a direct measurement of ozone dry deposition velocity. The data of Helmig et al. (2012) are the only such measurements available over the ocean.

We looked up the very sparse datasets by other researchers summarised by the above authors. None of these studies involved surface-based eddy-covariance technique over the ocean. The ones that used such a technique were coastal measurements (i.e. Gallagher et al., 2001; Whitehead et al., 2010). The measurements by Lenschow et al. (1982) and Kawa and Pearson (1989) used aircraft-based eddy-covariance over the ocean.

The considerably larger sample size and the (perceived) use of improved instrumentation and analysis techniques in the cruise measurements of Helmig et al. (2012) compared to those reported by earlier studies provide an incentive to use these data.

In the original manuscript, on lines 23–26 on Page 14 we say "As in Luhar et al. (2017), the $v_d$ versus SST cruise data used for comparison with the model are those with the wind speed dependence retained (Ludovic Bariteau, personal communication, 2016) and not the data originally reported by Helmig et al. (2012) in which the wind-speed dependence was removed. While this approach is logically correct, there is not a large difference between the data with and without the wind-speed dependence." The same is stated in the revised paper.

**Changes in manuscript:** The above response is summarised on Page 20, Lines 5–11. Dr. L. Bariteau is now included in the acknowledgements (as was done in Luhar et al. (2017)).

(9) **Comment:** *Page 1/Line 11: I don't see what the term 'consistent' qualifies in this context (consistent with what?)? So, I recommend deleting this.*

**Response:** Done.

**(10) Comment:** *1/17: As detailed above, I think the term 'new' is a bit of an overstatement. Yes, this paper does present some advancements in the ozone ocean uptake modeling, but most of the mechanisms, considered reactants, and dependencies were presented in prior publications.*

**Changes in manuscript:** At this location, we now use 'two-layer' in place of 'new'. Also, as we mentioned earlier, we have changed the paper title slightly to read "A revised global ozone dry deposition estimate based on a new two-layer parameterisation for air-sea exchange and the multi-year MACC composition reanalysis."

**(11) Comment:** *1/25: Atmospheric models appear to mostly overestimate surface ozone [Parrish et al., 2014]. The results presented in this Luhar et al. manuscript show an increase of modeled ozone, thereby further increasing the discrepancy between models and observations. So, from that perspective, don't these changes go in the wrong direction?*

**Response:** The ACCMIP multi-model study by Young et al. (2013) shows an overall overestimation of ozone in the lower troposphere in northern mid-latitudes and underestimation in southern tropics and mid-latitudes. Parrish et al. (2014) also show that models overestimate ozone in northern mid-latitudes. Our version of UKCA in ACCESS generally underestimates observed tropospheric ozone, particularly in mid to high latitudes.

Our aim is to improve the process modelling of ozone deposition to the ocean. If a model performs worse when a particular process is improved then this points to issues with some other component(s) of the model.

To constrain the deposition fluxes better, we have used the MACC ozone reanalyses and in that case the deposition fluxes do not depend on ACCESS-UKCA's ozone chemistry because the modelled deposition velocity field used in the deposition flux calculation is solely a function of the physical component of the model and input parameterisations (e.g., SST, flow properties and turbulent mixing, reactivity, ozone molecular diffusivity and solubility in water, and surface characteristics).

**Changes in manuscript:** We note that because we have used the MACC ozone reanalyses the deposition fluxes do not depend on ACCESS-UKCA's ozone chemistry (Page 19, Lines 22–25).

**(12) Comment:** *3/1: Is this ('commonly') indeed still the case, given that Ganzeveld et al. [2009] published a process-based parameterization and model implementation some 8 years ago?*

**Response:** To our knowledge, all common global chemistry models reported in the literature continue to use the Wesely (1989) approach for deposition to the ocean (involving a constant value of $r_c$ of around 2000 s m$^{-1}$ for the surface resistance). Luhar et al. (2017) showed that the Wesely approach overestimates the deposition velocity data of Helmig et al. (2012). Ganzeveld et al. (2009) included the Fairall et al. (2007) one-layer scheme for ozone deposition velocity to the ocean in a global model, and found that compared to the Wesely approach it leads to only a small (6%) reduction in the total oceanic

deposition of ozone. Luhar et al. (2017) showed that Fairall et al.'s one-layer scheme overestimates the deposition velocity data due to flaws with its turbulence diffusivity assumption. Using better assumptions, the two-layer approach as presented in our paper describes these data much better.

**Changes in manuscript:** A summary of the above response is included in Introduction (Page 4, Lines 6–30).

(13) **Comment:** *3/9: Ganzeveld et al. [2009] should also be cited here?*

**Response:** We do not think the Ganzeveld et al. (2009) reference is appropriate here because they used a one-layer scheme (of Fairall et al., 2007) and not a two-layer scheme.

**Changes in manuscript:** None.

(14) **Comment:** *3/28: How is the oceanic layer between the surface and 10 m depth represented?*

**Response:** The assumption regarding how the top oceanic layer is represented goes in the derivation of the expression for the waterside component of deposition velocity (or conversely surface resistance). This expression is implemented in the dry deposition module of our atmosphere-only global model. The global model itself does not include any explicit oceanic layer since there is no ocean model coupled. The prescribed SSTs are used in some of the input parameterisations for the waterside deposition velocity (Eq. 16) obtained using the two-layer scheme. Our scheme assumes that: chemical reactivity (or reactant) is present throughout the oceanic mixing layer; in the top few micros of the oceanic layer ozone loss is dominated by chemical reaction (with no turbulent transfer); and in the oceanic layer below, both chemical reaction and turbulent transfer act together. We now provide a diagram (as suggested by Referee #3) to make this clearer (see our Response #2 to Referee #3).

**Changes in manuscript:** A new diagram (Figure 2) is provided to make this clearer (also see our Response #2 to Referee #3).

(15) **Comment:** *3/24: As mentioned earlier already, a section is needed here explaining how oceanic I-concentrations were derived and included in the modeling.*

**Response:** The following paragraph is added:

**Changes in manuscript:** On page 10, Lines 12–17, we add "This parameterisation is based on iodide data from cruises in the Atlantic and Pacific oceans covering the latitudes 50°S to 50°N, and is a function of SST (Ts (K)) which varies with location and time. Eq. (20) yields highest iodide concentrations in warm tropical waters and lowest in cool waters at higher latitudes. Chance et al. (2014) examined statistical relationships between iodide and parameters such as SST, nitrate, salinity, chlorophyll-a and mixed layer depth, and found that SST was the strongest predictor of iodide in surface waters. Ganzeveld et al. (2009) used oceanic surface nitrate as a proxy for iodide." Sensitivity to

Chance et al.'s (2014) iodide parameterisation is included as option 6 in Figure 5 and the behaviour discussed. The performance of this parameterisation reported in the new Table 1.

(16) **Comment:** *4/14: ...considered, but a ....*

**Response:** Done.

(17) **Comment:** *5/1: Why 'consistent' ?*

**Response:** The word has been deleted.

(18) **Comment:** *5/21: .... included, and a ....*

**Response:** Done.

(19) **Comment:** *16/1: ....that the new ....*

**Response:** Done.

(20) **Comment:** *16/2: .....(2017), but unlike the latter, the new ....*

**Response:** Done.

(21) **Comment:** *Figure 5: As mentioned earlier, this figure nicely shows that improvements made through this work are merely nuances, while very large uncertainties and deficiencies in other areas are overlooked.*

**Response:** As we said in our Response #1 to a comment by Referee #1 and in the paper, there is only a small difference between the new and old two-layer schemes in terms of their performance compared to the data. However, it can be said that unlike the old two-layer scheme, the new scheme leads to right results for right reasons. The present work also emphasises the importance of the impact of chemistry-turbulence interactions on deposition.

**Changes in manuscript:** Some changes made in the relevant text to better clarify the differences between the schemes (Page 4, Lines 16–30; Page 5, Lines 9–15; last para on Page 21). Figure 5 in the original manuscript is deleted as it does not add much to what is stated in words, i.e. the model-data agreement using the new two-layer scheme is very similar to that obtained by the old two-layer reactivity scheme (Page 21, Lines 24–27). A new Table 1 is added to report model-data comparison

statistics for various deposition schemes/configurations considered. The questions about uncertainties are considered in Responses #3–6.

(22) **Comment:** *Figure 6: Ganzeveld et al. [2009], in their Figure 3a and 3b provide similar analyses for January and July. Unfortunately, they do not show annual mean analyses. However, comparing their data with this Figure 6 reveals some very large differences. While Ganzeveld et al. [2009] report the high latitude oceans exhibiting the highest ozone deposition velocities, this Figure 6 shows that the ocean deposition velocity is highest over the tropical oceans. Isn't that a rather large disagreement that should trigger an in depth analysis and discussion?*

**Response:** The work by Ganzeveld et al. (2009) has been adequately commented upon by Luhar et al. (2017), including their Figure 3a and 3b (see in the latter paper the last para on Page 3761, and also in the first para of Section 7.3 on Page 3762). We do not think it is necessary to repeat that exercise here.

**Changes in manuscript:** None.

(23) **Comment:** *18/9-10: ACCESS-UKCA then seems to differ from other models that seem to overestimate surface ozone [Parrish et al., 2014]?*

**Response:** As shown by Woodhouse et al. (2015) and Luhar et al. (2017) our version of UKCA in ACCESS generally underestimates observed tropospheric ozone, particularly in mid to high latitudes. The potential reasons for the model underestimation of tropospheric ozone include inaccuracies in the emission fields of precursor species, and shortcomings in chemical or physical processes simulated in the model. However, as mentioned earlier, we determine the ozone deposition flux using the MACC ozone reanalyses and the modelled deposition velocities which do not depend on ACCESS-UKCA's ozone chemistry. Thus this model's performance for ozone is not relevant in our paper.

**Changes in manuscript:** A clarification is given in the last two paragraphs of Section 3.

(24) **Comment:** *20/6: As mentioned earlier, this seems to disagree with the results from [Ganzeveld et al., 2009]?*

**Response:** That is true, and as mentioned earlier the differences with the Ganzeveld et al. (2009) have been adequately commented upon by Luhar et al. (2017).

**Changes in manuscript:** None.

(25) **Comment:** *20/10: Replace 'concentration' with 'mixing ratio'.*

**Response:** Done.

**(26) Comment:** *21/5: This really should not be called 'uncertainty' then. Maybe use the term 'error bar'.*

**Response:** We now say error bounds.

**(27) Comment:** *26/15-21: In this discussion about the differences between this and the previous studies, changes are attributed to a better representation of the commonly applied constant rc of Wesely's scheme, as already demonstrated by Ganzeveld et al. [2009]. Their process-based approach arrived at a global O3 oceanic deposition budget that was not that different from models using Wesely's constant rc. This, in my opinion, calls for a discussion of how these large differences between these two process-based approaches, one being extended to two layers, only considering I-, and the other one using a single layer but including more reactants including DOM, can be reconciled.*

**Response:** The Referee's is correct in saying that the one-layer, process-based scheme (of Fairall et al. (2007)) implemented by Ganzeveld et al. (2009) gives a global $O_3$ oceanic deposition budget that is not too different from models using Wesely's constant surface resistance ($r_c$) approach. We have clarified the reason for this earlier, and it is discussed at length by Luhar et al. (2017). The main reason for that is that the one-layer scheme overestimates the turbulence-chemistry interaction in the waterside viscous sublayer by assuming a turbulent diffusivity that increases linearly with depth. This assumption is not consistent with existing knowledge about turbulence in the viscous sublayer (Fairall et al., 2000). The two-layer approach eliminates this problem. What this also implies is that getting the waterside turbulence-chemistry interaction correct in the model formulation can be more important than including additional reactants (e.g. DOM).

The topic of DOM is discussed in detail in our reply to Referee #3.

**Changes in manuscript:** As mentioned earlier, a discussion on DOM based on our Response #3 to Referee #1 is included as a new Section 2.3. The Introduction is revised to clarify the differences between the various schemes better. New Table 1 and new Figure 2 are also added.

**(28) Comment:** *26/29: …., whereas that ….*

**Response:** Done.

**(29) Comment:** *27/26: Given my reservations detailed above in my opinion this is a rather subjective and invalid evaluation.*

**Response:** In our work, we developed a new two-layer parameterisation for deposition velocity that builds upon, and corrects a flaw of, the previous process-based schemes, tested it within the limitations of the available data and input information required, and used it with the 10-year MACC global ozone reanalyses for calculating deposition budgets, with a comparison of these budgets with those from other studies.

While we do not agree with Referee's comment, we have clarified various points (e.g. uncertainty, DOM, iodide, and coastal grids) raised by the referee in our replies above, which we think address this particular comment.

**Changes in manuscript:** None.

(30) **Comment:** *28/8: ….deposition, an increase ….burden, and an ….*

**Response:** Done.

**Reply by the authors to the Referee #3's comments on**

"A revised global ozone dry deposition estimate based on a new two-layer parameterisation for air-sea exchange and the multi-year MACC composition reanalysis" (#acp-2017-768)

**Anonymous Referee #3 (RC3)**

We are grateful to the Referee for his/her comments. In the following, we provide our responses to these comments. The locations of the changes made refer to those in the non-tracked version of the revised manuscript.

(1) **Comment:** *1 Overview*

*The work described in this manuscript builds on the previous work of Luhar et al., 2017 in which the authors developed a more detailed, process based, two layer parametrization for dry deposition of ozone to oceans. In this study the two layer parametrization is refined and then implemented in the UKCA model. The model output is combined with MACC reanalysis data to calculate new estimates for global ozone deposition to water/oceans and total global ozone deposition. These new estimates are considerably less than current estimates of global ozone deposition. The model output combined with MACC reanalysis data is also used to analyse inter-annual trends in ozone dry deposition.*

**Response:** Thank you for your comment.

**Changes in manuscript:** None.

(2) **Comment:** *2 General comments*

*Overall this manuscript is well written and describes an improved parametrization for ozone dry deposition to water. The improved parametrization addresses uncertainty in deposition of ozone to water, which is the main driver of uncertainty in global ozone dry deposition. The manuscript is generally well laid out and the figures are clear. My main comments refer to Section 2. This section is quite important as it describes the new deposition parametrization, but it is a bit hard to follow.*

*(i) It would helpful if the authors could include a diagram of the different layers that form within the sea surface micro layer (e.g. reaction-diffusion sub-layer, bottom layer) that shows a summary of the processes (e.g. chemistry, chemistry/turbulence and reaction with iodide) that occur for in each layer and the main equations that are used to parameterize these processes.*

*(ii) I think it would also be helpful in Section 2 if the authors could more clearly describe how their improved scheme differs from that described in Luhar et al., 2017.*

**Response and changes in manuscript:** (i) We agree with the referee. We include two new diagrams to illustrate the different ocean layers (see Figures 1 and 2).

(ii) The old two-layer scheme assumes that chemical reactivity is present only in the reaction-diffusion sublayer a few microns thick even though chemical reactivity is observed to be present throughout the ocean mixing layer. The new two-layer scheme eliminates this arbitrary assumption. However, as mentioned in the paper in the last paragraphs on Pages 4 and 21, the results obtained by the two schemes are similar. This is because in the old scheme assuming chemical reactivity only in the reaction-diffusion sublayer was necessary to compensate for the overestimation of the impact of turbulence that results from the use of a waterside turbulent diffusivity parameterisation that is not valid very close to the water surface. The new scheme overcomes that limitation by assuming that only molecular diffusion occurs in the reaction-diffusion sublayer and from then on both molecular diffusion and turbulence are present. Chemical reaction takes place in both layers. This is an approximation to the nature of mixing near surfaces.

We clarify the differences between the two schemes better in Introduction (Page 4, Lines 16–30; Page 5, Lines 1–15) and through the addition of new Figure 2.

**(3) Comment:** *3 Specific comments*

*3.1 Section 1*

*P4, L13-14: Consider rephrasing to "A more appropriate parametrisation for Kt which varies with zm in the viscous sublayer..." to improve the readability and meaning of the sentence.*

**Response:** Done.

**(4) Comment:** *P4, L20-22: Could the authors provide a brief description of the "asymptotic behaviour" (also mentioned in the abstract). Or refer the reader to section 2.1.*

**Response:** Done.

**(5) Comment:** *3.2 Section 2*

*P5, L20-22: Consider rephrasing to "The second layer, which is deeper than the reaction-diffusion sublayer,..." to improve the readability and meaning of the sentence.*

**Response:** Done.

**(6) Comment:** *P6, L12-13: Consider rephrasing to "The first two, namely the flux at the water surface (z = 0) obtained using Eq. (4) should be equal to F0 and the concentration at the interface..." to improve the readability and meaning of the sentence.*

**Response:** Done.

**(7) Comment:** *Figure 1 caption: Consider rephrasing to "Figure 1: Variation of the oceanic component of ozone dry deposition velocity multiplied by ozone solubility as a function of sea surface temperature (SST, °C), (a, c); and reactivity a (s-1), (b, d). Curves determined using the two-layer deposition scheme (Eq. (16)) for several c0 values used in $\delta m = c0 \, l \, m$, (a, b) and several $\delta m$ values, (c, d). The variations obtained using the one-layer deposition scheme with (Eq. (18)) and without (Eq. (19)) waterside turbulent transfer (i.e. reaction-diffusion only) are also shown. The waterside friction velocity ($u*w$) used was 0.01 m s-1." to improve the readability.*

**Response:** Done

**(8) Comment:** *3.3 Section 5.2*

*P21, L5-10: Can the authors suggest why there are larger ozone dry deposition velocities in the Northern Hemisphere?*

**Response and changes in manuscript:** We state in the paper that "Oceanic deposition in the Northern Hemisphere ($49.0 \pm 3.4$ Tg yr$^{-1}$) is somewhat larger than that in the Southern Hemisphere ($44.9 \pm 4.5$ Tg yr$^{-1}$) due to the higher $O_3$ concentrations and slightly larger oceanic deposition velocities in the former, although the Earth's area covered by the ocean is larger by approximately 30% in the Southern Hemisphere."

The average oceanic deposition velocity (weighted by the grid-cell area) for the Northern Hemisphere is slightly larger than that for the Southern Hemisphere (i.e. 0.020 vs. 0.017 cm s$^{-1}$ for the year 2005). The main reason for this difference is that the average sea surface temperature (SST) (weighted by the grid-cell area) for the Northern Hemisphere is larger than that for the Southern Hemisphere (i.e. 295.3 K vs. 291.2 K for the same year). As mentioned in the paper, in our formulation of deposition velocity to the ocean is dominated by the surface-resistance term ($r_c$) which in turn depends on SST. Overall the higher the SST the higher the oceanic deposition velocity.

We include the above clarification on Page 29, Lines 14–18.

**Reply by the authors to L. J. Carpenter's comment on**

"A revised global ozone dry deposition estimate based on a new two-layer parameterisation for air-sea exchange and the multi-year MACC composition reanalysis" (#acp-2017-768)

**Comment:** *I enjoyed reading this paper which carefully lays out improvements to the authors' previous oceanic O3 dry deposition formulation by including chemical reactivity below the reaction-diffusion sublayer. I have a question which I don't think any of the reviewers raise, on the reaction-diffusion sublayer thickness: how were the values of the constant c0 chosen?*

**Response and changes in manuscript:** We thank Prof. Lucy Carpenter for her views on our work.

In the reaction-diffusion sublayer, $l_m = (D/a)^{1/2}$ is an appropriate length scale. Thus, using scaling argument, it is reasonable to assume that the thickness of the reaction-diffusion sublayer ($\delta_m$) is proportional to $l_m$ with the coefficient of proportionality ($c_0$) being a constant of the order unity. In Figures 1a and 1b of our paper, we plot $1/r_c$ curves for three values of $c_0$, viz. 0.2, 0.4 and 0.7, which fall within the two asymptotic limits (equivalent to $c_0 \to 0$ and $c_0 \to \infty$). The value $c_0 = 0.4$ was selected for further sensitivity analysis reported in Figure 3 because it leads to a $1/r_c$ variation that roughly lies in the middle of the two asymptotic limits as shown in Figures 1a and 1b. As mentioned on Page 12 Line 13, in all our subsequent deposition calculations we used Option 4 with $\delta_m = 3$ microns (see the 1$^{st}$ para on Page 13) which obviously does not need a specification of $c_0$ (but of course there will be an implied variation of $c_0$ via the relation $c_0 = \delta_m / l_m$).

We include the above clarification in the revised version of the paper (Page 14, Lines 5–9).

[revised manuscript text omitted]